# Bathymetry-constrained warm-mode melt estimates derived from analysing Oceanic Gateways in Antarctica

Lena Nicola[1,2], Ronja Reese[1,3], Moritz Kreuzer[1,2], Torsten Albrecht[1,4], and Ricarda Winkelmann[1,2,4]

[1] Potsdam Institute for Climate Impact Research (PIK), Member of the Leibniz Association, P.O. Box 60 12 03, D-14412 Potsdam, Germany
[2] Institute of Physics and Astronomy, University of Potsdam, Karl-Liebknecht-Str. 24-25, 14476 Potsdam, Germany
[3] Department of Geography and Environmental Sciences, Northumbria University, Ellison Place, NE1 8ST, Newcastle Upon Tyne, UK
[4] Department of Integrative Earth System Science, Max Planck Institute of Geoanthropology, Kahlaische Str. 10, 07745 Jena, Germany

**Correspondence:** Lena Nicola (lena.nicola@pik-potsdam.de) and Ricarda Winkelmann (ricarda.winkelmann@pik-potsdam.de)

**Abstract.** Melting underneath the floating ice shelves surrounding the Antarctic continent is a key process for the current and future mass loss of the Antarctic Ice Sheet. Troughs and sills on the continental shelf play a crucial role in modulating sub-shelf melt rates, as they can allow or block the access of relatively warm, modified Circumpolar Deep Water to ice-shelf cavities. Here we identify potential oceanic gateways in at least 7 out of 19 regions subdividing the Antarctic continent that could channel warm water masses to Antarctic grounding lines, based on access depths inferred from high-resolution bathymetry data. We analyse the properties of water masses that are currently present in front of the ice shelf and that might intrude into the respective ice-shelf cavities in the future in case of changes in the ocean circulation. We use the ice-shelf cavity model PICO to estimate an upper bound of melt rate changes in case off-shore, intermediate layer warm water masses gain access to the cavities. Depending on the presence of an oceanic gateway and the current ice-shelf melt conditions, we find up to 42-fold larger basal melt rates. The identification of oceanic gateways is thus valuable for assessing the potential of ice-shelf cavities to switch from a 'cold' to a 'warm' state, which could result in widespread ice loss from Antarctica.

## 1 Introduction

The current mass loss from the Antarctic Ice Sheet is mainly triggered by thinning of the surrounding ice shelves (Pritchard et al., 2012; Paolo et al., 2015; Gudmundsson et al., 2019). This is caused by ice-shelf basal melting, that varies by orders of magnitude depending on the prevailing ocean conditions: a sub-shelf circulation that is initiated by sea-ice formation or tidal pumping and driven by the so-called 'ice pump' (mode 1 or 3-melting in Jacobs et al., 1992, respectively) causes melt rates at the order of centimetres to a few metres per year. For example, area-averaged observed melt rates at Filchner–Ronne Ice Shelf are around $0.3\pm0.1\,\mathrm{m\,yr^{-1}}$ (Ronne) and $0.4\pm0.1\,\mathrm{m\,yr^{-1}}$ (Filchner) as estimated by Rignot et al. (2013). In these ice shelves, mode 1 melting plays a major role towards the grounding line and mode 3 melting near the ice-shelf front (Silvano et al., 2016). Where melting is driven by Dense Shelf Water (DSW, mode 1) or surface waters (mode 3), generally water masses close to

the surface freezing point are present within the cavity which can be hence classified as 'cold' – such as for Filchner–Ronne, Ross or Amery Ice Shelves (Joughin et al., 2012; Silvano et al., 2016). DSW, due to the higher density from e.g. brine rejection from sea-ice formation, sinks to the ocean floor and spreads to the grounding line (Silvano et al., 2016). Ice-shelf thinning and upstream mass loss are currently not observed in these 'cold'-cavity regions (Joughin et al., 2012; Paolo et al., 2015; Greene et al., 2022). A different mode of sub-shelf melting is driven by an inflow of water masses from the continental slope (mode 2-melting in Jacobs et al., 1992), bringing water with temperatures well above the pressure-melting point into the ice-shelf cavity. Such cavities can be classified as 'warm' (Joughin et al., 2012). They experience melt rates up to the order of tens of metres per year (cf. area-average basal melt rates for Pine Island and Thwaites in Rignot et al., 2013).

The exchange of water masses between the continental shelf and the open ocean is strongly influenced by bathymetry (Thoma et al., 2008; Nicholls et al., 2009; Hellmer et al., 2012; Pritchard et al., 2012; Tinto et al., 2019; Sun et al., 2022), but the processes that lead to on-shelf transport of warm water masses, leading to a switch to a 'warm' cavity or mode 2-melting, are highly complex and an active field of research. To what extent the inflow of warm waters from the continental-shelf break into ice-shelf cavities can be related to anthropogenic changes (Holland et al., 2022) or natural variability (Jenkins et al., 2016, 2018) alone, remains to be determined. Once, however, warmer water masses enter an ice-shelf cavity, this can lead to a strong increase in sub-shelf melt rates and in further consequence can cause the adjacent ice streams to thin, accelerate, and retreat. The highest thinning rates in Antarctica are found for ice shelves in the Amundsen Sea, where relatively warm, modified Circumpolar Deep Water (mCDW) accesses the ice shelves at depth through submarine troughs (Nitsche et al., 2007; Walker et al., 2007; De Rydt et al., 2014; Mouginot et al., 2014; Jenkins et al., 2016; Millan et al., 2017; Naughten et al., 2023). This mCDW comprises relatively warm and salty water masses which reside at mid-depth, on average at around $500\,\mathrm{m}$, in the Southern Ocean in front of the continental shelf (Schmidtko et al., 2014; Holland et al., 2020).

Ocean access to ice-shelf cavities is often modulated by geological structures on the continental shelf that block or channel the distal inflow of deeper and warmer water masses off the continental shelf, i.e. CDW. The abyssal Southern Ocean bathymetry raises towards the continent to form the shallow continental shelf that has a mean depth of about $500\,\mathrm{m}$ (Heywood et al., 2014), with the transition zone being called the continental-shelf break (CSB). The width of the continental shelf varies around Antarctica from tens of kilometres, in East Antarctica or the West Antarctic Peninsula, to hundreds of kilometres in the Ross or Weddell Sea (Heywood et al., 2014). While large data gaps still exist, recent Antarctic bathymetry data incorporate major glacial troughs, ridges or other features of basal topography crosscutting the continental shelf (Arndt et al., 2013; Morlighem et al., 2020). These bathymetric features were mostly formed by erosion and sedimentation due to dynamic changes of the ice sheet during glacial cycles, e.g. ice streams leaving behind deep troughs when retreating (Bart, 2004; Hein et al., 2011; Morlighem et al., 2020).

The grounding line (or grounding zone, cf. Li et al., 2023), marks the transition between the grounded ice sheet and the floating ice shelves and thus constitutes the triple point of bedrock, ice, and ocean, see Fig. 1. Grounding lines in Antarctica can be found at depths down to $3000\,\mathrm{m}$ due to the erosion over long time scales. When considering the general features of sub-shelf melt patterns in each ice shelf, sub-shelf melt rates are generally higher near the grounding line and lower towards the ice shelf's calving front (Lambert et al., 2023), if mode 3-melting is absent (Silvano et al., 2016). This general pattern is

modulated by exchanges of water masses within the cavity and through other dynamical processes at play (e.g. the Coriolis effect). Ice-shelf thinning caused by melting close to the grounding line has been found to have the largest impact on the adjacent ice masses, resulting in higher ice fluxes across the grounding line due to a loss in buttressing (Reese et al., 2018b; Goldberg et al., 2019).

Distinct geological structures, such as troughs, are crucial boundary conditions for modelling ocean dynamics and the inter- action of the ocean with the Antarctic Ice Sheet (Thoma et al., 2008; Hellmer et al., 2012). However, previous studies do not systematically investigate the bathymetric access points or pathways to the grounding lines with regards to ice-sheet modelling and focus only on specific regions (see e.g. Herraiz-Borreguero et al., 2015; Tinto et al., 2019). For instance, Hellmer et al. (2012) and Naughten et al. (2021) simulate an inflow of warm water masses through Filchner Trough which subsequently

access large parts of the ice-shelf cavity of that region, leading to a drastic change in sub-shelf melt rates i.e. a switch from a 'cold' to a 'warm' cavity under high emission scenarios.

     Here, we present a simple approach to analyse *oceanic gateways* in all Antarctic regions, including Filchner Trough, to the base of the Antarctic Ice Sheet, specifically to the ice-sheet's grounding lines, see Fig. 1. Our study provides a sensitivity- experiment, where in case of a trough-like feature, we assume that off-shore ocean water is accessing the cavity (as in the

case of Filchner Trough), leading to a drastic change in sub-shelf hydrography. For this, we combine observations of bedrock topography and ocean water masses to assess present-day pre-conditions for enhanced melting in all Antarctic regions. While no dynamic changes in the ocean are taken into account, our analysis serves as a first-order assessment of an upper bound on melt rates that would be caused by an inflow of warm water masses at depth in Antarctica. Our approach of identifying relevant water masses that drive melting in cavities is also useful to improve the input for parameterisations of sub-shelf melt rates:

for the ice-shelf cavity model PICO (as used in this study, see methods; Reese et al., 2018a) for instance, ocean temperature and salinity input are averaged over a certain depth to be used as input in the box model. With our analysis, we aim at better estimating this depth, by i.e. re-aligning the ocean regions over which input is averaged horizontally to include the relevant depth levels and oceanic gateways.

     In this study, we aim to estimate the impact of potential future warm water inflow on basal melting for all Antarctic regions.

In order to do so, we (1) analyse the bathymetry and identify trough-like features that potentially provide access of off-shore warm waters into ice-shelf cavities, (2) calculate the increase in thermal forcing resulting from such a regime shift and (3) compute the respective increase in sub-shelf melting. First we describe our methodology in Sect. 2, which is followed by the presentation of our results in Sect. 3. In Sect. 4 we discuss our approach and findings, with a general conclusion included in Sect. 5.

## 2   Methodology

The goal of our approach is to quantify an upper bound to melting if cavities switch to a 'warm' mode. To this extent, we use PISM-PICO to compute ice-shelf basal melting for given ocean temperatures and salinities: As a present-day estimate, we take temperatures at the calving front. For the 'warm' mode, we use temperatures at mid-depth at the continental-shelf break (CSB).

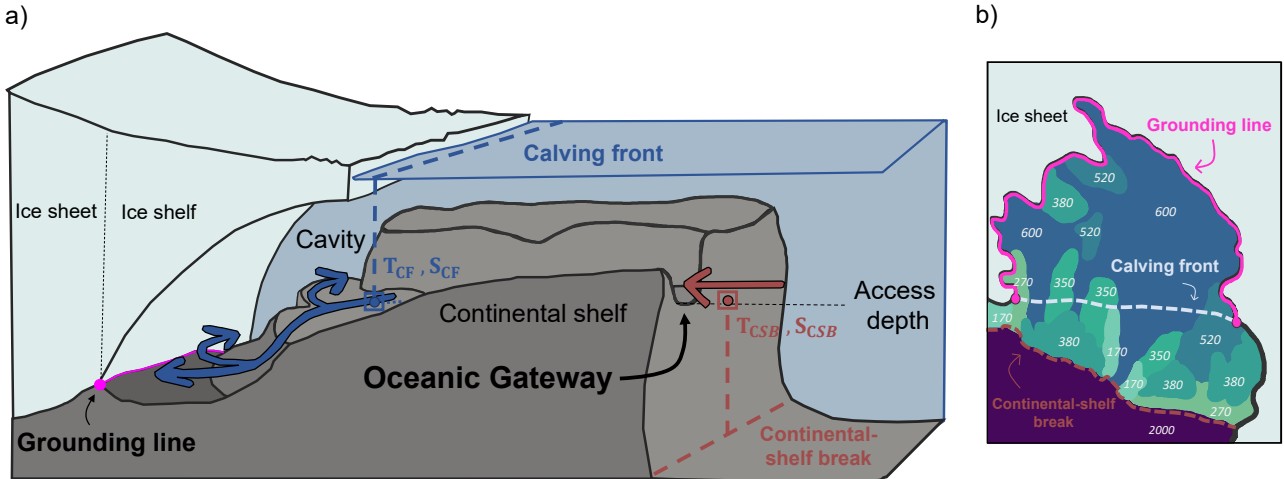

**Figure 1. Illustration of used concepts in this study. a)** Schematic of stylized oceanic gateway cross-cutting the continental shelf. Beyond the continental-shelf break, relatively warm Circumpolar Deep Water is present at mid depth. Its access to ice-shelf cavities is modulated by ocean circulation and bathymetry. Ocean temperatures at the bottom topography near the calving front ($T_{CF}$) provide information about the water masses that can already access the ice-shelf cavity (in the case of mode 1-melting, blue). If an oceanic gateway is present, water masses with a mean temperature $T_{CSB}$ from the continental-shelf break (at the gateway's *access depth*, red arrow) can potentially reach large parts of the grounding line (triple point of bedrock, ice and ocean; magenta line) of the respective ice shelf. **b)** Access depths for each part of the continental shelf is obtained via a connected-component analysis. This yields a 2D field showing at what depths the ocean floor inside the ice-shelf cavity is connected to the open ocean. Analysing the 2D field at the grounding line of the region (magenta line) provides an estimate of the potential impacts when warm water masses are redirected from the continental-shelf break to the grounding lines.

In order to constrain that latter depth and to estimate the potential impacts of this selection on the melting at the grounding line (accounting for bathymetric constraints), we define oceanic gateways based on access depths found in each region. In the following section, we define the concept of oceanic gateways and access depths (Sect. 2.1), then describe the used ocean data (Sect. 2.2) and summarise how we compute sub-shelf melting with PICO (Sect. 2.3).

## 2.1 Identifying oceanic gateways from bathymetry

Our analysis is based on BedMachine v3 bathymetry (Morlighem et al., 2020; Morlighem, 2022), which is provided on a $500 \times 500$ m grid spacing and contains ocean bathymetry from IBCSO v2 (Dorschel et al., 2022). From this, we calculate *access depths* for every location on the Antarctic continental shelf and in the ice-shelf cavities. The access depth, $d$, for each point on the continental shelf is the deepest vertical level (largest positive depth) for which there is a horizontal connection to the open ocean, not obstructed by bathymetry. We obtain these via a 'connected component analysis' (CCA). More specifically,

we use the connected-component approach implemented by Khrulev (2024), with an algorithm similar to He et al. (2010). The algorithm iterates through the vertical column from 0 m to 3500 m, spreads out in all horizontal directions, fills connected cells with the value of the depth at which they are connected, until it reaches boundaries or encounters obstacles i.e. cells with shallower bathymetry (the criterion is whether grid points are horizontally connected to the deep ocean at 3500 m or not). We have included Fig. S1 in the supplement to help visualising this analysis tool. As a result, our analysis yields circum-Antarctic access depths which are available as a 2D field on a 500 m×500 m horizontal grid spacing, following the resolution of the BedMachine data (Morlighem et al., 2020; Morlighem, 2022). When newer bathymetry fields become available, this data field can be easily updated with our processing scripts. We define the deepest access depth found along the grounding line of each basin as $d_{GL,0}$ and express the fraction of how much the grounding line at that depth is connected to the open ocean with values ranging from 0 % to 100 %. If a comparably large part of the grounding line is reached by only a small increase in vertical access level, an 'oceanic gateway' is present i.e. a deep trough connecting the (overdeepened) ice-shelf cavity to the open ocean past the continental-shelf break. We thus interpret an oceanic gateway to be the horizontal pathway from the open ocean to the grounding line of the ice sheet along the deepest possible ocean-connection between the two. For each region, we ascribe an oceanic gateway as 'major', if a global maximum (highest peak) in access depth along the grounding line is found at $d_{GL,0}$. On the data grid, we define the grounding line as the contour that delineates the contiguous grounded continental ice sheet (excluding larger islands and ice rises). We analyze oceanic gateways for 19 Antarctic regions based on the drainage basins defined in Zwally et al. (2012) and extended into the ocean, with the Filchner–Ronne and Ross basins congregated as in Reese et al. (2018a), see also Section 2.3.

## 2.2  Deriving changes in ocean forcing

Based on the bathymetric information obtained in the previous step, i.e. identifying the deepest topographic features that connect the deeper open ocean and the ice-shelf cavity (assuming that water follows this pathway), we derive the associated change in oceanic forcing, that results in a hypothetical switch from 'cold' to 'warm'- cavity conditions.

We analyse the properties of water masses based on the ISMIP6 ocean temperature and salinity climatology (Jourdain et al., 2020). The dataset is available at a 8 km×8 km horizontal and 60 m vertical resolution. The data points indicate temperatures and salinities averaged over the period 1995–2017. While observational datasets have many data gaps and thus do not provide sufficient horizontal as well as vertical coverage (especially on the continental shelf), the ISMIP6 approach fills these gaps with a specific extrapolation technique: while accounting for topographic barriers, the temperature and salinity fields from observations are extended, i.e., flooded into the ice-shelf cavities and regions below sea level that are currently covered by grounded ice. Due to this approach and the extended spatial coverage, we consider the ISMIP6 ocean dataset to be very well suited for our study. While the basic concept is the same, the ISMIP6 code is different to our analysis (Asay-Davis et al., 2020): our approach of quantifying the connectedness of the grounded ice to the open ocean aims at identifying pathways through which already existing warm water masses could fuel high melting rather than providing an extrapolated forcing field for projections. We therefore take into account the depth of grounding lines.

We extract ocean properties near the ice shelf's calving front, along the oceanic gateways as well as along the continental-shelf break, based on the local bathymetry and the access depths at the grounding lines for each basin $b = 1, 2, ...19$. The temperatures in front of the ice shelves (at the calving front) serve as a proxy for ocean water masses that can currently reach the ice shelves' deep grounding lines, similar to the case when mode 1-melting is dominant (cf. Silvano et al., 2016). The calving front (CF) is defined through the native BedMachine mask as the horizontal boundary between floating ice and the ocean. We calculate horizontal averages of temperature and salinity in the bottom layer, just above the bathymetry (topg), along the calving front and define $T_{\text{CF, mean}}$ and $S_{\text{CF, mean}}$ per basin as

$$T_{\text{CF, mean}}(b) = \text{mean}\left\{ T(x, y, z) \mid (x, y) \in \text{CF}(b) \text{ and } z = \text{topg}(x, y) \right\} \tag{1}$$

and

$$S_{\text{CF, mean}}(b) = \text{mean}\left\{ S(x, y, z) \mid (x, y) \in \text{CF}(b) \text{ and } z = \text{topg}(x, y) \right\}. \tag{2}$$

For estimating the change in melt rates, when assuming a basin-wide transition into a melt regime where melting becomes dominated by relatively warm CDW (mode 2 in Silvano et al., 2016), we derive properties along the CSB at the deepest grounding line access depth for each basin and compare it to the estimates from the calving front, our proxies for mode 1-melting. We define the CSB to lie in an around 40 km-wide perimeter along the horizontal coordinates where the bathymetry is at a depth of 1800 m (i.e. a band of five grid cells along the 1800 m isobath). We assume that once warm water is flowing onto the continent, it will eventually reach the grounding line, as CDW is not only warmer, but also saltier and therefore denser than on-shelf waters. We thus expect it to sink from the shallowest overflow point eventually towards the grounding lines, filling up the cavity basin and replacing the less dense waters at lower depths.

We define the average temperature and sanility along this transect as $T_{\text{CSB, mean}}$ and $S_{\text{CSB, mean}}$, respectively, as

$$T_{\text{CSB, mean}}(b) = \text{mean}\left\{ T(x, y, z) \mid (x, y) \in \text{CSB}(b) \text{ and } z = d_{\text{GL},0}(b) \right\} \tag{3}$$

and

$$S_{\text{CSB, mean}}(b) = \text{mean}\left\{ S(x, y, z) \mid (x, y) \in \text{CSB}(b) \text{ and } z = d_{\text{GL},0}(b) \right\}. \tag{4}$$

Following the approach, to estimate an upper bound to melt rate changes, we will also use the maximum temperature found along the continental-shelf break to estimate melt rates, which we call $T_{\text{CSB, max}}$ and define as

$$T_{\text{CSB, max}}(b) = \max\left\{ T(x, y, z) \mid (x, y) \in \text{CSB}(b) \text{ and } z = d_{\text{GL},0}(b) \right\}. \tag{5}$$

We will compare these estimates to the mean, but also the maximum temperatures found along the calving front $T_{\text{CF, max}}$,

$$T_{\text{CF, max}}(b) = \max\left\{ T(x, y, z) \mid (x, y) \in \text{CF}(b) \text{ and } z = \text{topg}(x, y) \right\}. \tag{6}$$

To find the highest potential of temperature change, we therefore arrive at three $\Delta T$-estimates:

$$\Delta T_{\text{mean-mean}}(b) = T_{\text{CSB, mean}}(b) - T_{\text{CF, mean}}(b), \tag{7}$$

$$\Delta T_{\text{max-mean}}(b) = T_{\text{CSB, max}}(b) - T_{\text{CF, mean}}(b), \tag{8}$$

and

$$\Delta T_{\text{max-max}}(b) = T_{\text{CSB, max}}(b) - T_{\text{CF, max}}(b). \tag{9}$$

The latter allows us to quantify the change in melting also in those regions, where melting is already driven by relatively warm water masses at depth i.e. where $T_{\text{CF}}$ is already warm. Using the minimum temperature found along the calving front, $T_{\text{CF, min}}(b)$, and comparing it with the maximum temperature along the continental-shelf break, $T_{\text{CSB, max}}(b)$, would yield the highest temperature difference which would follow our narrative of deriving an upper bound on sub-shelf melting. What is important to note here however, is that the CF and CSB values are defined differently: CF is averaged over the deepest depth levels along the calving front, i.e. the minimum value will most likely be derived from comparably shallow regions, where waters do not influence melting near the grounding line (Silvano et al., 2016). We therefore do not use $T_{\text{CF, min}}$ in our analysis.

## 2.3 Upper bounds of sub-shelf melting computed with the ice-shelf cavity model PICO

We compute the change in sub-shelf melt rates with the Potsdam Ice shelf cavity mOdel (PICO, Reese et al., 2018a). PICO extends the ocean box model by Olbers and Hellmer (2010) to be applicable in 3D-ice sheet models. It mimics the vertical overturning circulation present in ice-shelf cavities and can reproduce the wide range of average observed melt rates for 'warm' and 'cold' cavities. Ocean input is considered in PICO as an average per basin and once water masses reach the grounding line, they rise along the ice-shelf base towards the calving front, driven by the ice pump (Lewis and Perkin, 1986).

In Reese et al. (2023), PICO model parameters $C$ (in $\text{Sv}\,\text{m}^3\,\text{kg}^{-1}$) that describes the strength of the vertical overturning circulation and the heat-exchange coefficient $\gamma_T^*$, given in $10^{-5}$ $\text{m}\,\text{s}^{-1}$, are tuned to capture the sensitivity of melt rates to ocean temperature changes (cf. Reese et al., 2023). Input (T,S) to PICO in Reese et al. (2023) is based on temperature and salinity observations compiled by Schmidtko et al. (2014). In the tuning process, temperatures on the continental shelf were corrected for, similarly to the approach by Jourdain et al. (2020), such that the melt rates calculated by PICO match present-day observations compiled by Adusumilli et al. (2020a). We here calculate melting resulting from a sudden warming of the cavities to the temperatures from the continental-shelf break by applying the differences $\Delta T_{\text{mean-mean}}$, $\Delta T_{\text{max-mean}}$, and $\Delta T_{\text{max-max}}$ as anomalies to the temperature fields from Reese et al. (2023).

To capture the parameter uncertainty in our estimates, we use the 'best' and 'max' parameter combinations from Reese et al. (2023); $\left\{ C = 2.0\,\text{Sv}\,\text{m}^3\,\text{kg}^{-1}, \gamma_T^* = 5\times10^{-5}\,\text{m}\,\text{s}^{-1} \right\}$, and $\left\{ C = 3.0\,\text{Sv}\,\text{m}^3\,\text{kg}^{-1}, \gamma_T^* = 7\times10^{-5}\,\text{m}\,\text{s}^{-1} \right\}$ respectively. The maximum number of boxes (N = 5) are used as in Reese et al. (2023). We use the PICO implementation in the Parallel Ice Sheet Model (PISM; https://www.pism.io; Bueler and Brown, 2009; Winkelmann et al., 2011) and as initial conditions, ice thickness and bed topography from the BedMachine v3 dataset on a $4\,\text{km}\times4\,\text{km}$ grid spacing. We consider this resolution for estimating basal melt rates a good compromise between having a high resolution at the grounding line, on the one hand, and computational feasibility on the other hand.

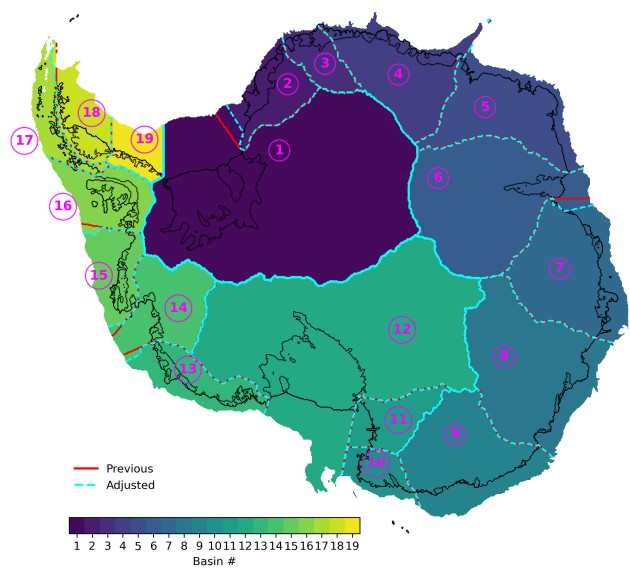

**Figure 2. PICO model basin boundaries.** The inland boundaries are based on satellite-derived drainage basins from Zwally et al. (2012) and were consolidated to 19 regions in Reese et al. (2018a). For the purpose of PICO, the basin boundaries were mostly extended along meridians into the ocean (red), which we have now partly adjusted (cyan) based on the derived access depths.

As our connected-component analysis yields a 2D field of access depths that identifies which parts of the continental shelf are topographically connected to the individual ice-shelf regions (i.e. on the same access depth), we use it to first correct for the existing basin boundaries by which the continental shelf is subdivided for in the PICO model. The boundaries on land are based on ice drainage basins from Zwally et al. (2012), were consolidated to 19 regions in Reese et al. (2018a) and for the use for PICO mainly extended along meridians into the ocean. In previous studies (Reese et al., 2018a, 2023; Sutter et al., 2023; Wirths et al., 2024), those basin boundaries in the ocean were used to extract a basin average for temperature and salinity (i.e. average over the region) to feed into the box model. Figure 2 shows the new basin boundaries that we will use throughout this study. We have changed the basin boundaries near Filchner–Ronne and Amery ice shelves, inside the Amundsen Sea region and near George VI Ice Shelf in the Bellingshausen Sea based on the region's access depths. For this, we have extended their region's boundaries (by overlaying the access-depth field with the bathymetry) to incorporate the detected pathways through which warm water masses could gain access to the ice-shelf cavities. We have also aligned the basin boundary at the North tip of the Antarctic Peninsula with the local bathymetry of the continental shelf. From here on forward, we use these updated basin boundaries (a first result of our analysis) and encourage other PICO users to do the same. We provide the new basin mask as NetCDF-file as well as the corresponding script to create those boundaries in our data repository.

## 3  Results

In the following, we include, firstly, the main results from our access depth analysis that helps identifying oceanic gateways around Antarctica (Section 3.1) and, secondly, discuss the 'warm'-mode melt estimates for those regions that we find feature such gateways to the ice sheet's grounding line (Section 3.2). We then compare the derived temperature changes and subsequent melt estimates for all Antarctic regions in Section 3.3.

### 3.1  Major oceanic gateways in 7 out of 19 PICO regions

Figure 3 shows the main differences between the computed access-depth field and the bathymetry taken from BedMachine v3 Antarctica (Morlighem et al., 2020). This comparison highlights which parts of the continental shelf (and ice-shelf cavities) are shielded by topographic barriers, potentially blocking the access of warm water masses from the open ocean at depth. The most pronounced differences are found underneath Amery Ice Shelf, where the differences between the two fields can be larger than 1000 m. At Amery, around 91% of the ice-shelf cavity is shielded by shallower bathymetry, i.e. the access depth is shallower than the topography in 91% of the cavity area. In contrast, this applies only to about a third of the cavity area for basins 7 or 17. We later see that this can be linked to the absence of any oceanic gateway structure in those regions.

The access depths evaluated at Antarctic grounding lines ($d_{\text{GL},0}$) are also included in Fig. 3. The deepest access depth, at which each of the 19 regions is unobstructedly connected to the open ocean, ranges from 283 to 610 m, a similar depth at which warm CDW resides off the continental shelf. We find the deepest ocean access at Cook and Mertz Ice Shelves (basin 9) at 610 m, followed by 595 m at Filchner–Ronne (basin 1) to the shallowest of 283 m at the Western Antarctic Peninsula (basin 17). The 2D field of access depths for all locations on the Antarctic continental shelf and its ice-shelf cavities is provided in Supplement Fig. S2.

The distributions of the 2D access-depth field evaluated at the respective grounding lines, $d_{\text{GL},0}$, are shown in Figure 4. We find an oceanic gateways-like access to the ice sheet's grounding line in 7 out of 19 PICO regions (cf. the first 'spike' in the distribution shown in Fig. 4): Filchner–Ronne (basin 1), Amery (basin 6), Ross (basin 12), the Amundsen Sea region (basin 14), Drygalski (basin 11), Larsen (basin 18) and Getz (basin 13). Our analysis shows that Filchner Trough in basin 1 is the most prominent, or 'major', oceanic gateway found in the 19 regions. It connects around 81% of its region's grounding line to the open ocean at an access depth of 595 m. At Amery (basin 6), we find a gateway connecting around 67% of the region's grounding line to the open ocean at a depth of $d_{\text{GL},0} = 526$ m. At Ross (basin 12), it is around 33% of the region's grounding line at a depth of 570 m and in the Amundsen Sea region it is around 31% at a depth of 575 m. Filchner–Ronne, Ross and Amery are the regions where not only the grounding lines, but, together with Fimbul (basin 3) and Totten (basin 8), also the cavities are most shielded by shallower bathymetry (Fig. 3). In Sect. 3.2, we give more context and details which troughs constitutes those oceanic gateways in the first four mentioned basins (Filchner-Ronne, Amery, Ross and the Amundsen Sea region). The access depths distributions shown in Figure 4 also highlight those regions, where more than one gateway is present in the region. We find two or more spikes (or local maxima) in the access depth distribution that can be matched to the access to individual ice-shelves comprised in this region: In the Getz region (basin 13), 10% of the grounding line of the

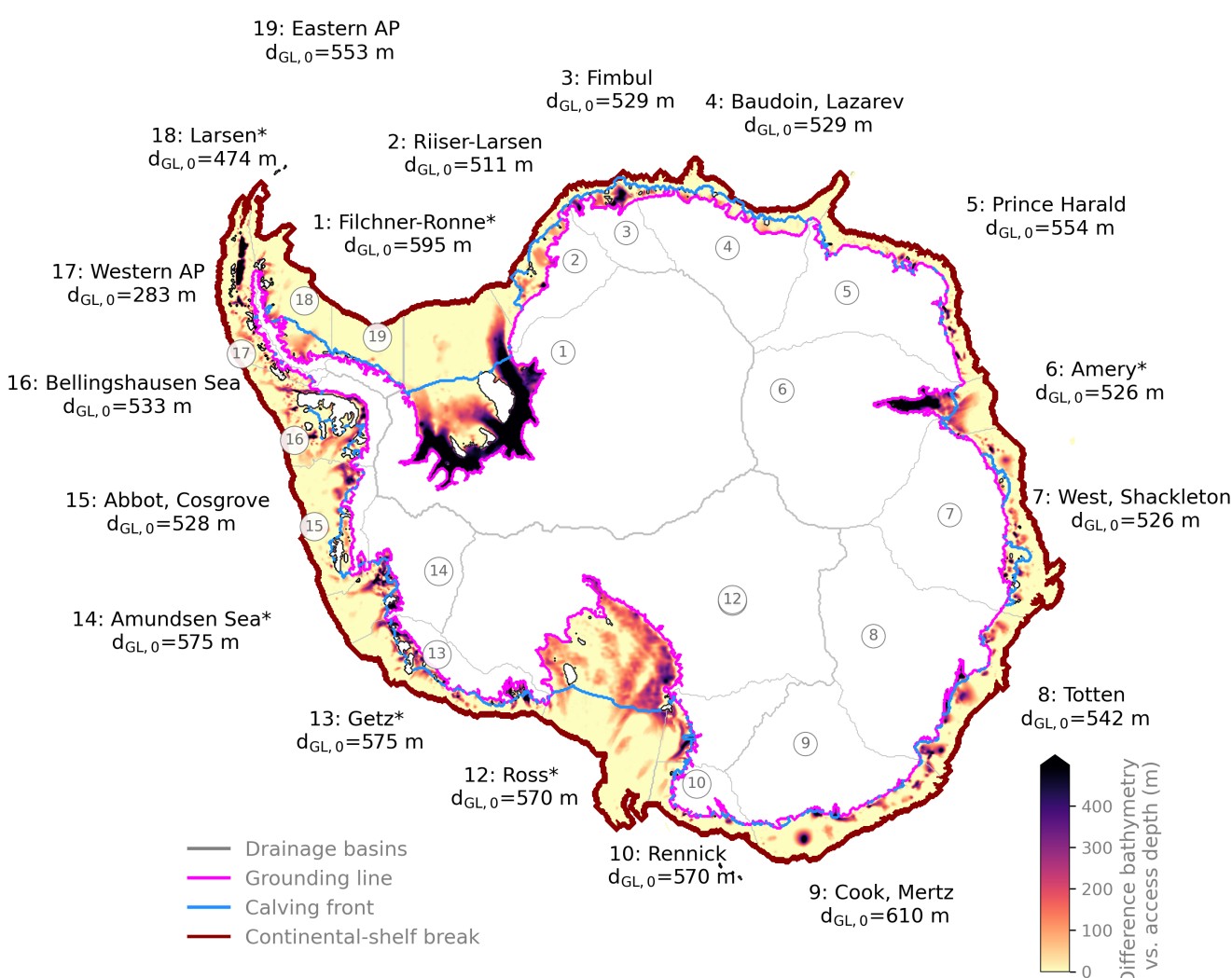

**Figure 3. Grounding line access depths and regions of the Antarctic continental shelf shielded by topographic features.** Color shading indicates the difference between the computed access depth over the continental shelf and in the ice-shelf cavities, compared to BedMachine v3 Antarctica bathymetry data (Morlighem et al., 2020). Evaluating the access depths found at each region's grounding line $d_{GL}$ reveal 'major' oceanic gateways in 7 out of 19 PICO regions, labelled with *. There, the deepest access depth found at the grounding line, $d_{GL,0}$ is most representative for the entire basin, as it represents the largest share of the grounding line. The drainage basins (grey outlines) are based on Zwally et al. (2012), consolidated as in Reese et al. (2018a), and labelled according to prominent ice shelves (with AP = Antarctic Peninsula). Coloured contour lines show the ice sheet's grounding line (magenta), the calving front (blue) and the continental-shelf break (red). For the 2D field of computed access depths see Supplement Fig. S2.

entire PICO region is accessed at Getz Ice Shelf at $d_{GL,0}$ =575 m. However, at an access depth of $d_{GL}$ = 489 m, the cavity of Sulzberger Ice Shelf (in basin 13; for location see e.g. Davison et al., 2023), which constitutes 8% of the total region's

grounding line, can be, in theory, horizontally accessed by open ocean water masses. We find multiple gateways also in PICO basin 3 that not only comprises Fimbul Ice Shelf ($d_{GL,0}$=529 m), but also Jelbart Ice Shelf (where 18% of the grounding line is accessed at $d_{GL}$ = 427 m) and Ekström Ice Shelf (19% is accessed at an access depth of $d_{GL}$ = 391 m). We see the same feature

in PICO basin 2 (comprising the Riiser-Larsen Ice Shelf, but also Brunt Stancomb Ice Shelf), basin 10 that holds Rennick and the smaller Lillie Ice Shelf, as well as in basin 8, in which the two spikes in the access depths distribution can be matched to the individual access to Totten and Moscow Ice Shelves, respectively. PICO basin 4 also shows a noteable difference in access to Roi Boudain Ice Shelf (12%) compared to Nivi Ice Shelf (11%). In PICO basins 5 (Prince Harald), 19 (Eastern AP), 9 (Cook, Mertz), 15 (Abbot, Cosgrove), 16 (Bellingshausen Sea), 7 (West, Shackleton) and 17 (Western AP), we do not find a significant

spike in the access depth distribution along the grounding line and hence conclude that no (seen from an Antarctic-wide scale) oceanic gateway is present. This is despite the fact that parts of the respective cavities are shielded by shallower bathymetry (in all regions at least a third of the cavity is shielded). Note that the existence of an oceanic gateway does not influence the melt rates calculated with PICO in Section 3.3, as we use $d_{GL,0}$ for all regions. However, it allows us to gauge the validity and limitations of our assumptions in each basin.

## 3.2 Potential sub-shelf melt changes in oceanic gateways regions

In the following, we further analyse our results for the Filchner–Ronne Ice Shelf, Amery Ice Shelf, Ross Ice Shelf and for the ice shelves in the Amundsen Sea for which we identified 'major' gateways. For the case in which a region comprises two or more ice-shelves, we exemplarily present results from the Totten region thereinafter.

### 3.2.1 Filchner–Ronne Ice Shelf

The Filchner–Ronne basin features an oceanic gateway at $d_{GL,0}$ = 595 m, through which 81% of the grounding line is horizontally connected to the open ocean. The identified oceanic gateway is Filchner Trough, which is a characteristic feature of the submarine topography in the Filchner–Ronne Ice Shelf region, see also Fig. 5. The trough extends from around Foundation Ice Stream to more than 450 km into the Southern Weddell Sea (distance measured from the ice-shelf front and taken from Larter et al., 2012). Its width varies between 125 to 175 km (Larter et al., 2012) and it terminates with a sill on its end towards

the Weddell Sea (Hellmer et al., 2012). The sill depth determines the region's access depth in our study. As the mean depth of the basin's grounding lines is around 1000 m, water flowing in at the access depth of 595 m could reach much of the region's grounding zone at once. The deepest grounding lines are found down to around 2000 m in the BedMachine dataset (see Fig. 5 b).

At present, Filchner–Ronne Ice Shelf has a relatively cold cavity, with observed melt rates of around 0.1 to $0.32\pm0.1\,\mathrm{m\,yr^{-1}}$

(Adusumilli et al., 2020a; Rignot et al., 2013). It currently contributes 10 % of the total ice-shelf mass loss around Antarctica (Mueller et al., 2018). In our analysis, water masses along the Filchner–Ronne calving front are close to the pressure melting point (with $T_{\mathrm{CF,\,mean}}$= -1.92°C at the ocean floor, cf. Fig. 5 c). A slope front in front of the ridge in Filchner Trough (Fig. 5 c) currently blocks warmer water masses that are present along the continental-shelf break (0.31°C in the mean or 0.53°C at maximum) from entering the cavity. If these were to enter the cavity, Filchner–Ronne would transition from a 'cold' to a

'warm' cavity, as also modelling studies suggest (Hellmer et al., 2012, 2017). At present, high-salinity shelf water (HSSW) is flowing into the ice-shelf cavity from the Ronne basin, while ice-shelf water (ISW) mainly flows outward through Filchner Trough (Nicholls et al., 2009; Naughten et al., 2021; Darelius et al., 2023). In our analysis, we also find colder HSSW residing in front of the sill of Filchner Trough on top of warmer water masses at depth, see Fig. 5 c.

The temperatures along the calving front do not have a wide spread, so that the derived temperature differences for a potential

'warm'-mode onset lie close together ($\Delta T_{\text{mean-mean}}$= 2.2°C, $\Delta T_{\text{max-max}}$=2.3°C and $\Delta T_{\text{max-mean}}$=2.5°C). When assuming that warm water masses from the continental-shelf break reach all the way into the cavity, we estimate basal mass fluxes to be two orders of magnitude higher and increases from 73.9–78.2 $\text{Gt yr}^{-1}$ (or 0.2 $\text{m yr}^{-1}$, see Reese et al., 2023), to 1466–2050 $\text{Gt yr}^{-1}$ (or 3.8–5.3 $\text{m yr}^{-1}$) using the difference in average temperatures, $\Delta T_{\text{mean-mean}}$, depending on the used PICO parameter combination. The heighted melt rates roughly correspond to the warm melt mode found at present at Getz Ice Shelf

(see Fig. S6; Reese et al., 2018a; Adusumilli et al., 2020a).

Using the difference in maximum temperatures along the the calving front and continental-shelf break, $\Delta T_{max-max}$, and the 'max' PICO parameter combination results in a basal mass flux of 2112 $\text{Gt yr}^{-1}$. The upper bound estimate using $\Delta T_{max-mean}$, i.e. the largest difference and the 'max' PICO parameter combination yields 2367 $\text{Gt yr}^{-1}$, which signifies a 30-fold increase (with melt rates as high as 6.1 $\text{m yr}^{-1}$).

Hellmer et al. (2012) found that a redirection of the slope current through Filchner Trough could occur within the 21st century under high greenhouse gas emissions and find a heightened basal mass flux of around 1600 $\text{Gt yr}^{-1}$ on average, which is on the lower end of our estimate. Naughten et al. (2021) find a two-timescale response of the Filchner–Ronne Ice Shelf under climate change, where warm water begins to intrude into the cavity only at approximately 7°C warming above pre-industrial levels. In an abrupt-4xCO2 scenario, due to the inflow of warm water masses, cavity temperatures are 2.7°C warmer, resulting

in melt rates that are 21× higher than the control, > 1400 $\text{Gt yr}^{-1}$. While our temperature differences of 2.2–2.5°C are slightly lower (we also do not consider a warming scenario here), we conclude that our obtained basal melt estimates are very close to the published literature. As for the drivers for such a regime shift, Haid et al. (2022) find that the density balance between the shelf waters originating from sea-ice production and the warmer water at the continental-shelf break is the most decisive factor for the Filchner–Ronne ice-shelf cavity to tip into a 'warm' state.

### 3.2.2 Amery Ice Shelf

Towards Amery Ice Shelf, we identify a gateway through Prydz Channel, see Fig. 6, along which the Amery grounding zone has retreated since the Last Glacial Maximum (Mackintosh et al., 2014). Similar to Filchner Trough, Prydz Channel is an example that shows that gateway-like features can often be linked to glacial erosion. The grounding line of Amery Ice Shelf lies very deep, at a mean depth of around 1100 m in the BedMachine dataset, while the deepest parts of the grounding line

are found at 2950 m depth (see Fig. 6 b). Once water flows onto the continental shelf at a depth of $d_{\text{GL},0}$ = 525 m, it could potentially reach large parts of the basin's grounding line (>60 % in our analysis).

The temperatures at the ocean floor near the calving front are -1.84°C on average. At the continental-shelf break, the mean temperature is 0.35°C, but temperatures are up to 0.60°C at maximum. When it comes to average melt rates, Rignot et al.

(2013) list observed melt rates at Amery Ice Shelf as $0.6\pm0.4$ m yr$^{-1}$ and Adusumilli et al. (2020a) as around 0.7 m yr$^{-1}$,
which is both similar to 0.6 m yr$^{-1}$ in Reese et al. (2023). In our study, melt rates would increase to 15.4–21.4 m yr$^{-1}$
(depending on the PICO parameter combination), when applying the temperature anomaly of 2.2°C ($\Delta T_{\text{mean-mean}}$). Our upper-
bound estimate for Amery Ice Shelf using $\Delta T_{\text{max-mean}}$= 2.4°C and the 'max' PICO parameter combination, yields basal melt
rates of 24.9 m yr$^{-1}$ (or 1339 Gt yr$^{-1}$), which would result in a 42-fold increase in melting.

Whether warm CDW residing at the continental-shelf break will actually pass through the identified gateway remains uncertain. Williams et al. (2016) find a different pathway of modified Circumpolar Deep Water towards Amery Ice Shelf, through
Four Ladies Bank more to the East, see Fig. 6 a, which is much shallower than Prydz Channel. Here our core assumption that
CDW always takes the deepest entry / gateway towards the ice shelf is challenged. Our quantitative estimates however fit to
a recent preprint, in which Jin et al. (2024), using a regional ocean model, show that melt rates could reach up to 17 m yr$^{-1}$
given a regime shift in the next century under a high-emission scenario. Amery Ice Shelf is located downstream of Lambert
glacier that is draining about 16 % of the grounded East Antarctic Ice Sheet (Fricker et al., 2000). Enhanced melting due to
warm water inflow at depth could hence produce an increase in sea-level contribution from large portions of the East Antarctic
Ice Sheet.

### 3.2.3 Ross Ice Shelf

For the majority of the Ross Ice Shelf cavity, we identify the Glomar Challenger Basin (see also Fig. 7) as the topographic feature which provides access at a depth of $\text{d}_{\text{GL},0}$ = 570 m. The basin is a north-east trending cross-shelf paleo-trough (Owolana,
2011). We determine its lower lying western sub-basin as an important gateway, that provides access to around 33 % of the
basin's grounding line. At $\text{d}_{\text{GL},0}$ = 570 m, the grounding lines of Mac Ayeal, Bindschadler and Mercer/Willans Ice Streams
(Western side) are reached as well as the grounding line of Byrd Glacier on the eastern side of Ross Ice Shelf. The mean depth
of the basin's grounding lines is rather shallow at around 575 m, but can reach around 1000 m in the BedMachine dataset (see
Fig. 7 b). These deep-lying grounding lines are accessed at $\text{d}_{\text{GL},0}$, but as visualized in Fig. 3 the cavity is less overdeepened as
a whole than those at Amery and Filchner–Ronne.

Similar to Filchner–Ronne Ice Shelf, cold water masses are found along the ice-shelf front (Fig. 7 c) with warmer water
masses beyond the continental slope front. The derived temperatures are in the mean at $T_{\text{CF, mean}}$=-1.9°C near the calving
front and $T_{\text{CSB, mean}}$=1.1°C in the mean at the continental-shelf break, with temperatures of up to $T_{\text{CSB, max}}$= 1.4°C. Observed
melt rates lie at $0.0\pm0.1$ m yr$^{-1}$ for the Western and $0.3\pm0.1$ m yr$^{-1}$ for the eastern part of the ice shelf (Rignot et al.,
2013). In Adusumilli et al. (2020a) they are around 0.1 m yr$^{-1}$. The Ross basin can hence be considered a 'cold' cavity,
like Filchner–Ronne Ice Shelf. Present-day melt rates from Reese et al. (2023) lie at 0.3 m yr$^{-1}$. In our analysis, melt rates
would increase to 5.7–7.8 m yr$^{-1}$ assuming a transition to mode 2-melting by around 3.0°C warmer water entering the cavity
($\Delta T_{\text{mean-mean}}$=3.0°C). Using $\Delta T_{\text{max-mean}}$=3.3°C, i.e. the largest temperature difference we consider for this basin, the heightened
bass mass flux corresponds to a roughly 29-fold increase in basal mass flux from around 132 Gt yr$^{-1}$ to 3815 Gt yr$^{-1}$.

Tinto et al. (2019) find that high-salinity shelf water flows under the ice front near Ross Island to the East, then moves
southward towards the East Antarctic side of the ice shelf, and eventually exits through Glomar Challenger Trough to the

Ross Sea. They highlight that the tectonic boundary between the East and West Antarctic side of Ross Ice Shelf impact the vulnerability to sub-shelf melting, since the part of the cavity near Siple Coast is rather isolated from the influence of in-flowing (warm) water masses. Here we assume, however, an inflow of warm water masses through Glomar Challenger Basin reaching those ice streams, given an access of water masses at $d_{GL,0} = 570$ m. The rest of the cavity near Siple Coast shows generally more shallow access depths in our analysis, which can be linked to the tectonic boundary and the difference in the crustal composition that influence the bathymetry (Tinto et al., 2019).

### 3.2.4 Ice shelves in the Amundsen Sea

At present, ice shelves in the Amundsen Sea have 'warm' cavities and therefore dominate the current mass loss in Antarctica (see e.g. Pritchard et al., 2012), indicating that this region is already out of balance with the current oceanic forcing. Here, comparably warm water masses have already found their way underneath the ice shelves, in contrast to the three ice-shelf regions detailed before.

When examining the bathymetry in the region, we find the most direct i.e. horizontally closest connection from the grounding lines to the open ocean to be Abbot Cosgrove Trough, an around 760 m deep feature that evolved through erosion along a paleo ice stream across the continental shelf (Hochmuth and Gohl, 2013; Klages et al., 2015). Abbot Cosgrove Trough feeds into Pine-Island Thwaites Trough, close to the 'Eastern Trough' as often referred to in the literature (see e.g. Dutrieux et al., 2014). However, the deepest access depth found at the grounding line in this basin is dominated by the access through Dotson-Getz Trough (at 575 m), that enables a potential pathway for water masses from Getz Ice Shelf (accessed via the Western Getz / Siple Trough from the open ocean, see Fig. 8) through Dotson and Crosson Ice Shelves eventually reaching Thwaites and Pine Island in our analysis. The mean depth of this basin's grounding line is at around 680 m, but the deepest parts lie at $> 1500$ m, that are reached by the access depth, $d_{GL,0}=575$ m. The ocean dataset we used shows warm water masses along the entire transect from the deep ocean up to the ice-shelf front (Fig. 8 c).

Following the analysis steps similar to the other regions, we derive a mean temperature of $T_{CF, mean}$=-0.23°C near the calving fronts of this region. This is considerably cooler than the near-bottom temperature presented in Dutrieux et al. (2014), namely 1.2°C in 2012 at the Pine Island Glacier calving front. The latter average is much closer to our workflow-generated $T_{CSB}$ estimates for that region, $T_{CSB, mean} = 1.42$ °C and $T_{CSB, max} = 1.54$ °C. In Fig. 10, we show the large spread in temperatures along the calving front in this basin. The highest temperatures along the calving front are found near Pine Island Glacier with a maximum temperature of $T_{CF, max}$=1.17 °C. In contrast, the mean temperature of -0.23°C is influenced by much colder bottom temperatures found at Crosson Ice Shelf (-1.2°C; cf. Supplement Fig. S7). Considering that warm waters are already present within most of the Amundsen Sea's ice-shelf cavities, $T_{CF, mean}$ can thus be considered an unrepresentative metric for deriving bathymetric constrained 'warm'-mode melt estimates in this region. For this region, we thus use the difference of $T_{CF, max}$ to $T_{CSB, max}$, $\Delta T_{max-max} = 0.4$°C, to derive an upper bound melt estimate i.e. assuming an inflow of more, unmodified, CDW to all grounding line parts of the region. See Fig. 10 for the other temperature differences, $\Delta T_{mean-mean}$ and $\Delta T_{max-mean}$ to compare.

Pine Island Glacier with observed melt rates of $16.2\pm1.0$ m yr$^{-1}$ and Thwaites glacier with $17.73\pm1.90$ m yr$^{-1}$ (Rignot et al., 2013), respectively, have been considered belonging to one basin in our analysis. In Adusumilli et al. (2020a), the

melt rates are stated to be around $9.1 \, \mathrm{m \, yr^{-1}}$ for that basin, while melt rates of $14.4$–$14.5 \, \mathrm{m \, yr^{-1}}$ result from the parameter tuning in Reese et al. (2023) Using $\Delta T_{\text{max-max}}$, sub-shelf melting in the Amundsen Sea could increase to $19.9$–$21.7 \mathrm{m \, yr^{-1}}$ or up to $314 \, \mathrm{Gt \, yr^{-1}}$ (1.5–fold increase from around $210 \, 314 \, \mathrm{Gt \, yr^{-1}}$ at present). In our study, this change is the second lowest; neighbouring Getz Ice Shelf region (basin 13) experiences almost no change in melting ($\Delta T_{\text{max-max}}=0.001°\text{C}$). Since the cavities at Amundsen are already in a 'warm' state, it is unsurprising that our analysis shows only little increase in melting.

How do these findings fit to the published literature? Thoma et al. (2008) simulate an inflow of CDW onto the Amundsen shelf and find that the warm water reaching Pine Island Bay are guided trough a submarine trough from the continental-shelf break, close to where we estimate the continental-shelf break temperatures $T_{\text{CSB}}$ in our study. Haigh et al. (2023) find that the ridge that is indicated in our study as the overflow point (see Fig. 8 c), blocks inflow from the Bellingshausen Sea at depth, so that water masses rather originate from the Pine Island Thwaites Trough, similar to Thoma et al. (2008) and Naughten et al. (2022).

### 3.2.5 The case of multiple gateways as found in the Totten region

For PICO, the Totten region incorporates Totten and Moscow University Ice Shelves. Totten Ice Shelf has a direct ocean access at a depth of $496 \, \mathrm{m}$ through a trough near the Law Dome peninsula, that constitutes around $23 \, \%$ of the total grounding line length, see Fig. 9. Our $d_{\text{GL},0}$ estimate for the entire basin is $542 \, \mathrm{m}$ however, meaning that there are some deeper grounding line parts that have a deeper horizontal connection to the ocean, see the Totten sub-panel in Fig. 4. These deeper parts constitute less than $1 \, \%$ ($0.4 \, \%$) of the region's grounding line. The mean depth of the basin's grounding line is $635 \, \mathrm{m}$, but the deepest parts go down to around $2100 \, \mathrm{m}$. Moscow University Ice Shelf has a slightly shallower access depth of $384 \, \mathrm{m}$ (see the second spike in Fig. 4), compared to Totten Ice Shelf.

In the ISMIP6 climatology, warm temperatures are not only present along the continental-shelf break but can also be found on the continental shelf in front of Totten Glacier (cf. 190–250 km along the transect in Fig. 9, c). Water masses near the calving front have a mean temperature of $-1.4°\text{C}$, but $0.01°\text{C}$ at maximum. At the continental-shelf break, water masses are $0.67°\text{C}$ warm, with a maximum of $1.2°\text{C}$. The mean temperatures thus differ by $2.1°\text{C}$ ($\Delta T_{\text{mean-mean}}$). Rignot et al. (2013) find melt rates at Totten Ice Shelf to be $10.47 \pm 0.7 \, \mathrm{m \, yr^{-1}}$ (around $7.8 \, \mathrm{m \, yr^{-1}}$ in Adusumilli et al., 2020a). According to our analysis when applying $\Delta T_{\text{mean-mean}}$, melt rates at Totten would see an increase in basal mass flux from around $90 \, \mathrm{Gt \, yr^{-1}}$ in Reese et al. (2023) to $501$–$664 \, \mathrm{Gt \, yr^{-1}}$ (around $7$ to $40$–$53 \, \mathrm{m \, yr^{-1}}$), depending on the PICO parameter combination. When assuming an inflow from the warmest waters near the continental-shelf break at $496 \, \mathrm{m}$, using $\Delta T_{\text{max-mean}}$, the basal mass flux would increase up to 9-fold (to $826 \, \mathrm{Gt \, yr^{-1}}$). Those water masses are $2.6°\text{C}$ warmer compared to those present at the ocean floor near the calving front.

Totten Ice Shelf is the floating extension of Totten Glacier, that drains a catchment containing ice with an equivalent of $3.5 \, \mathrm{m}$ of global sea-level potential (Greenbaum et al., 2015), and currently experiences the largest thinning rate of all East Antarctic regions (Pritchard et al., 2009; Flament and Rémy, 2012; Greenbaum et al., 2015). Here, elevated sub-shelf melt rates due to warm water inflow onto the continental shelf could already be the cause for the adjacent glacier to thin. Further ocean-induced melting can therefore have significant consequences to global sea-level rise. Moscow University Ice Shelf is

included in the same region as Totten Ice Shelf. From our analysis, we determine the relevant access depth, $d_{\mathrm{GL},0}$ to be 373 m, which resembles the second spike in the distribution when evaluating the access depths of the entire region's grounding line(s), see Fig. 4. For more specific regional results for 'warm'-mode melt estimates, the two ice shelves need to be treated as separate basins, which we leave for future research.

### 3.3 Change in melt rates assuming transitions towards 'warm'-mode melting in all Antarctic regions

After having considered specific aspects of our analysis in individual ice shelf basins we now derive generalized insights from our analysis, evaluating all Antarctic regions: Average temperatures along the ice-shelf fronts, $T_{\mathrm{CF,\ mean}}$, which are derived at the ocean floor in the individual basins, are lower than temperatures found at the relevant access depth at the continental-shelf break, $T_{\mathrm{CSB,\ mean}}$, see Figure 10. This is not surprising as $T_{\mathrm{CSB}}$ incorporates the warm CDW which resides at mid-depth off the Antarctic continent, while $T_{\mathrm{CF}}$ often reflects the cold outflow of ice-shelf melt water at depth. Figure 10a shows the distribution of temperatures along these two locations for each basin. Mean $T_{\mathrm{CF}}$ estimates range from -1.92°C at Filchner–Ronne, to 0.19°C in basin 16 (Bellingshausen Sea). Especially in West Antarctica, the spread in $T_{\mathrm{CF}}$ is very large due to warm water masses being present in some troughs along the CF, compared to, for instance, the large ice-shelf regions of Filchner–Ronne, Ross or Amery, where those water masses are not found. Mean temperatures along the continental-shelf break, $T_{\mathrm{CSB,\ mean}}$, range from $-0.29$°C in basin 2 (which incorporates the Riiser-Larsen Ice Shelf) to 1.74°C found in the Bellingshausen Sea region (basin 16). The maximum temperatures near the continental-shelf break, $T_{\mathrm{CSB,max}}$ are highest in West Antarctica with the Bellingshausen Sea region reaching 1.86°C at maximum.

The temperature differences, shown in Figure 10b, range from 1.0 to 3.0°C when comparing mean estimates off-shore and along the calving front ($\Delta T_{\mathrm{mean\text{-}mean}}$). We find the largest difference in basin 12 that incorporates Ross Ice Shelf ($\Delta T_{\mathrm{mean\text{-}mean}} = 3.0$°C), in basin 7 ($\Delta T_{\mathrm{mean\text{-}mean}} = 2.4$°C) and in basins 1 (Filchner–Ronne Ice Shelf), 6 (Amery), 10 (Rennick), and 13 (Getz) with $\Delta T_{\mathrm{max\text{-}max}} = 2.2$°C. Comparably warm temperatures at the calving front ($T_{\mathrm{CF,\ max}}>0$°C) are found in basins 2, 4, 5 and basins 8 to 10 in East Antarctica and in basins 13–18, i.e. in all West Antarctic basins, so that the difference to the continental-shelf break temperature is rather small. Especially in West Antarctica, high $T_{\mathrm{CF,\ max}}$ can be related to warm water already being present in some troughs along the calving fronts. When accounting for this i.e. using $\Delta T_{\mathrm{max\text{-}max}}$, the largest temperatures differences can still be found in the large, at present 'cold'-mode, ice shelves Filchner–Ronne (2.3°C) and Ross (3.0°C). In West Antarctica, the difference range from almost zero (at Getz) to 0.4 (Amundsen Sea) and 0.6°C (Bellingshausen Sea). Compared to that, assuming an inflow of the minimum temperatures found along the CSB, $T_{\mathrm{CSB,\ min}}$, yields higher temperature differences (0.8–2.3°C) when excluding basin 2 and 17, where the difference is negative ($T_{\mathrm{CSB,\ min}} < T_{\mathrm{CF,\ mean}}$). Since we want to provide an upper-bound estimate for bathymetry-constrained 'warm'-mode melt rates around Antarctica, we employ the anomalies of $\Delta T_{\mathrm{max\text{-}mean}}$ (i.e. taking the highest continental-shelf break temperature and compare it to the basin-mean along the calving front). $\Delta T_{\mathrm{max\text{-}mean}}$ range from 1.6 (basin 2 and 18) to up to 3.3°C (at Ross). It is interesting to note here, that for the big ice-shelf regions Filchner–Ronne, Ross and Amery, the way we obtain the temperature anomaly in case of basin-wide transition to 'warm'-mode melting does not matter much: $\Delta T_{\mathrm{mean\text{-}mean}}$, $\Delta T_{\mathrm{max\text{-}max}}$ and $\Delta T_{\mathrm{max\text{-}mean}}$ are very similar in these regions, which is due to the narrow temperature distribution along the present-day calving front (Fig. 10a). Since those

three regions feature 'major' oceanic gateways, our approach of assuming an inflow from around $\Delta T_{\text{max-mean}}$=3°C warm water masses to the respective grounding lines is most valid there. As the Drygalski region (basin 11) shares the continental-shelf break with the Ross region, we do not provide an estimate for $T_{\text{CSB}}$ here. For this region, subsequent melt rates are not estimated either. Temperatures relative to the in situ freezing point, i.e. the thermal driving, is provided by in Supplement Fig. S3 and the actual PICO forcing temperatures are attached in Supplement Fig. S4. In all basins, the water masses from the continental-shelf break are saltier than compared to those near the calving front ($S_{\text{CSB}} > S_{\text{CF}}$). The difference in the extracted salinity inputs is however small, ranging between nearly 0 PSU at Filchner–Ronne (basin 1) to 0.6 PSU at the Bellingshausen Sea region (basin 16). All salinity estimates are shown in Supplement Fig. S5.

Melt rates computed with PICO for the anomalous ocean temperatures and salinities are displayed in Figure 11. Almost all regions show a strong increase in sub-shelf melting when assuming that warm waters from the continental-shelf break can reach the ice-shelf cavities, all the way to the grounding line. Relative to their present-day estimates, melt rates increase most in the big ice-shelf regions of Amery, Filchner–Ronne and Ross that show a >20-fold increase in melting, see Fig. 11c, when assuming that warm waters from the continental-shelf break can access the respective ice-shelf cavities. We find the largest increase at Amery Ice Shelf, where melt rates could increase up to 42-fold, cf. Fig. 11c. With our access depth analysis in these three regions, we have found that, at the access depth used for extracting the temperatures at the CSB, $d_{\text{GL},0}$, more than 30% of the respective grounding lines are accessed. This is why we classified these regions to have 'major' oceanic gateways see Fig. 4. This gives our results significance, as in those regions, our assumptions with PICO are particularly valid e.g. warm water from the CSB is channelled to the respective grounding lines.

Ice-shelf regions in West Antarctica do feature oceanic gateways as well (e.g. Getz with 9.8% of its grounding line reached at $d_{\text{GL},0}$) but since there is already warm water present at the CF, there is little potential for as drastic changes in the sub-shelf melt rates through bathymetric constrained inflow as in 'cold' cavities at present. Generally, our analysis highlights the generally strong sensitivity of the large, 'cold' cavities (Filchner–Ronne, Ross, Amery), which stands in contrast to regions with already warm cavities (Getz – Western AP). These 'cold' cavity-regions show 'major' oceanic gateways (Fig. 4) that allow access into a well-shielded cavity (especially true in the case of Filchner–Ronne and Amery, cf. Fig. 3). The Filchner–Ronne, Ross and Amery regions hold together more than 30 m sea-level equivalent ice volume, with dire consequences in case of a switch to 'warm'-mode in any of these regions. Please note, that in all cases we assume that the increase in melting is mainly driven by the changes in temperature: the melting effect of the salinity differences of a maximum of 0.6 PSU, are by around one order of magnitude smaller.

## 4 Discussion

Our data analysis infers potential pathways for warm water inflow into ice-shelf cavities from access depths for 19 drainage basins in Antarctica and provides estimates for induced changes in sub-shelf melt rates. The results of the analysis need to be evaluated in light of the key assumptions and limitations of our approach: firstly, we assume that ocean waters in front of the ice shelf serve as valid proxy for water masses that currently drive melting underneath the ice shelf, which is generally

valid for 'cold'-mode ice shelves, but not for shelves with 'warm'-mode melting (Silvano et al., 2016). Not all ice shelves are considered 'cold'-mode ice shelves at present, most notably the ice shelves in the Amundsen Sea region. We partially considered this special case in our analysis by providing the $\Delta T_{\text{max-max}}$ estimates in our analysis. Second, we estimate the continental-shelf break temperatures at the region's deepest grounding line access depth, $d_{\text{GL},0}$, assuming that water masses simply follow the bathymetry when flowing onto the shelf, and not follow isopycnals (Drijfhout et al., 2013). Ocean dynamics, which crucially determine sub-shelf circulation patterns and thereby influence the access potential (Nicholls et al., 2009; Williams et al., 2016), are not considered in this study. Our analysis is thus a sole representation of the role of the geometry of the continental shelf including the ice-shelf cavities and connecting features such as the oceanic gateways. Our study could therefore be improved by considering specific ocean circulation patterns informed by high-resolution ocean models, such as in Naughten et al. (2023), that can also assess the boundary conditions for mode 2 onset in all regions. Please note that our results are not directly dependent on the grounding line coverage at the deepest access depth, but it enables us to contextualize the results. Our temperature and melt rate changes would not differ if at $d_{\text{GL},0}$ only 1% instead of a higher percentage were horizontally connected to the open ocean. PICO uses one temperature (and salinity) estimate per basin to compute sub-shelf melt rates. However, the existence of a 'major' oceanic gateway means that a substantial portion of the grounding line is reached at $d_{\text{GL},0}$ and the PICO input values are a good representation of potential results. In the case of the Filchner–Ronne basin for instance, $d_{\text{GL},0}$ reaches more than three quarters of the cavity and is thus, in our conclusion, adequately representative for the entire shelf e.g. for estimating a bathymetric-constrained 'warm'-mode melt estimate. In the other case, that no major oceanic gateway exists, PICO input values represent an upper bound on the oceanic properties that would reach the grounding line. Cold and dense shelf waters flowing out of ice-shelf cavities generally shield the ice shelf from warm CDW inflow at depth (Janout et al., 2021). The circulation patterns in the ice-shelf cavity system such as Filchner–Ronne, are strongly controlled by dynamical processes, for instance by the Coriolis force or, for instance, the interplay of sea-ice production and polynya formation which is in turn linked to anomalies in the large-scale atmospheric circulation around Antarctica (Alley et al., 2015; Janout et al., 2021; Haid et al., 2022). However, our identified gateways could be an entry point to cross-cut the density barrier (i.e. the Antarctic Slope Current) in front of the continental shelf (Hirano et al., 2023). Furthermore, changes in the thermocline depth and resulting changes in density could lift up water masses over topographic features (Assmann et al., 2013; Dutrieux et al., 2014; Hattermann, 2018; Daae et al., 2020). Here again, high-resolution ocean dynamical models could suggest that access is more likely through shallower channels, or that even deeper ocean levels than at access depth should be considered.

Typically, if CDW flows onto the continental shelf, it mixes with fresh and colder on-shelf water masses (Wang et al., 2023). This modified Circumpolar Deep Water (mCDW) is generally colder than the temperatures estimated in this study: Williams et al. (2016) define the maximum potential temperature of mCDW to lie between -1.7 and 0°C, while Ribeiro et al. (2021) use a range from -1.7 to 1.5°C for mCDW, when classifying water masses near Totten Ice Shelf. Since we neglect the modification of Circumpolar Deep Water when accessing the grounding lines in the ice-shelf cavities, our findings should be understood as upper-bound estimates.

Cavity-resolving ocean models are computational very expensive and therefore limited to simulations on centennial timescales. Millennial timescale studies or large ensembles of simulations thus often rely on parameterisations to infer ocean-driven sub-

shelf melting. We here use the PICO model to estimate sub-shelf melt rates based on the temperatures and salinities in front of the ice shelves as well as from the continental shelf-break. Favier et al. (2019) find that a box parameterisation that mimics the vertical overturning in the cavity, such as PICO, provides melt estimates that are comparable to coupled ice-ocean simulations. However, our melt rate estimates could differ when using an alternative melt parameterisation or assuming a higher melt rate sensitivity to thermal forcing, e.g. by using a quadratic melt relationship (Burgard et al., 2022). It should be noted that Burgard et al. (2022) does not find good agreement between PICO and a reference coupled model, but the PICO implementation in that study also uses a completely different PICO parameter tuning. In our study, we assume that once waters can reach the grounding line it can access all parts, as one temperature and salinity estimate is applied to the whole length of the grounding line in the box model. With a spatially more explicit approach, with which one could provide temperature (and melt) locally to each grid cell, one could apply the extracted temperature offset to only those parts of the grounding lines that are connected to the open ocean at the deepest access depth found at the region's grounding line.

PICO does neither include horizontal ocean circulation, modification of water masses on the continental shelf or blocking of water masses entering the continental shelf nor mode 3-melting (where surface waters cause melting near the ice-shelf front). This might bias melt rates in 'cold' cavities at the moment. Furthermore, the melt pattern in PICO is spatially less variable than in ocean circulation models or observations, which means that PICO does not reach the very high melting at the order of 100 $\mathrm{myr}^{-1}$ reported close to grounding lines (Dutrieux et al., 2014; Paolo et al., 2015). The relevance of this for ice sheet model studies needs to be further assessed (some first analyses were done in Reese et al., 2018b; Berends et al., 2023). A recent study suggested that bulk melting is more relevant than spatial patterns for the small, constrained Pine Island Glacier Ice Shelf (Joughin et al., 2021). The question if bulk melting or the melt pattern is more relevant, is not resolved yet, but our study does not aim to estimate this and we would hence refer to future work.

Furthermore, we use only two parameter combinations for the overturning and heat exchange coefficients in PISM-PICO, to capture the parameter uncertainty in the melt estimates when assuming a warm water inflow from the continental-shelf break. We use those parameter sets that were selected to match the sensitivity of melt rates to temperature changes for present-day Antarctica (Reese et al., 2023). However, a full model ensemble would be required to estimate the full uncertainty that arises from the choice of the PICO parameters. Despite these limitations, we want to stress that melt parameterisations such as PICO are essential for large-ensemble studies or long-term studies that cavity-resolving ocean circulation models cannot cover due to computational costs. They will thus serve an important purpose also in future ice-sheet model simulations and projections.

We further assume that the bathymetry is time-invariant, which is not the case when considering longer time scales. Sill depths and grounding line location and thus access depths may change by hundreds of meters in response to erosion, sea-level changes and glacial isostatic adjustment effects (see Kreuzer et al., 2025a, accepted).

As we have shown, the analysis of access depths on the continental shelf helps to better inform the basin boundaries in PICO, that could be applied to different melt parameterisation in ice-sheet models as well. However, there are a number of alternative subdivisions of the Antarctic continent, as for example in van der Linden et al. (2023), following Levermann et al. (2020), in which they differentiate between the Ross, Amundsen, Weddell, Peninsula, and an East Antarctic Ice Sheet ocean sector. In the Ross Sea however, they separate between the ocean in front of Victoria land (Drygalski region) and the rest of the Ross

Sea. This makes their classification not suitable for our analysis, as we consider the continental-shelf break in front of Ross Sea representative for both regions.

Using the 19 PICO basins in our study, we conclude for basins 1, 6, 12, 14, 11, 18 and 13 (7 out of 19 regions; cf. Fig.4), that $d_{\mathrm{GL},0}$ is representative for the entire basin in the case of an oceanic-gateway driven switch to 'warm'-mode melt conditions, as $d_{\mathrm{GL},0}$ represents the largest grounding line share. In other regions (e.g. in basins 3, 2, 10), however, $d_{\mathrm{GL},0}$ and subsequent temperature offsets and melt rates based on that estimate are less representative for the entire basin but constitute even more an upper-bound estimate as $d_{\mathrm{GL},0}$ is lower, hence represents warmer CDW. This could be fixed by simulating each individual ice

shelf separately. Finer resolutions i.e. on the individual ice shelf level would reveal more individual gateways but this analysis is out of scope of this manuscript.

When it comes to the effects of the potential warm water inflow, as analysed in our study, the difference in temperatures is small in some regions for physical reasons: this can be the case if the access depth of the basin is shallow and encompasses slightly colder water masses at the CSB, i.e. representing surface waters not CDW, or if the calving front temperatures are

already relatively warm, as in the case of the Amundsen region. In those regions, changes in melting may be more sensitive to gradual offshore changes in continental-shelf break temperatures instead of a qualitative circulation change, i.e. a regime shift of cavity inflow leading to a switch from a 'cold' to a 'warm' cavity, which our method is designed to assess. When considering estimates on CDW-inflow driven sub-shelf melting, one has to consider, however, that ocean temperatures are projected to become warmer in the future, for instance, by 1.2°C as found by Gómez-Valdivia et al. (2023) that employ a

global climate model on a relatively coarse resolution (1° ocean model).

The temporal evolution of warm water accessing the Antarctic grounding lines at depth depends on the complex interplay of ice, ocean, atmosphere, and solid Earth. Importantly, the timing would mainly depend on the future climate change scenario determining the change in oceanic boundary conditions. We here aim at quantifying the potential effect this might have in the future. Ocean model projections show that warm water access under the Filchner–Ronne Ice Shelf may occur due to ongoing

climate change, but that it is unlikely to happen within the next decades (Hellmer et al., 2012; Naughten et al., 2021; Haid et al., 2022). Other regions might also be susceptible to a basin-scale transition to mode 2-melting: When assuming that sub-shelf melting becomes intensified by warm water from the continental-shelf break, Jordan et al. (2023) find that the East Antarctic Ice Sheet might lose up to 48 mm of sea-level equivalent ice volume over the next 200 years. However, they artificially alter the ocean forcing to represent a shift to stronger on-shelf CDW transport.

All in all, cavity geometries are highly heterogeneous and the impact of the onset of mode 2-melting should thus be determined individually in a follow-up study taking into account other measures for the response of the grounding line, e.g. buttressing, as in Naughten et al. (2023). Our analysis follows only an idealized approach; for realistic projections of potential future regime shifts in the Antarctic ice-shelf regions, more sophisticated approaches are needed. These approaches at best have a coupled ice-ocean-atmosphere representation, with interactive ice sheets and ice shelves at high resolution in space and

time.

## 5    Conclusion

In our study, we present a simple approach to calculate the access depths of water masses to Antarctic grounding lines. We combine available bathymetry data with present-day ocean temperature and salinity data. Thereby, we identify 'major' oceanic gateways in 7 out of 19 regions through which warm water masses residing off the continental-shelf break could potentially access large parts of the deep grounding lines in several Antarctic regions. Warm-water inflow to regions with deep-lying grounding lines and subsequent increased sub-shelf melting can have a strong impact on the ice flux across the grounding line and therefore the overall mass balance of the Antarctic Ice Sheet (Reese et al., 2018b; Goldberg et al., 2019).

Perturbing the current state of the Antarctic Ice Sheet with warmer temperatures at the continental-shelf break helps estimating an upper bound on melt rate changes. All regions would experience a strong increase in sub-shelf melting, while basal melt rates would increase up to 42-fold in cavities that are currently in a 'cold' state, are well-shielded by shallower bathymetry and have a 'major' oceanic gateway that could channel warmer water masses to the grounding lines. We estimate an increase in temperatures at a maximum of 3.3°C. As our quantitative results match findings from regional modelling studies that exist in some basins, we are cautiously optimistic that our findings can be taken as upper-bound estimates for other regions too. The increase in temperature we estimate here could hence be employed by ice-sheet modellers to calculate an upper-bound estimate of the consequences of a flip of all Antarctic cavities into a 'warm' state for current ocean conditions.

While high-resolution ocean modelling could provide a more detailed estimate on the effect of oceanic gateways on melting, our first-order approach is instead straight-forward and easy to run, meaning that only a few analysis scripts are necessary to (re-)produce our results. When new bathymetry or ocean temperature data becomes available, our study can be repeated in an instant, even on a 500 m×500 m grid spacing (within < 30 minutes). The presented approach serves as a refinement on identifying those ocean regions most relevant as input for PICO or other melt parameterisations. We recommend other PICO users to take into account the connectedness of the continental bathymetry when preparing the relevant input data. By identifying potential oceanic gateways and analysing the thermal properties of ambient water masses, our study thus contributes to assessing the current and potential future vulnerability of the Antarctic Ice Sheet to changes in its surrounding ocean.

*Code and data availability.*    The data and relevant code to reproduce the figures are archived at https://(zenodo link will be inserted). Therein, the code to adjust the PICO boundaries are also included. The software scripts to generate and process the access depth fields and ocean data are included and archived at https://doi.org/10.5281/zenodo.14824284 (Kreuzer et al., 2025b).

The connected component analysis code used for the access depth analysis is archived at https://github.com/pism/label-components/ (last access: 19 January 2025; Khrulev, 2024).

The BedMachine dataset is available at https://nsidc.org/data/nsidc-0756/versions/3 (last accessed March 26, 2025; Morlighem, 2022); for the ISMIP6 ocean forcing see Jourdain et al. (2020), ice velocity data are available at https://nsidc.org/data/nsidc-0754/versions/1 (last accessed March 25, 2025; Mouginot et al., 2014) and melt rates from Adusumilli et al. (2020a) can be found here: https://doi.org/10.6075/J04Q7SHT (last accessed March 25, 2025; Adusumilli et al., 2020b).

*Author contributions.* RW conceived the study and together with LN and RR designed the project. LN and RR performed initial analyses and proof of concept. LN made further analyses and together with MK, TA and RR refined the methodology. MK and TA performed the melt experiments. MK optimised the code and analysis workflow. LN made the figures and, together with RR and RW drafted the manuscript, with strong support from all authors.

*Competing interests.* The authors declare that they have no conflict of interest.

*Acknowledgements.* LN was financially supported by a stipend from the Studienstiftung des Deutschen Volkes (German National Academic Foundation). LN, RR, MK and RW gratefully acknowledge support by the European Union's Horizon 2020 research and innovation programme under Grant Agreement No. 820575 (TiPACCs). RW further acknowledges support by the European Union's Horizon 2020 under Grant Agreement No. 869304 (PROTECT), by Deutsche Forschungsgemeinschaft (DFG) through grants WI4556/3-1 and WI4556/5-1. MK was financially supported by Deutsche Forschungsgemeinschaft (DFG) through grant WI4556/4-1 and the Potsdam Graduate School. TA and RW acknowledge funding by the PalMod project (FKZ: 01LP1925D, 01LP2305B), supported by the German Federal Ministry of Education and Research (BMBF) as a Research for Sustainability initiative (FONA). RR was supported by the Natural Environment Research Council (NERC) [grant number NE/Y001451/1 RASP]. This work used resources of the Deutsches Klimarechenzentrum (DKRZ) granted by its Scientific Steering Committee (WLA) under project ID bk0993 (PalMod project). The authors gratefully acknowledge the European Regional Development Fund (ERDF), the German Federal Ministry of Education and Research and the Land Brandenburg for supporting this project by providing resources on the high performance computer system at the Potsdam Institute for Climate Impact Research. Development of PISM is supported by NASA grants 20-CRYO2020-0052 and 80NSSC22K0274 and NSF grant OAC-2118285. The authors thank Ralph Timmermann and Hartmut Hellmer for fruitful discussions at an early stage of the project.

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

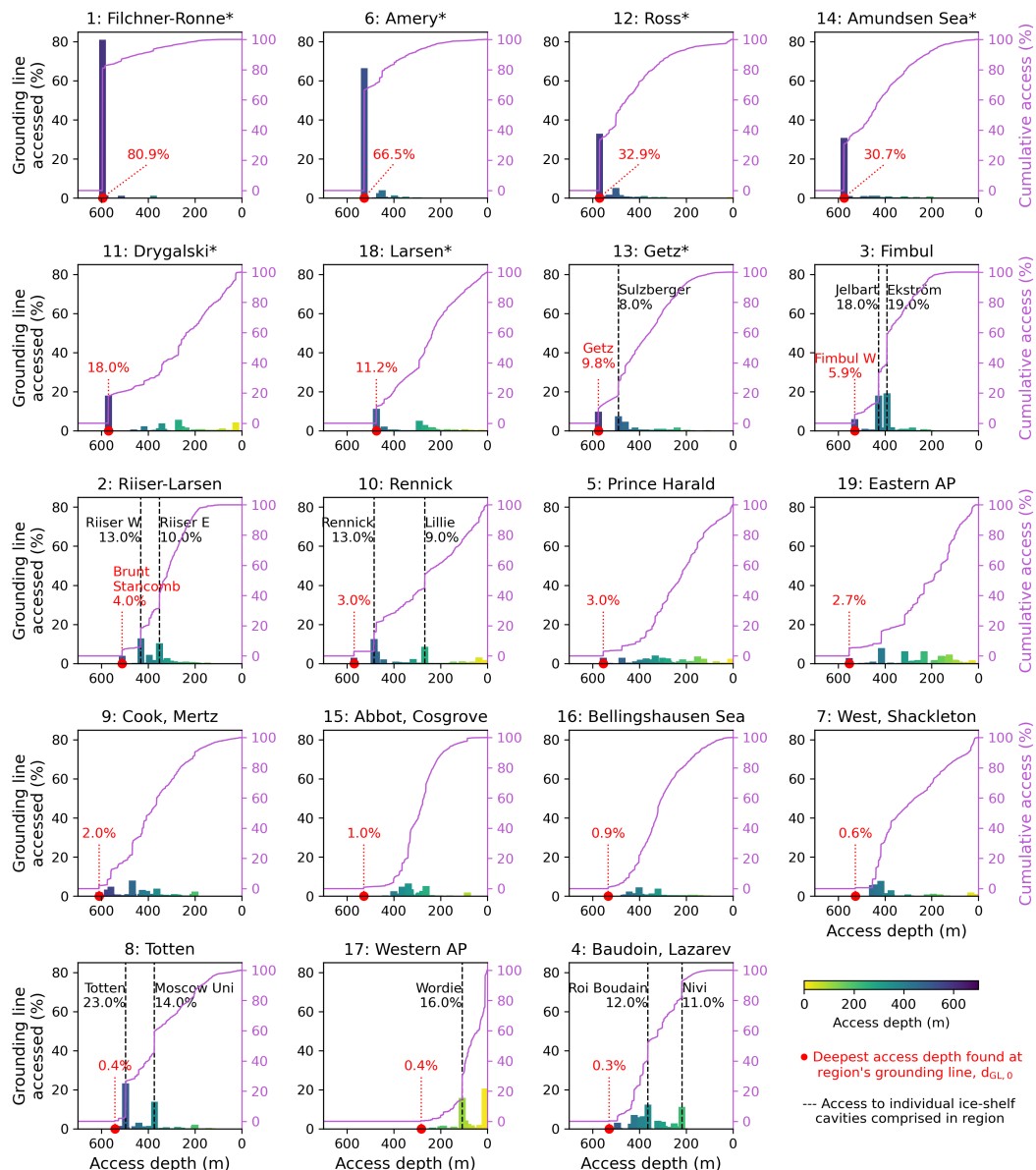

**Figure 4. Distribution of access depths at region's grounding lines.** For each depth level, it is shown how much of the region's grounding line is accessed (fraction given in percent, bin width = 30 m). Magenta line shows the cumulative access, when adding up all depths levels. The different regions are labelled according to prominent ice shelves (with AP = Antarctic Peninsula) and sorted by $d_{GL,0}$ i.e. how much of the region's grounding line is horizontally connected to the open ocean at the deepest access depth found. Where a spike in the distribution can be linked to an individual (smaller) ice shelf comprised in the larger region, the specific access depth is labelled accordingly. The regions follow the drainage basins are based on Zwally et al. (2012), consolidated as in Reese et al. (2018a) and adjusted based on our access depth analysis (but only in the ocean, see above).

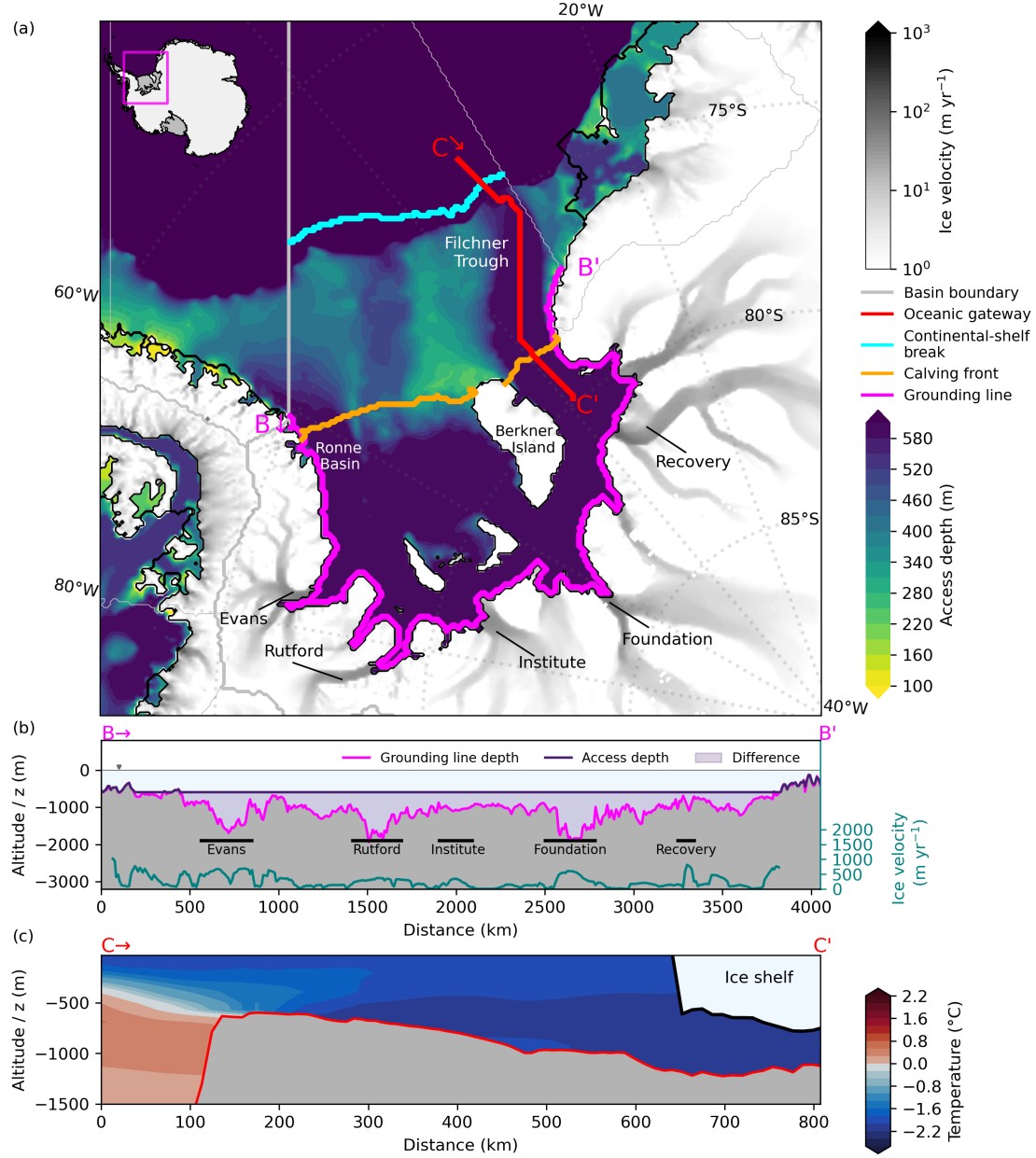

**Figure 5. Access depths and temperature profile at Filchner–Ronne Ice Shelf. (a)** Access depths within the Weddell Sea indicate a prominent oceanic gateway along Filchner Trough towards Filchner–Ronne Ice Shelf. The transects denote vertical profiles along **(b)** the grounding line and **(c)** the oceanic gateway through Filchner Trough showing the potential temperature profile along the transect. Speed of grounded ice in grey shading showing the location of major ice streams (in a) and as blue-green line (in b), taken from Mouginot et al. (2019). Magenta line (in b) indicates grounding line depth, while the dark purple line (in b) shows the derived access depth (e.g. $d_{GL,0} = 595$ m throughout most of the cavity).

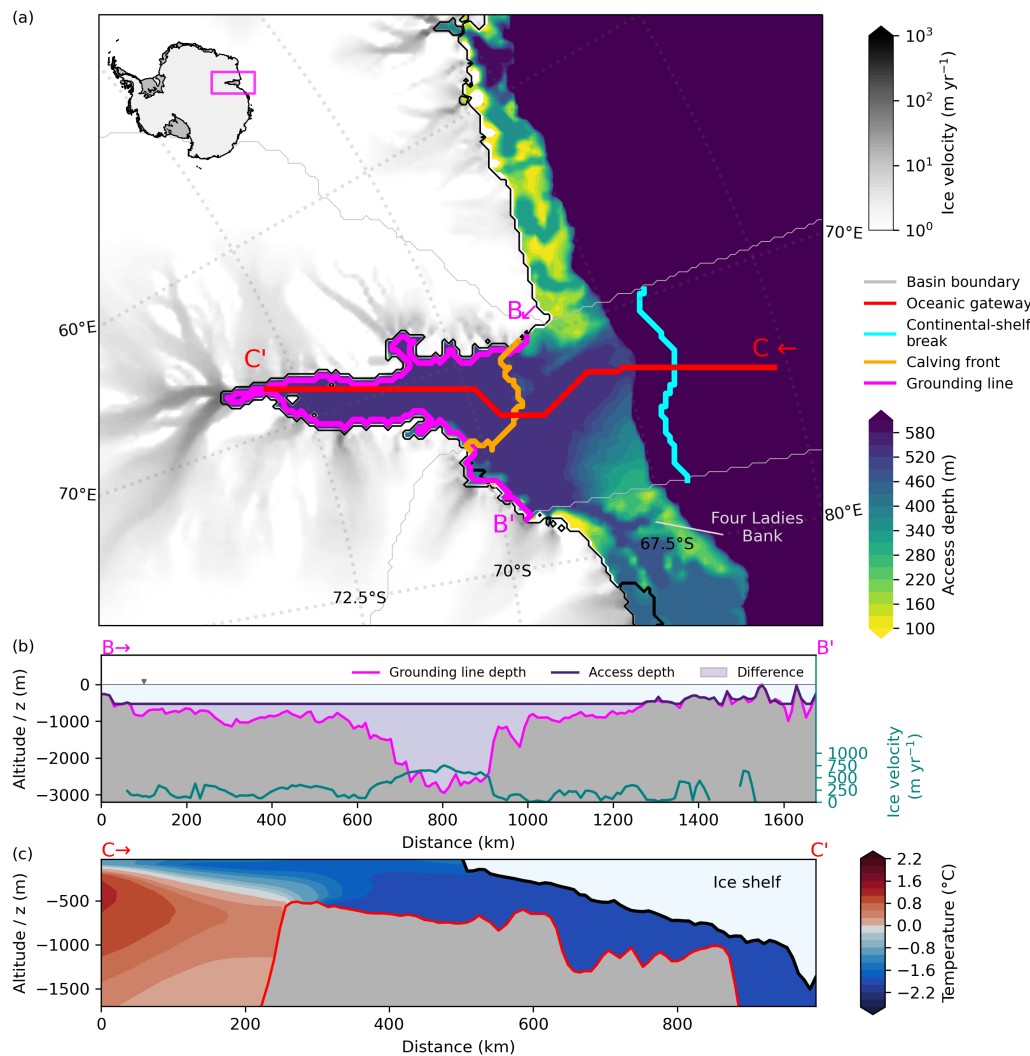

**Figure 6. Access depths and temperature profile for Amery Ice Shelf. (a)** Computed access depths at Amery Ice Shelf indicate a prominent oceanic gateway along Prydz Channel. The transects denote vertical profiles along **(b)** the grounding line and **(c)** the oceanic gateway through Prydz Channel showing the potential temperature profile along the transect. Speed of grounded ice in grey shading shows the location of major ice streams (in a) and as blue-green line (in b), taken from Mouginot et al. (2019). Magenta line (in b) indicates grounding line depth, while the dark purple line shows the derived access depth (e.g. $d_{\mathrm{GL},0} = 525\ \mathrm{m}$ throughout most of the cavity).

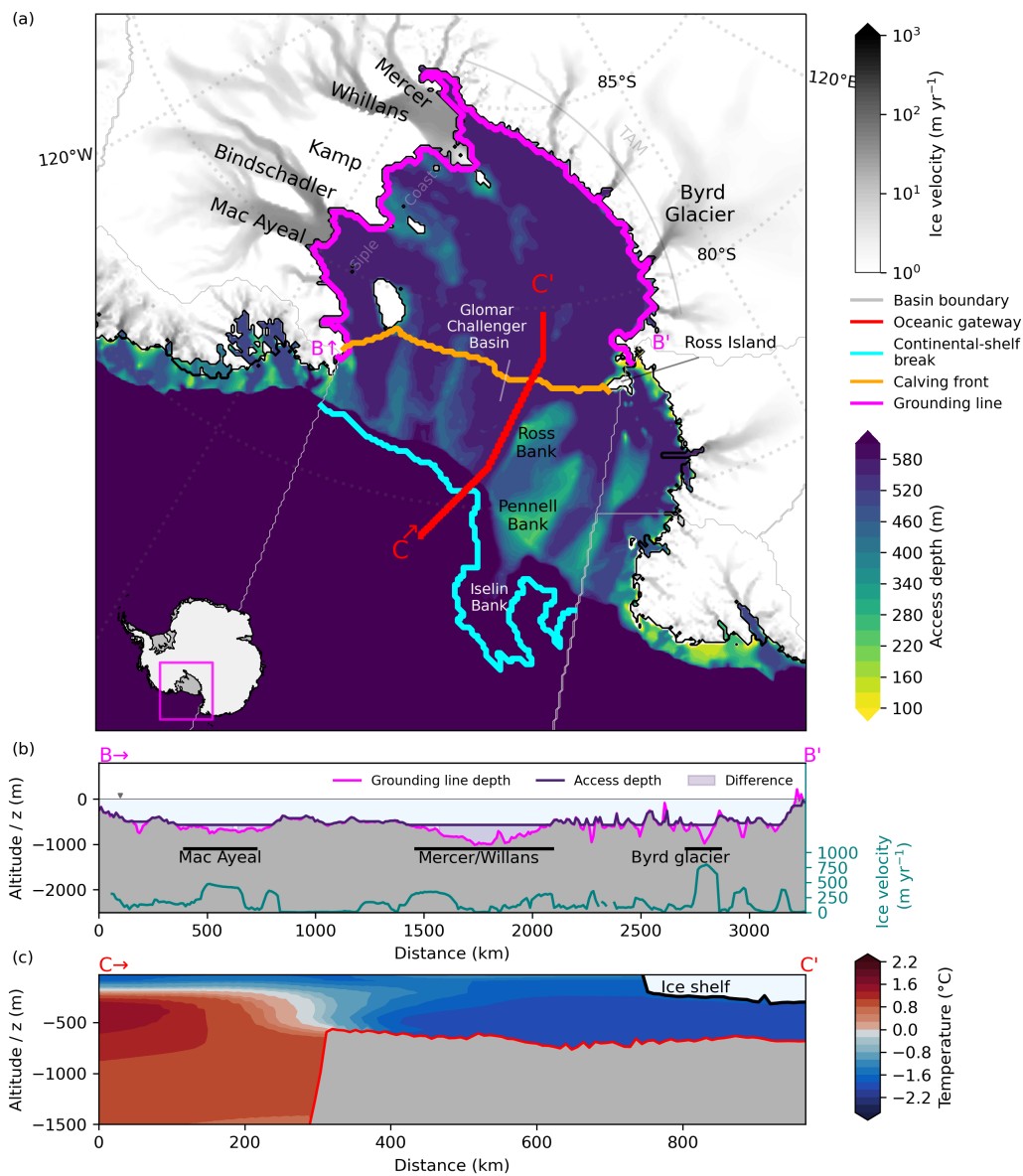

**Figure 7. Access depths and temperature profile for Ross Ice Shelf.** **(a)** Computed access depths in the Ross Sea indicate a prominent oceanic gateway through Glomar Challenger Basin towards Ross Ice Shelf. The transects denote vertical profiles along **(b)** the grounding line and **(c)** the oceanic gateway through Glomar Challenger Basin showing the potential temperature profile along the transect. Speed of grounded ice in grey shading shows the location of major ice streams (in a) and as blue-green line (in b), taken from Mouginot et al. (2019). Magenta line (in b) indicates grounding line depth, while the dark purple line shows the derived access depth. TAM = Transantarctic Mountains.

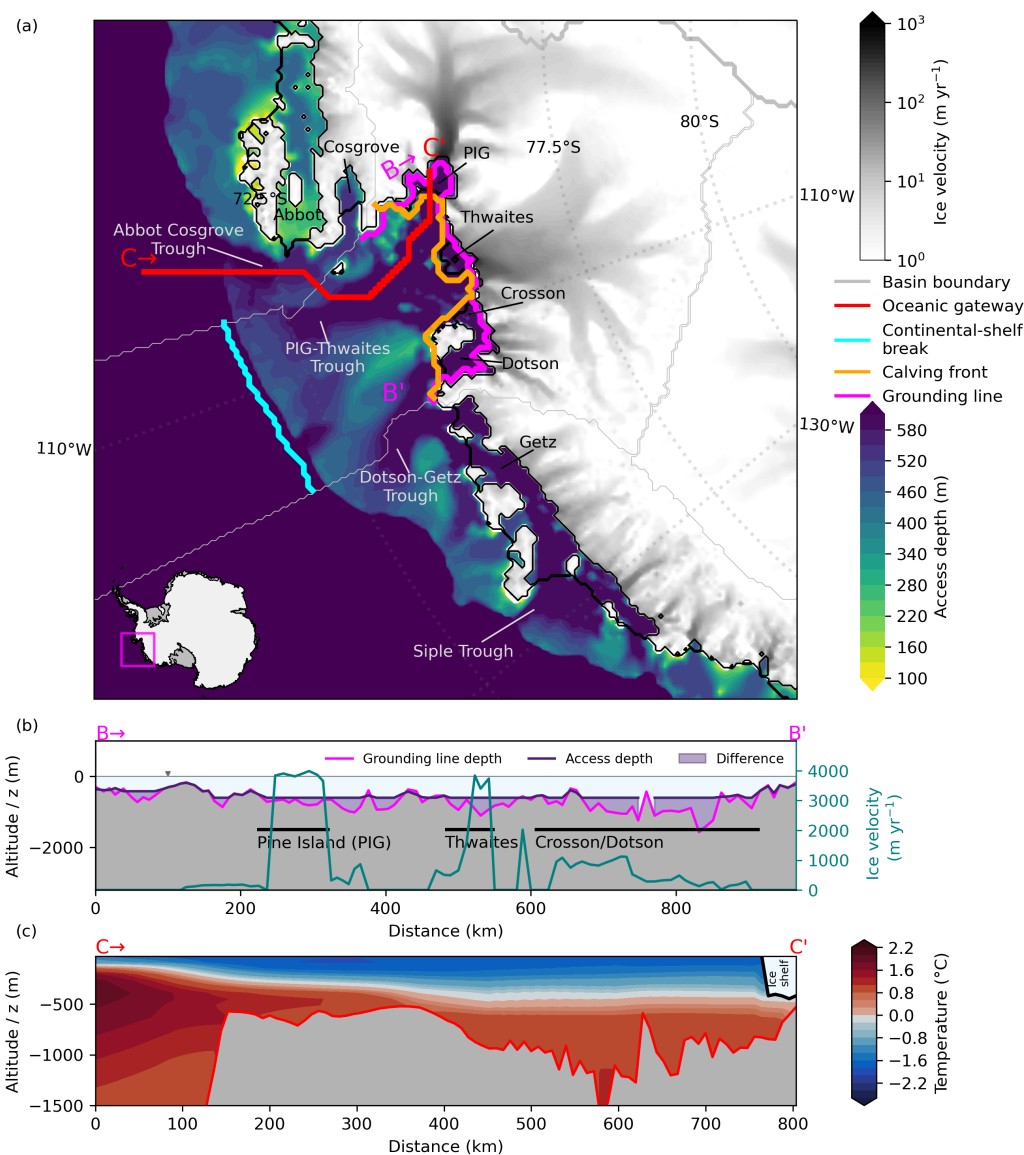

**Figure 8. Access depths and temperature profile in the Amundsen Sea region. (a)** Computed access depths in the Amundsen Sea indicate a prominent oceanic gateway through Abbot Cosgrove Trough towards Pine Island and Thwaites glaciers. The transects denote vertical profiles along **(b)** the grounding line and **(c)** the pathways from the open ocean to the floating extension of Pine Island Glacier showing the potential temperature profile along the transect. Speed of grounded ice in grey shading showing the location of major ice streams (in a) and as blue-green line (in b), taken from Mouginot et al. (2019). Magenta line (in b) indicates grounding line depth, while the dark purple line (in b) shows the derived access depth (e.g. $d_{GL,0} = 575$ m).

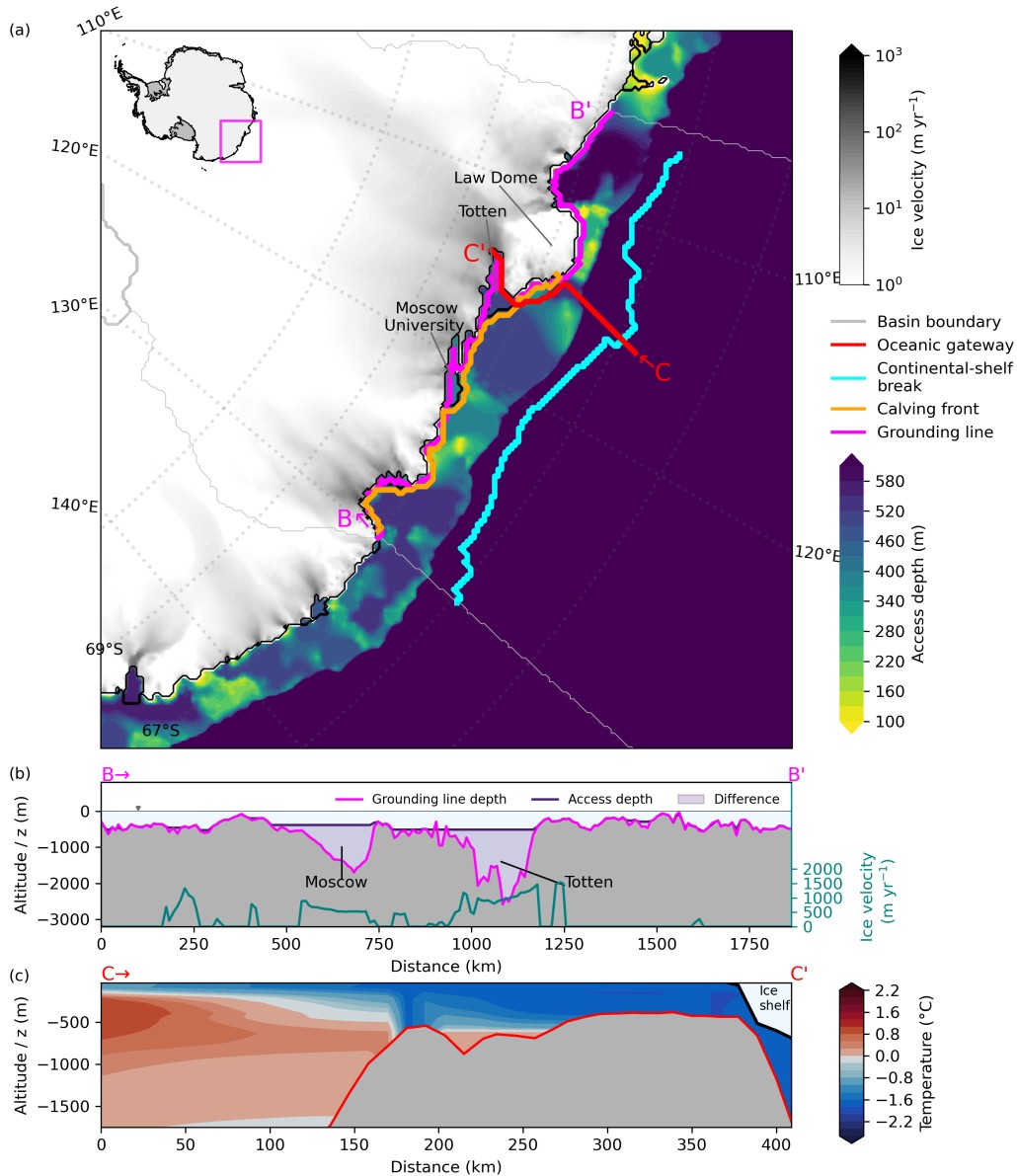

**Figure 9. Access depths and temperature profile in the Totten region, East Antarctica. (a)** Computed access depths near Totten glacier indicate a prominent oceanic gateway near the Law Dome peninsula (cf. transect C). The transects denote vertical profiles along **(b)** the grounding line and **(c)** the found oceanic gateway showing the potential temperature profile along the transect. Speed of grounded ice in grey shading showing the location of major ice streams (in a) and as blue-green line (in b), taken from Mouginot et al. (2019). Magenta line (in b) indicates grounding line depth, while the dark purple line (in b) shows the derived access depth at Totten Glacier.

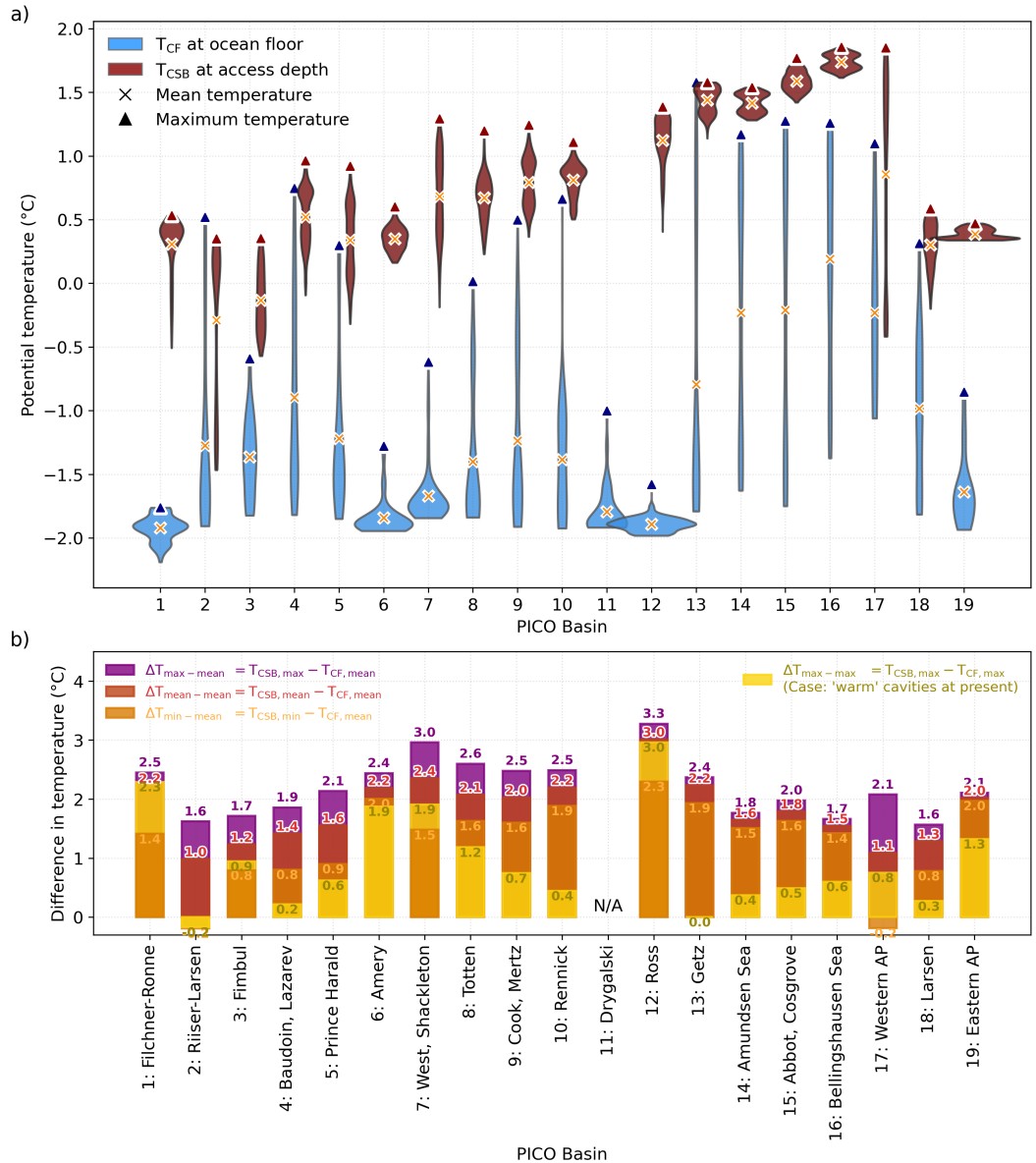

**Figure 10. Assessment of extracted temperatures around Antarctica. a)** The distribution of the $T_{CF}$ (blue) and $T_{CSB}$ (red) estimates for all 19 PICO regions are shown as kernel density estimates, along with the mean and maximum temperatures. The width of the curves depict the approximate frequency of data points within the respective temperature range. $T_{CF}$ incorporates the bottom most temperatures along the calving front, while $T_{CSB}$ is evaluated at the relevant access depth (the deepest along the region's grounding line) at the continental-shelf break, a roughly 40 km wide area where the continental shelf transitions to the open ocean (at 1800 m). Corresponding salinity estimates are found in Supplement Figure S5. **b)** Differences in temperatures from a) when comparing mean temperatures along the calving front with mean, minimum and maximum temperatures found along the continental-shelf break ($\Delta T_{\text{min-mean}}$, $\Delta T_{\text{mean-mean}}$ and $\Delta T_{\text{max-mean}}$ in orange, red and purple, respectively). For the case of 'warm' cavity regions at present, $\Delta T_{\text{max-max}}$ is included (yellow). Temperatures relative to the in situ freezing point, i.e. the thermal driving, is provided in Supplement Fig. S3. Resulting temperature forcing for PICO is obtained by adding the differences in b) onto the tuned forcing fields from Reese et al. (2023), see Supplement Fig. S4.

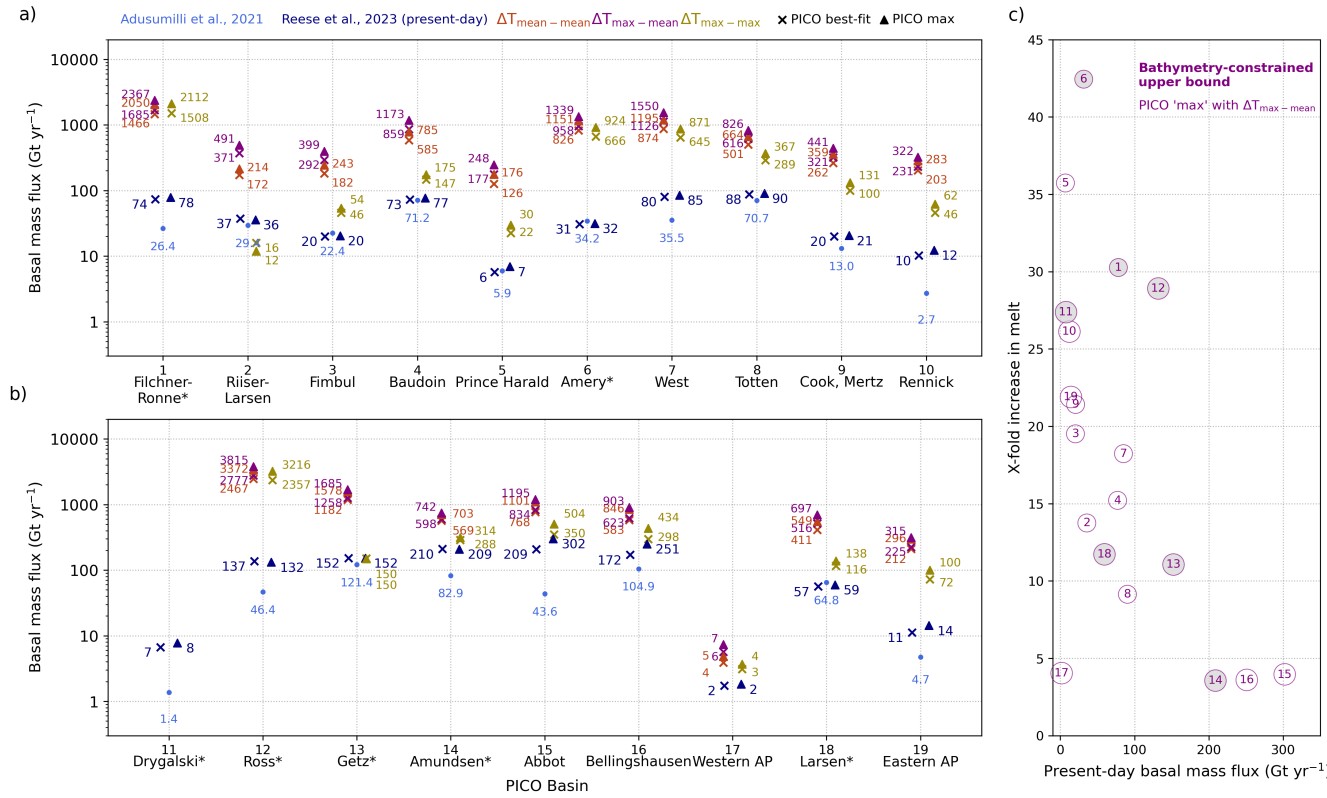

**Figure 11. Circum-Antarctic PICO basal mass flux estimates.** Estimates for basins 1 to 10 (**a**) and basins 11 to 19 (**b**) when assuming an inflow of warm water masses from the continental-shelf break (red/orange: comparing mean temperatures, purple: comparing $T_{CSB, max}$ and $T_{CF, mean}$, and yellow: comparing maximum temperatures), to present-day basal mass flux estimates of Adusumilli et al. (2020a) (lightblue) and the tuned forcing fields of Reese et al. (2023) (darkblue) for two different PICO parameter combination (best-fit and max), respectively. Basin averaged melt rates given in $\mathrm{m\,yr^{-1}}$ are included in Supplement Fig. S6. **c**): Upper-bound estimates using the 'max' PICO parameter combination relative to present-day basal mass flux from Reese et al. (2023) using $T_{CSB, max}$-$T_{CF, mean}$. The circle-markers indicate the respective basin number. The 7 regions we find feature a 'major' oceanic gateway are labelled with * in a) and b); in c) those regions have a grey circle background.