# Peer review of "Bathymetry-constrained warm-mode melt estimates derived from analysing Oceanic Gateways in Antarctica"

_EGUsphere, 2023_

## Referee Comment (RC1)

[referee-annotated manuscript omitted]

---

## Referee Comment (RC2)

Review of:

Oceanic gateways to Antarctic grounding lines – Impact of critical access depths on sub-shelf melt

Lena Nicola et al.

Date: 26 Feb 2024
Assessment: Major revision
Reviewed by: Erwin Lambert

The authors have assessed oceanic gateways for deep water access to the grounding lines of 19 Antarctic basins. Ice shelf melt rates were computed using the PICO parameterisation based on two input forcings: T and S at the calving front, and at the continental shelf break. The melt difference between these two forcings was interpreted as an upper limit for the potential melt increase due to a change in oceanic circulation on the continental shelf (i.e. an onset of an 'optimal mode 2' system).

The idealised approach to representing deep water access and conversion to melt rates is an interesting way to achieve a 'first order' estimate for potential melt increases. Overall the manuscript was an easy read due to a good writing style. I do have a number of concerns though, particularly regarding the methodology, interpretation, and relevance, that in my eyes should be resolved before considering publication. Hence, I recommend a major revision of the manuscript. Below I list my major and minor comments.

Major comments:
1. Motivation / scope. At first (partly based on the title), I thought the main aim was to define the oceanic gateways, and was puzzled by the focus on CF and CSD temperatures. Only when reaching section 3.2, it became clear to me that computing the difference in melt rates (T_CSD – T_CF) was the ultimate goal to quantify the impact of a hypothetical 'mode 2' onset. To better clarify this, I suggest:
   a. Revising the title, with a focus on, in the author's own words, quantifying the melt increase due to 'mode 2' onset, as constrained by bathymetry.
   b. Throughout introduction and methods, clearly write towards this goal. In particular in section 2.2, this should be very explicit.
   c. Include a clear narrative how a switch from CF to CSD forcing may occur (e.g. based on 3D modelling) and why, according to the authors, their approach is the optimal way to quantify the 'first order' impact.
2. Possible errors. I detected two results which the authors should critically reassess, as I believe them to result from errors in the methodology.
   a. In the caption of Fig 3, the authors state that 'parts of the grounding lines are situated above 0 m'. This should not be possible and makes me wonder whether cliff faces (boundary between grounded ice and open ocean) are included. In some regions (5, 10, 11, 17), this appears to be the case for >20% of the assessed grounding line.

b. Present-day temperature at CF in the Amundsen Sea was determined at -0.88 C. This is substantially colder than the +1.1 C observed by Dutrieux et al (2014) at the Pine Island calving front. Note that this is very close to the +1.26 C the authors find at CSD, implying a present day state which is very close to the 'optimal mode 2' state. I also recommend that the authors double-check their tuning parameters, as 20.5 m/yr with a deep water temperature of -0.88 C appears very high to me.

3. Methodology. A few methodological choices have been made which I believe require more scrutiny / quantitative assessment / reconsideration.
   a. By design, PICO takes one value for the deep water temperature and applies this to the first box near the grounding line. Hence, even in cases where g = 10%, the associated temperature at this depth is applied to 100% of the grounding line. The authors should be transparent about this limitation, and ideally give a quantitative assessment of its impact on their results.
   b. The results are highly dependent on the choice of PICO, making it difficult to interpret the results. I would recommend a comparison to the 'quadratic parameterisation' (Favier et al 2019), also used in ISMIP6. This parameterisation can be applied uniformly (as PICO), and regionally (accounting for the limited access to the grounding line). Hence this would allow the authors to quantitatively reflect on point 3a above, and quantitatively assess the impact of the choice for PICO.
   c. It is unclear how the authors treat individual cavities. For example, Totten/Moscow University. The authors focus on a trough providing access to the Totten cavity. Does their method implicitly assume that the same oceanic access exists to the Moscow University cavity? If so, the authors should elaborate on this and estimate the quantitative impact of this implicit assumption. If not, the authors should explain how they deal with separated cavities within the Zwally regions.

4. Relevance. It is always tricky to explain the relevance of a relatively idealised study (trust me, I've been there!). I have some suggestions though to enhance the relevance of this study.
   a. Coming back to the motivation, the authors should have a clear take home message (in abstract and conclusions). Also, it is a bit unclear who the target audience is: the oceanographic community (providing guidance for further research on circulation changes) or the ice sheet community (providing an upper limit to melt increases)? The manuscript would benefit from having a clear target audience and a clear message to this audience.
   b. Where possible, the results should be compared quantitatively to more realistic studies. For example, the authors cite previous studies on Filchner-Ronne (e.g., Naughten et al 2021), but do not compare their quantitative results to those. Throughout section 3.3, I suggest the authors maximise the quantitative comparison to previous numerical modelling studies to place their results in perspective. In addition, this quantitative comparison should be reflected on in the discussion.

c. Also, the authors should mention and reflect on the discrepancy between present-day melt rates (Fig 5a) and observed ones (Fig 5c), and how this impacts their results. In some regions, the relative difference between observations and T_CF is larger than that between T_CSD and T_CF. Can the authors explain this, and convince me and other readers that this does not impact the trustworthiness of their results and conclusions?

d. As the authors state in their introduction, they aim for a 'first order assessment of the maximum changes in temperature and melt'. To interpret this maximum, the reader requires some estimate of the magnitude of uncertainties associated with made changes. For example, how important is it that off-shelf temperatures are assumed to be constant regardless of the circulation change? In fact, the authors provide a narrative for a thermocline shoaling to provide CDW access. How significant is the assumption of constant off-shelf hydrography? The same holds for assumptions like fixed cavity geometries, and the choice for PICO (see points above). The question that remains in my head: could the author's assessed 'maximum' melt increases be twice as high due to unconsidered processes, or is it a reasonable estimate?

e. The authors focus on g=50% in their figures and most quantifications. What is the logic behind this? I would assume that access to the deepest grounding line parts is most important, and would think that g=10% may perhaps be more relevant. Does it make sense to have a fixed g for all regions? Can the discrepancies between present-day melt rates and observations (Fig 5) be (partly) explained by the choice of g? And can a relevant value for g per region be determined from this comparison to observations? How should future research treat these values? Stick to 50%, or optimise it per region? Dedicating one or two paragraphs to this in the discussion would significantly enhance the relevance of this work.

Minor points:

l. 10. the 200-fold larger melt rate is highly dependent on the uncertain reference state. The authors should pick a more relevant metric for their abstract, such as the total increase in BMB (Gt/yr).

l. 16. The concept 'mode 1 and 3' is used, but not really explained. Either stick to more generally known concepts, or give a brief explanation here if it's important.

l. 25. 'tens of metres per year'. Regionally yes, ice-shelf average, this is only possibly the case for Thwaites (and perhaps some tiny ice shelves). Revise this statement and provide a reference.

l. 30. Here the transition to a warm cavity is described, without mentioning it explicitly. As this is a central aspect in this study, the authors should be more explicit here that they are talking about a qualitative change in hydrography and not a smooth warming.

l. 52. Provide a reference for these statements. A highly biased suggestion from my side would be Lambert et al. (2023).

l. 55. Is this related to near-grounding line melt? If so, mention explicitly, if not, remove this statement.

l. 80. (and other places). I don't think 'diagnose' is the correct verb. Replace it with 'parameterise' or something equivalent.

l. 82. Does this dataset include the latest version of IBCSO? If so, reference this explicitly, as Bedmachine uses external sources for its bathymetry. If it's not the latest version, mention this explicitly in the methods/discussion.

l. 84. I don't understand this sentence 'The grounding lines...'. Please revise.

Fig. 1. The straight grounding line is odd, as it does not follow the 'triple point' between ice, bedrock, and ocean. It should deepen in the trough.

l. 98. The access depth is implicitly determined in ISMIP6 as well. Explain clearly what the added benefit is of your methodology in reference to ISMIP6. This difference/overlap is a bit unclear to me.

l. 110. The equations and text here do not make the whole methodology very clear (to me..). Consider visualising this, either in Fig 1 or in a new schematic figure, so the reader fully understands what's going on. I'm strongly in favour of a new (schematic) figure which also illustrates the parameter g.

l. 131. 'Input is based...' Does this refer to Reese et al '23? If so, mention explicitly, if not, explain why you deviate from the ISMIP6-based forcing here.

l. 134. Most ice sheet models have PICO included, so I think this is an irrelevant statement (which causes more confusion than clarifying anything). So rather remove it.

l. 159. First $T_{CF}$ is 'generally lower' than $T_{CSB}$, in the next sentence, $T_{CSB}$ is 'much warmer than $T_{CF}$. Align these two statements.

Fig 2. For the bars, T=0 is used as a reference, which is a bit arbitrary. I'd use freezing temperature as a reference for visualisation (so that the bar heights reflect the Thermal Forcing).

l. 172. 'highest grounding line depths' -> 'shallowest/deepest grounding line depths'

l. 218. Compare 4.65 m/yr to numerical modeling studies.

l. 233. Is this other pathway deeper? How is it (or can it be) relevant to your study?

l. 318. These kinds of sentences are fine, but compare to other studies where possible.

l. 328. Is it possible to quantify the difference between CDW and mCDW? For example looking at Amundsen Sea (observed $T_{CF}$ = 1.1; $T_{CSB}$ = 1.26). Does this example mean that the difference is negligible?

l. 335. Good point regarding the trough width. What ocean dynamics control this minimum width? Are there specific troughs you highlighted in this study which are very narrow where this may have an impact? How can you/future researchers incorporate this concretely?

l. 397. MISI is new info, which should not appear in the conclusions. If it's relevant, include it in the intro/discussion. Conclusion should only contain previously presented information that is specific to your study.

l. 409. Again, geoengineering is new info. Put this in the discussion if you want to include it; stick to your own work and its implications/relevance in the conclusions.

References:
Dutrieux et al. 2014. https://doi.org/10.1126/science.1244341

Naughten et al. 2021. https://doi.org/10.1038/s41467-021-22259-0

Lambert et al. 2023. https://doi.org/10.5194/tc-17-3203-2023

---

## Author Comment (AC1)

**Additional information on preprint** *Oceanic gateways to Antarctic grounding lines – Impact of critical access depths on sub-shelf melt* **(DOI: 10.5194/egusphere-2023-2583)**

**How we derive access depths based on bathymetry data**

We determine the access depth of each point on the continental shelf (including the ice-sheet's grounding line) through a flood-fill algorithm (for more information on this algorithm, see e.g. https://en.wikipedia.org/wiki/Flood_fill, last access: 16.01.2024). We let the algorithm start in the open ocean (off the continental shelf) at a depth of -2000m. Like a slowly filling bathtub, the flood-fill iterates through the dataset grid and fills / connects pixels that are connected (at the same or lower depth). Since the continental shelf lies shallower than -2000m, at a certain depth the flood-fill "flows" onto the continental shelf. This is indicated by the white arrows in Figure 1. Once done, one can then compare the bathymetry value from the BedMachine Dataset with the value from the Access Depth array, which results from the flood-fill: if the bathymetry is deeper than the derived access depth at the same point x,y; the point may be "shielded" by shallower topography "blocking" the way of e.g. warm water onto the continental shelf. Evaluating the access depth array as a 2D field (Figure 2), and more closely at the grounding line we can discuss the connectedness of the cavity and grounding line to the open ocean.

[Figure]

**Figure 1.** Transect at Ross Ice Shelf showing how flood-fill algorithm determines access depth at grounding line (marked with magenta dot). Location of A and A' are marked in Figure 2.

[Figure]

**Figure 2.** Overview of 2D-field of obtained access depths at Ross Ice Shelf.

In Figure 2, one can see that for instance Mercer Ice Stream has a deep access depth; other parts of this region's grounding line have shallower values of access depths. To quantify this situation, we express the critical access depth of the region's grounding line in terms of how much percent of the total grounding line in the Ross Sector has the access depth of d=x m. For example, for the Ross Sector, 30% of the grounding line has the deepest access depth of -570m. This evaluation of ocean access to the different sectors' grounding lines is displayed in Fig. 3 of the submitted manuscript. The different concepts can be summarized in the following Figure 3:

[Figure]

**Figure 3.** Overview of used concepts applied to the Ross Ice-Shelf region.

Two animations showing how the flood-fill iteratively fills cells around Antarctica will be added in the supplement of the revised manuscript.

---

## Author Comment (AC2)

*We thank the three reviewers for carefully reviewing our manuscript and are happy to strengthen our study according to their suggestions.*
*In addition to our initial reply to RC1, in the following, we provide a general response to the overall and key aspects mentioned in the three reviews together, as well as a point-to-point response to all reviewer comments respectively.*

*Our point-to-point responses to the individual comments are given below in blue and italics compared to the reviewer comments which are given* in black without italic font.

**Author response to key aspects common to all RCs**

*With regards to the comments made on **novelty and relevance** of our study, in the revised manuscript, we try to be more direct about **our goals** and **main motivation**. Namely, we aim to estimate the Antarctic Ice-Sheet's vulnerability to changes in ocean thermal driving. To get a rough estimate of potential changes in thermal forcing, we identify features from continental shelf bathymetry that are similar to Filchner Trough which could provide warm water masses access to the grounding line once the cavity switches into a "warm" state. Furthermore, we calculate the connectedness of Antarctic grounding lines to the open ocean. Based on this, we derive an upper-bound of temperature (and thus basal melt rate) changes in ice shelf cavities around Antarctica.*

*We strengthen the relevance of our study by additionally providing the derived access depths data for a 500m x 500m grid spacing in the supporting material as well as the code to update the access depths when new data becomes available. In a revised manuscript, we will further extend the section on how our findings may influence PICO tuning in future studies and how this relates to the current settings and other melt parameterizations. To this end, we will now also include an updated delineation of basin boundaries for PICO in the ocean based on our findings (taking up the suggestion made by RC1).*
*As suggested by RC2, we will also revise the title of our manuscript to strengthen the main message i.e. that we look into bathymetric constrained warm mode melt estimates derived from analyzing Oceanic Gateways in Antarctica.*
*As suggested by RC2, we will overall stream-line our manuscript to fit our target audience, namely stand-alone ice-sheet modelers (using PICO) and those working on the coupling to coarse grid ocean models, as our study provides bathymetry-informed estimates for temperature and salinity as used in basal melt parameterizations such as PICO.*

*Concerning the comments made towards our **terminology and methodology,** we will include a more in-depth and more comprehensive explanation of the concept of access depths, through a revised Figure 1 and by rephrasing 1) how access depths are defined and derived, 2) how we define critical access and 3) by showcasing how we conducted the analysis, providing estimates on the computational cost and the numerical code of the flood-fill.*
*In the revised manuscript we will also add further explanations how our approach relates to the **ISMIP6 protocol,** i.e. that we foremost analyze bathymetry and subsequent grounding line connectedness and that our analysis can be adapted to whatever ocean dataset there is available.*

*To reduce complexity, we will rework the use of **our parameter "g"** that we introduced to describe how much percentage of a basin grounding line can be accessed by open ocean*

*water masses at depth. When revising our quantitative results, we will now additionally exclude all parts of the ice sheet where ice is grounded >0m (as correctly pointed out by RC2). We will propose a new Figure visualizing the distribution of access depth at the grounding line (formerly Figure 3).*

*Generally, **more than one ice shelf** can be included in one PICO basin. We will make sure to discuss related caveats in the revised manuscript (which was especially raised in RC2). We will clarify the use and definition of a **prominent gateway**, that we had initially defined as "one or several deep troughs that provide access to most of a region's grounding line". Here, "most of a region's grounding line" had referred to 10% or more of the grounding line accessed at one distinct access depth level in our preprint and was highlighted by the magenta boxes in Fig. 3. In regions where we do not see this feature, but a rather gradual increase in access with grounding line portion/fraction, we actually cannot state that an oceanic gateway is present.*

*In the revised manuscript, we will update our **temperature estimates** and **change the overall narrative on our two scenarios.** We will refrain from speaking about "warm-water intrusions"; instead, we will give an "upper-bound" estimate, as we rather consider a macro-scale/basin wide transition in melt mode associated with prevalent access of ocean water from off-shore. To this end we will also assess how the T_CSB estimates change when taking the maximum (instead of the mean) temperatures along the CSB. This would follow our refined intention of a bathymetry-derived upper bound to melt rate changes. Further, we will clarify the paragraph in which we define the temperature estimates and add further discussion on how ocean temperatures may change (e.g. CDW->mCDW) when intruding onto the continental shelf (what we do not resolve). While we take the temperature estimates T_CSB as proxy for mode 2 melting conditions, we will state in the revised manuscript more clearly that T_CF is representative for mode 1 melting (after Jacobs et al., 1992). To this end, we will consider the bottom temperatures at the calving front, instead of averaging them at the overflow / access depth of that basin.*

*In order to resolve the discrepancy of our estimated "present-day" melt rates to observations (as pointed out by all three reviewers), we will change our methodology for the melt estimates as follows:*

*Our used PICO parameters from Reese et al., 2023 were tuned to represent bulk present-day melt rates as well as to match the melt sensitivity at Filchner-Ronne (cold based) and Amundsen Sea ice shelves (warm based). In the tuning process, the input temperatures from Schmidtko et al., 2014 for each basin were adjusted so that melt rates, as well as melt rate sensitivities, would be in line with observations. These necessary, yet to a point unphysical, temperature corrections can hence be seen as an additional factor in the tuning. To be consistent with the tuned parameters, we will propose to take the forcing field from Reese et al. 2023 as present-day baseline temperatures. For estimating "upper bound" estimates of bathymetric-constrained warm mode onset, we will then add the difference of T_CSB minus T_CF (both derived from ISMIP6 dataset) to the existing forcing field. This follows the same "anomaly idea" taken in Kreuzer et al. (in discussion, [doi.org/10.5194/egusphere-2023-2737](doi.org/10.5194/egusphere-2023-2737))). We will make sure to expand the explanation of these temperature adjustments within the PICO tuning process in our method section.*

*Once we have new estimates we will include a more thorough comparison of our temperature as well as basal melt rate estimates to findings from previous literature, specifically in the key regions that the reviewers mentioned e.g. at Ross Ice Shelf and in the Amery region (both mentioned in RC1 on page 12), the gateways we find in the Amundsen Sea (esp. Abbot Cosgrove Trough, mentioned in RC3) as well as our temperature estimates in this region (mentioned in RC2), and subsequent melt rate estimates.*

*We will clarify the sign convention of z vs. depth and align it with commonly used definitions (in reply to RC1) and we will provide melt estimates in Gt/yr (in reply to RC2). We will further rework our Figures as suggested by all reviewers.*

*We will further gratefully take up the suggested language changes to specific wording within the text (see respective point-to-point response below).*

*For the specific comments made by the individual reviewers, please consult the respective point-to-point responses.*

**Response to RC1**

Nicola et al. use the BedMachine bathymetry product and ISMIP6 dataset (observations + extrapolation of temperatures (T) and salinities (S) around Antarctica and its continental shelf) to discuss some features of these datasets, for example the relevance of different troughs in delivering heat to the grounding lines (for BedMachine) and the spatial differences in ocean temperatures and salinities (for ISMIP6 dataset).The authors then calculate ice shelf basal melt rates using the box model PICO, where the T/S input is a) conditions at the calving front and b) conditions at the continental shelf. The latter is presented as the "upper limit of melt rate changes".

*Dear Anonymous Reviewer 1,*
*Thank you very much for your detailed review of our manuscript and your feedback.*
*In addition to our initial reply to your comment (https://egusphere.copernicus.org/#AC1), please find our detailed point-by-point responses (written in blue and italics)* to your comments (in black) *below.*

**General**

I have to admit that I have struggled with the aims and the novelty of this manuscript.
*We are sorry that the initial manuscript lacks to convey the aims, novelty and relevance of our approach clearly enough. In the revised manuscript, we will more clearly lay them out, as mentioned also in the general comment above.*

The first half of the paper introduces new terminology of 'oceanic getaways' and 'critical access depths' but it is really just talking about bathymetric features, specifically troughs that have received a lot of attention in the past decades as the sea floor around Greenland and Antarctica have become better mapped.

*Yes, we analyze the bathymetric features in the used dataset. However, we consider those features in context of a whole basin geometry and further perform a systematic circum-Antarctic assessment of bathymetric access and potential pathways of ocean water to the grounding lines of the Antarctic Ice Sheet, which we believe is not included in publications related to new bathymetry surveys or ocean cruises that focus on specific regions.*

The same flood-fill algorithm that the authors use here was also used to produce and extrapolate the ISMIP6 dataset, so I don't think that part is anything new. The next bit of the paper that discusses ocean properties at the calving front vs the continental shelf doesn't do much more than stating the differences in the fields in ISMIP6 dataset and some generalities.

*Our study extends or rather complements the work by the ISMIP6 focus group. Our approach of quantifying the connectedness of the grounded ice to the open ocean takes into account the depth of the grounding lines. While the basic concept of the flood-fill is the same, their code is different, as it serves a different purpose. Their code can be found, to our knowledge at https://zenodo.org/records/3997257 under ismip6-antarctic-ocean-forcing-1.0/ismip-ismip6-antarctic-ocean-forcing-7ed686c/ismip6_ocean_forcing/extrap/ (last access 08.04.2024).*

*In our study, we do not extrapolate temperature and salinity values into ice-shelf cavities, but use the flood-fill algorithm to derive the access depths, to the goal of having a grounding line / geometric informed depth value, at which we can derive our PICO input from.*

The section "Oceanic gateways to major Antarctic ice shelves" is a mixture of literature review, and speculation about potential impacts of high melt rates (as calculated by using shelf break instead of calving front temperatures) - but the impacts are not modeled here so just a brief mention in discussion would be enough for that part.

*This section was meant to put our regional results into perspective to the existing literature, whether the main gateways we have identified correspond to the main inflow regions that we can observe/model today or in the future. We consider the potential impact of a regime shift by stating the change in melt rates. As we see our study as a sensitivity study of melt rates to a regime shift in all Antarctic regions, we would consider modeling the long-term ice-dynamical response (and eventually disintegration) of the ice sheet beyond the scope of our study.*

For the literature review - a lot of this refers to studies about ocean circulation beneath the specific ice shelves, but all that is ignored in PICO, so I don't see why that is reviewed here, since the box model doesn't know anything about horizontal circulation.

*We have included the mentioned literature in an attempt to discuss the limitation of our simple approach, e.g. using PICO that does not take into account the horizontal circulation inside the ice-shelf cavity. When reframing our manuscript (see comment above) we will re-evaluate which literature is needed to support our storyline and consider moving this specific part into the discussion section in the revised manuscript.*

Other references serve to show that bulk present day PICO melt rates are reasonable, but that was already tuned elsewhere in previous publications, so not sure why that is necessary here again.

*We have included these references in our manuscript to justify the use of PICO and the chosen parameter settings.*

I think that especially with the simple box model it is really easy to produce large increase in melt rates for any given ice shelf, all that is needed is a change in input temperature that is fed into box 0. Since there are no oceanic processes accounted for that would be allowed to mix or divert away this change, the model essentially by construction contains a tunnel that conducts outer T and S directly to the grounding line. So the result that large change in input (which is chosen here but not really physically justified) causes large change in output, which is the result of this manuscript, is definitely not a surprising one.

*It is correct that melt parameterizations like PICO assume a direct connection of ocean conditions outside the cavity and conditions at the grounding line, which neglects the mixing and entrainment processes. However, there are two PICO parameters, associated with the overturning strength and the turbulent temperature exchange respectively, to be tuned for whole Antarctica to produce a large range of realistic present-day melt rates (and different melt modes) for given outer temperature and salinity inputs. As the PICO tuning also covered the basin-specific melt rate sensitivities, we have some confidence in melt rate estimates for the range of temperature changes we find from the different input regions/depth levels.*

The question is whether the numbers produced here for the increased melt rates are realistic or otherwise useful in some way. I don't think the authors have even tried to make a case for either usefulness or realism of these high melt rates. The only argument that was provided in the paper is that this approach here is "straight-forward and easy to run" but without it being realistic or useful, simplicity on its own, is not enough of an argument for publication.

*Once we have our updated temperature, salinity and basal melt rate estimates we will include a more thorough comparison to findings from previous literature. In the revised manuscript we will also propose options for applications in prognostic ice-dynamical model studies.*

A general characteristics of this paper is that the authors state assumptions but don't justify them. A good example of an unjustified assumption is the one that grounding lines are always accessed via 'prominent getaways' - that clearly doesn't hold in present day for the cold ice shelves Ross and Filchner-Ronne and others, yet this inconsistency is not at all addressed.

*This was clearly a misunderstanding: We do not assume that ice-shelf regions are at present accessed by a prominent gateway. The first part of our study tries to identify gateways in all regions to find out whether or not there is a potential of an inflow at depth (and at what depth). In a second step, we want to estimate the potential change in melt rates assuming the continuous inflow of warm water masses from the continental-shelf break would be channelled through these gateways in all regions (if gateway-structure exists). We are sorry that the preprint did not convey this clearly enough. In a revised manuscript, we will make sure that all our assumptions are more clearly laid out.*

Also there is some misuse of terminology. For example the temperature the authors have chosen for the CSB T and S is quite arbitrary, yet they call their perturbed melt rate result "upper limit". Surely not everywhere is this arbitrary point the temperature max along the shelf break, so even higher melt rates could be reached with PICO.

*For a revised manuscript we will provide updated temperature estimates, with further explanations on where we would find the highest temperatures adhering to our upper-bound narrative.*

I don't think the term warm-water intrusion is accurate for the use in the context of a long term, large scale and lasting change. Intrusion is an intermittent oceanographic feature. The sensitivity study here assumes that oceanographic conditions within the cavity change, that is warm water from the open ocean comes across the continental shelf break and stays and that is something very different and more difficult to establish than an intrusion which would largely mix in with other water masses on its way to the grounding line and become much cooler and fresher by the time it comes in contact with the ice.

*Thank you for this comment. This is a good point and we will refrain from the term "warm-water intrusion" in the revised manuscript, as we rather consider a macro-scale/basin wide transition in melt mode associated with prevalent access of ocean water from off-shore.*

The PICO model has some clear biases compared with the observations of Adusumilli et al. 2020, namely it overemphasizes a melt rate pattern of high melt at grounding line and low melt towards the front and does not take into account the 3D structure of the circulation, which results, for example, in omitting mode 3 melting features near ice shelf fronts. Accordingly, the melt rates 'assuming warm water intrusion' have the same biased melt rate pattern as the original PICO melt rates except now the melting is higher. Can you comment on the bias and its implications? For example in the context of Reese et al 2018 - if grounding line are most sensitive to melt rate change, overestimating melting there could be problematic, yet it is probably happening since the bulk melt rates are tuned to agree with observations and mode 3 is absent - resulting in freezing or low malt rates near the front - positive bias in grounding line melt rates is clearly visible in sectors 3-5.

*In the revised manuscript we will include a more thorough discussion about the biases introduced by PICO. Please also refer back to our explanation in our initial reply to your comment that we have posted on February 2nd, 2024:*
*https://egusphere.copernicus.org/#AC1*

**Specific**

Access depth seems to be a key concept here but it is not clearly defined (I think the language is the problem). Figure 1 doesn't help - it is stated in the text that access depth is a field defined everywhere (and provided on a certain discrete grid) but the figure only points to a single point in the image, which is confusing. Further on Fig one - what is the -1800 m in the image showing horizontal distance? Shouldn't that be depth for the purposes of your continent definition?

*We will provide a new schematic Figure 1 to highlight key concepts used in the study. In this new Figure it will be shown how, from a 2D access depth field, we can extract the access depth at the grounding line. The z=-1800m contour is indeed used for the purpose of our continent shelf definition.*

Similarly g is not clearly defined. From the paper it is sort of clear what the authors mean from the context but that is relying on the reader being on board with the writers.
*A more comprehensive explanation will be given in the revised manuscript.*

Sign convention of z vs depth needs to be consistent.
*Will be clarified in the revised manuscript.*

Fig 7 and similar - x axis needs to be labeled on each subplot to make clear what distance is meant for each case
*Will be clarified/changed in the revised manuscript.*

other specific comments are in the attached pdf

*Please find below the individual replies to the comments that we extracted from the provided PDF using the function "Print comment summary" in Adobe Acrobat 2017. If necessary, we have cited the corresponding text passage* in black.

**Supplement comments (in attached PDF)**

**Page: 1**
l. 14: Reviewer Subject: Comment on Text: "Sub-shelf melting around Antarctica varies by orders of magnitude"
the term is typically "ice-shelf basal melting"
*Will be changed in the revised manuscript.*

**Page: 2**
l. 40:  Reviewer Subject: Highlight
in terms of sign convention, I think z is typically defined up from sea level positive and down negative, but depth (in oceanography) is positive down (defined as negative of z)
*Will be clarified in the revised manuscript. In the revised manuscript, we will use positive values when referring to "depths".*

l. 43: Reviewer Subject: Highlight
do you mean bathymetry here?
*In this manuscript, we look at bathymetry solely, but the data product (BedMachine) also covers the Antarctic subglacial topography, hence we said "topography data". We can call it bathymetry to make it more clear what we are focusing on in the study.*

l. 50:  Reviewer Subject: Highlight
is "effective erosion" a technical term different from "erosion" or do you just mean the effect of erosion?
*We used it to describe that erosion is very effective over long time scales. The word "effective" will be deleted in the revised manuscript for clarity.*

l. 50f.: Reviewer Subject: Highlight
but then higher near the very front again due to mode 3, and the reasons for this pattern are different in different types of cavities so maybe includign a line or two elaborating on this statement might help a more general reader
*Thank you for pointing this out to us. We will rephrase this part in the revised manuscript.*

l.51:  Reviewer Subject: Highlight
not just further modulated but that is also how it is set up to begin with
*We will omit the word "further" in the revised manuscript.*

l. 52: Reviewer Subject: Highlight

Coriolis effect
*Will be changed to "(e.g. the Coriolis effect)" in the revised manuscript.*

**Page: 3**
l.59f: Reviewer Subject: Highlight "..., but only a few studies investigate the bathymetric access points or pathways to the grounding lines in detail and…"
this is definitely untrue, the role of bathymetry in local sub-ice-shelf circulation and warm water access is the subject of many studies, typically that happens as soon as new bathymetry survey or ocean cruise takes places in the follow up publication
*We perform a circum-Antarctic assessment of bathymetric access and potential pathways of ocean water to the grounding lines of the Antarctic Ice Sheet, which we believe is not included in publications following up on new bathymetry surveys or ocean cruises that focus on specific regions. We will change the statement in the revised manuscript.*

l. 62: Reviewer Subject: Highlight *"assuming that water follow this pathway"* this is a strong assumption, and in case it is not like that reality, then I don't see a justification for this assumption
*In at least two modeling studies focusing on the Filchner-Ronne Ice Shelf we see an inflow through Filchner Trough (Hellmer et al., 2012, Naughten at al., 2021). Our study provides an analysis for a sensitivity-experiment, where in case of a trough-like feature, we assume the access of off-shore ocean water is possible (as in the case of Filchner Trough), but our model cannot tell under which conditions this access could be realized. Our assumption that, once warm CDW is flowing onto the continent, it will eventually reach the grounding line can be motivated by the fact that CDW is not only warmer but also saltier and therefore denser than on-shelf waters, such that we expect it to sink from the shallowest overflow point, eventually towards the grounding lines, and filling up the cavity basin, replacing the less dense waters below access depth.*

l. 68: Reviewer Subject: Highlight
which current?
*This was meant as "present-day" estimates, which we took as synonyms of T_CF conditions. Will be rephrased in the revised manuscript.*

l. 68: Reviewer Subject: Highlight
potential with respect to what?
*This was meant as the potential change in melt rates with respect to the estimate derived from the calving front (T_CF). Will be rephrased in the revised manuscript.*

l. 70ff: Reviewer Subject: Comment on Text: "Our approach of identifying relevant water masses that drive melting in cavities is also useful to improve the input for parameterisations of sub-shelf melt rates such as the ice-shelf cavity model PICO as suggested in Burgard et al. (2022)."
it is unclear what you mean here at this point, maybe more relevant for discussion section
*This sentence serves as motivation and framing of our study, as in some basal melt rate parameterizations depth levels for each basin have to be selected over which ocean temperatures and salinities are averaged to feed into the model. Here the concept of access depths for warm water masses off the continental shelf might be of interest for the ice-sheet*

*modeling community using such simple models. We will rephrase this section in a revised manuscript.*

l. 87f: Reviewer Subject: Highlight
I don't understand this definition, how are the routines defined, and what
decides which value is assigned to each cell?
*The flood-fill algorithm iterates through the grid and from a seed point spreads out in all four directions (the code checks the four neighbors of the current point: up, down, left, and right ), fills adjacent cells with a specific value e.g. "flooded". until it reaches boundaries or encounters obstacles (cells that are not below the threshold, i.e. shallower bathymetry).*
*For clarity, in the revised manuscript, we will add additional explanations on the used flood-fill algorithm, provide a new Figure 1 explaining the key concept, include the flood-fill code, as well as explanatory animations (see https://zenodo.org/records/10599774) in a supplement.*

l.88: Reviewer Subject: Highlight
Here is where your sign convention of depth becomes complicated - deepest means largest positive depth, which according your definition would be zero, as depth is negative below sea level - I don't think that is what you mean here
*Will be changed/clarified to "the deepest level (largest positive depth)" in the revised manuscript.*

l.88: Reviewer Subject: Highlight
from?
*Yes. Will be changed in the revised manuscript.*

**Page: 4**
l.89: Reviewer Subject: Highlight
how do I know which one?
*This can be done by comparing the bathymetry value from the BedMachine with the value from the Access Depth array (which results from the flood-fill): If the bathymetry at point x,y is deeper than the derived access depth at point x,y; one knows the point is "shielded" by shallower topography "blocking" the way of e.g. warm water onto the continental shelf.*
*The difference between access depth and BedMachine (on a 8km x 8km grid) is plotted hereinafter:*

[Figure]

*Fig. 1: Difference between the access depth field from the initial manuscript and BedMachine bathymetry input on an 8kmx8km grid spacing.*

*We plan to include such a figure in the supplement of the revised manuscript.*

l.93: Reviewer Subject: Comment on Text
this needs a more precise definition, it might be good to define d_c and g in two different sentences, two definitions in one is too much to be clear
*Thank you. We will rephrase this in the revised manuscript.*

l.94: Reviewer Subject: Comment on Text
Not sure what you mean here by this, g can take any value between 0 and 1 right? - not sure what these bounds mean and where they come from.
what kind of steps are taking?
*We evaluate the length of the grounding line of each basin as a whole. From the 2D field of access depths, generated through the flood-fill, we take the access depths at the grounding lines and see which values are present. For example, 50% of a basin Y's grounding line is connected to the open ocean at -550m, so d(Y,50%)=-550m. We aim to clarify the concept of g(%) in the revised manuscript, see overall comment above.*

l.101: Reviewer Subject: Comment on Text
similar or the same?

*See above. The basic concept of a flood-fill is the same. The code and used data are different. We only use the bathymetry to infer the access depth and do not extrapolate temperature and salinity estimates into the cavities. We will clarify this.*

**Page: 5**

l. 109f: Reviewer Subject: Comment on Text

according to this definition the ice shelf base would also pass s a calving front

*Will be changed to "the horizontal (map-view) boundary at the surface between floating ice and the ocean" in the revised manuscript for clarity.*

l.130: Reviewer Subject: Highlight

capture?

*Yes, we will change this in the revised manuscript.*

l.131: Reviewer Subject: Comment on Text

which variables are the input?

*We feed temperature and salinity estimates into PICO. We will clarify this in the revised manuscript.*

**Page: 6**

l.141: Reviewer Subject: Underline

a

*Will be changed in the revised manuscript.*

l.142: Reviewer Subject: Cross-Out

*Will be changed in the revised manuscript.*

l.144: Reviewer Subject: Comment on Text: ", we assume that the gateway(s) provide access to a significant amount of the grounded Antarctic Ice Sheet e.g. for dc(g = 50 %, b) in Fig. 2."

why does that need to be an assumption? can't you calculate that? or are you referring to things you omit here such as ocean circulation? Please describe what that assumption entails.

*Yes we did calculate the access depth but we do not simulate the actual ocean inflow. We also do not simulate the ice-dynamical response, such that we have to make assumptions about what portion of the grounding line needs to be accessed by waters at a certain depth to be "significant". This was displayed in Preprint Fig. 3 i.e. what fraction of the basin's grounding line is connected to the ocean at which depth. For visualization purposes we picked g=50%, to reference a value in Preprint Fig.2 or Fig. 5. We aim to clarify the concept of g(%) in the revised manuscript (see overall comment above), and refrain from displaying the values only for an arbitrary value of g=50% for each region.*

**Page: 7**

Figure 2: Reviewer Subject: Sticky Note

What are the straight white lines along on the continental shelves?

*The straight lines stem from the used boundaries of the 19 PICO basins. We will add the basin boundaries as a new legend entry and in the figure description, so that it will be clarified in the revised manuscript.*

Figure 2 (caption): Reviewer Subject: Highlight
maybe visualize it with a contour?
*We plan to leave the reference values of g=50% out of the figure.*

l.170: Reviewer Subject: Highlight
shallower? or narrower?
*This was meant in terms of 'provided access to the grounding line'. We see that the sentence could have been misunderstood and we will rephrase it to "regions with less pronounced gateways" in the revised manuscript.*

**Page: 8**
Figure 3: Reviewer Subject: Sticky Note
for Totten the -370 is 3 boxes, not 2 as highlighted
*Thanks for this hint, will be corrected in the revised manuscript.*

l.171: Reviewer Subject: Highlight
I don't understand what you mean here
*We mean that the difference between T_CSB and T_CF is very pronounced, when only considering a small amount of the grounding line. Will be clarified in the revised manuscript.*

l.171ff: Reviewer Subject: Highlight "If those parts of the grounding line also have the highest grounding line depths, warm water intrusion at depth could cause significant melting in the region."
I am not following here - but either way this sentence sounds more like discussion and not results so perhaps better placed there with some context?
*We propose to rephrase this part to "If those parts also coincide with thick and thus deep-lying ice at the grounding line, access of warm water at depth could cause significantly more melting in the region (i.e. as the pressure melting point of ice decreases with depth)".*

l.173: Reviewer Subject: Cross-Out
*Will be changed in the revised manuscript.*

l.176: Reviewer Subject: Highlight
I don't think you defined what critical T and S are
*We here again apologize for the imprecise handling of our terminology. The term "critical temperature/salinity" was used by us to indicate the temperatures/salinity related to a specific value of dc(g) which was derived from the access depth. Since we discuss the access depths based on grounding line access (=g, in different % of total grounding line in a specific region), we see that the word "critical" might not be appropriate here. We will clarify this in the revised manuscript.*

l.177: Reviewer Subject: Highlight
can you explain why shallow depths are a problem?
*In our initial analysis, some of this region's continental ice is grounded above sea-level and has no access to the (open) ocean, only 20% (displayed in Preprint Figure 3). To be consistent with the ice shelf parts, where PICO parameterizes melting, we will leave out all parts of the grounding ice that are > 0m.*

l. 178: Reviewer Subject: Highlight
both Ross and Drygalski?
*No, for both basin 17 and basin 11. Will be clarified in the revised manuscript.*

**Page: 12**
l. 233f: Reviewer Subject: Comment on Text
How do the numbers change if you use that one instead?
*We will include this in a revised manuscript.*

l.236: Reviewer Subject: Comment on Text
here and elsewhere, see major comment regarding terminology "intrusion"
*Will be clarified in the revised manuscript (see general comment on the terminology above).*

l. 261f: Reviewer Subject: Comment on Text
Can you justify why this is a good assumption rather than just stating that you
are making it?
*In our study we identify the deepest trough (Glomar Challenger) and assume that warm water could be redirected through it to the region's grounding line. This assumption is based on our scientific question for this study: By how much would the melt rates increase around Antarctica if warm water masses off the shelf would intrude onto the continental shelf through the deepest trough / at depth?*

l. 262f: Reviewer Subject: Comment on Text
I don't understand what new and independent you found that you are
comparing to the finding of Tinto et al here.
*Our findings are in line with Tinto et al. that find that the bathymetry constrains sub-ice-shelf ocean circulation, protecting the ice shelf grounding line from moderate changes in global ocean heat content.*

**Page: 13**
l.283ff: Reviewer Subject: Highlight
How well does the model resolve the region, does it have ice shelves and if so
what melt rate change it produces?
*Gómez-Valdivia et al., (https://doi.org/10.1029/2023GL102978) employ the UKESM1 Earth System Model with a relatively coarse resolution (1° ocean model) according to the paper. Their paper focuses strongly on the evolution of the Ross Gyre. As far as we can tell, the paper does not state melt rates and does not give any explanation whether or not interactive ice shelves are included in the model. We can add "employ a global climate model on a relatively coarse resolution" to this part in the revised manuscript for clarity.*

l.293ff: Reviewer Subject: Highlight
Those two studies aren't comparable as one uses small number of point
measurements and the other number is an area average
*Thank you for pointing this out to us. We will correct this in the revised manuscript. As we compare basin wide average we will omit citing the Vaňková et al., 2023 study and rephrase the sentence to: "Rignot et al., 2013 find melt rates at Totten Ice Shelf to be 10.47±0.7 m yr−1".*

l.310: Reviewer Subject: Cross-Out
*Will be changed in the revised manuscript.*

l.312: Reviewer Subject: Comment on Text
"assume that ocean waters in front of the ice shelf serve as valid proxy for water masses that currently drive melting underneath the ice shelf, which is generally valid for cold-mode ice shelves (Silvano et al., 2016)."
can you comment on warm ice shelves also?
*We will reword this sentence and include "which is not true for warm-mode ice-shelves".*

l.313: Reviewer Subject: Cross-Out
"assuming that flow simply follows the "
isobaths
*We assume that the water intruding into the cavity is following the bathymetry at the same and lower depth. Using "Isobaths" would in our view hence not be appropriate here. As stated above, we assume that the warmer/saltier/denser CDW sinks from the shallowest overflow point eventually towards the grounding lines, fills up the cavity basin and replaces the less dense waters below the access depth.*

l.313: Reviewer Subject: Highlight
"Second, we estimate the continental-shelf break temperatures at the same depth, assuming that flow simply follows the bathymetry, and not, e.g., isopycnals (Drijfhout et al., 2013)."
what flow are you talking about actually?barotropic? baroclinic? When speaking about intrusion - it would typically be along ispycnals
*In the mentioned text passage we exactly wanted to acknowledge that warm water intrusions occur along isopycnals, and that we do not account for that due to the simplicity of our approach. Here, our assumption is, that the CDW is not only warmer but also saltier and therefore denser, such that we expect it to sink from the shallowest overflow point on the continental shelf, eventually towards the grounding lines, and filling up the cavity basin, replacing the less dense waters below access depth. We can rephrase the sentence in a revised manuscript for clarity.*

l. 314: Reviewer Subject: Comment on Text
As you admit, ocean circulation is a crucial to sub-ice shelf conditions and melt,
how relevant is your study in light of this assumption?
*Our study aims to provide circum-Antarctic estimates for a potential mode 2 onset. Our estimates only take into account the bathymetric constraints. More sophisticated, high-resolution, coupled ice-ocean modeling is needed to assess the potential and boundary conditions for mode 2 onset in all regions. While we do not include ocean circulation, we here provide a first-order assessment, in case of a transition from mode 1 to mode 2 melting conditions.*

l.317: Reviewer Subject: Comment on Text

"Our study could therefore be improved by considering specific ocean circulation patterns informed by high-resolution ocean models."
what do you mean by our study - can you be specific here about what your
study achieves?

*For instance, high-resolution ocean dynamical models could suggest that access is more likely through the second deepest channel, or that even deeper ocean levels that at access depth should be considered. In a revised manuscript, we will elaborate on this.*

l.323: Reviewer Subject: Comment on Text
"our identified gateways could be an entry point to cross-cut the density barrier in front of the continental shelf (Hirano et al., 2023)"
what do you mean by a a density barrier? the tilted isopycnals? the slope
current? the pycnocline?

*Thank you for pointing out that this sentence needs rewording. We refer to the Antarctic slope current in this part. Will be clarified in the revised manuscript.*

l.324: Reviewer Subject: Highlight
it is density, not temperature alone that is the dynamically important quantity

*Yes agreed, we just wanted to highlight which quantity is more relevant for melting. Will be rephrased in the revised manuscript.*

**Page: 15**
l.330: Reviewer Subject: Highlight
limitations would be more accurate - uncertainty has a specific well defined
meaning

*Yes, we can agree on this; will be changed in the revised manuscript.*

l.332f: Reviewer Subject: Highlight
More than that, the bathymetry at most places is not even well known at this
resolution

*This is true, but as ice-sheet modelers we need to work with what is given, and we aim our analysis to be useful for us/them. A resolution of 500m in the BedMachine Dataset is the current state-of-the-art and incorporates most recent findings/discoveries.*

l.345: Reviewer Subject: Comment on Text
I don't know what you mean here

*The word "altered" was used here as a different word for "changed"/"modified"/"different". Will be changed to "our melt rate estimates could **differ** when using an alternative melt parameterisation" in the revised manuscript.*

l.346: Reviewer Subject: Sticky Note
It would be fair to state explicitly that this particular study did not find good
agreement between PICO and reference coupled model, since you say that
the earlier Favier et al does

*We can take this up in a revised manuscript, but the PICO implementation in that study used a completely different PICO parameter tuning.*

l.350: Reviewer Subject: Comment on Text

I don't know if full ensamble but a few endmembers might be useful to provide some sort of uncertainty
*In a revised manuscript, we aim at providing the obtained basal melt rate estimates using the minimum, best-fit and maximum PICO parameter values from Reese et al., 2023.*

l.352ff: Reviewer Subject: Sticky Note
This whole paragraph is unclear - explain what is being adjusted where, and why it is no longer needed and why it was needed earlier. Or alternatively leave out as this paragraph doesn't seem to be connected with the paper much - sounds more like an outcome of Reese et al 2023 since no new modifications of PICO were introduced in this paper
*Thank you for your feedback. We will rephrase this part in the revised manuscript (see also general comment above).*

l.353: Reviewer Subject: Highlight
formerly meaning when/where?
*It was meaning as "in earlier studies". Will be changed to "A comparison of input temperatures* **used in earlier studies** *with PICO and the temperatures extracted in this study" in the revised manuscript.*

l.354ff: Reviewer Subject: Comment on Text
This statement needs some context for those who don't know what adjusted temperatures are
*Will be clarified / extended on in the revised manuscript.*

l.356ff:  Reviewer Subject: Comment on Text
This should probably be in the methods together with where PICO is introduced
*We agree. Will be changed in the revised manuscript.*

l.364: Reviewer Subject: Highlight
Can you just propose a suitable subdivision in stead of speculating about one?
*Thank you for pointing out this aspect. Yes, considering the access depth and pathways of the major ice-shelf regions (namely FrIS, Amery and George VI) we will propose following new PICO boundaries:*

[Figure]

*Fig. 2: New proposed PICO basin boundary definitions based on Oceanic Gateway analysis in Filchner-Ronne, Amery and Amundsen Sea region.*

*We will provide this updated basin mask on an 8kmx8km grid spacing.*

**Page: 16**
l.371: Reviewer Subject: Highlight
do you mean sectors? since you don't actually analyze properties in cavities?
*Yes, we meant the sectors or basins, we differentiate based on the existing 19 PICO regions. Will be clarified to "in some regions" in the revised manuscript.*

l.378: Reviewer Subject: Highlight
"In some basins, warm water masses accessing g = 30 % of the region's grounding line could be sufficient to reach all *fast flowing ice* to cause significant ice loss, but in others g > 50 % is required."
is it necessarily true that change of thickness of already fast flowing ice produces more retreat than change of thickness of initially slow flowing ice? If not, why is it significant to reach fast flowing ice
*We apologize for any confusion caused regarding this point. Fast ice flow does not directly contribute to sea level rise, but it often coincide with ice stream structures in deep-laying fjords. Reese et al., 2018 (The far reach of ice-shelf thinning in Antarctica. Nature Clim Change 8, 53–57 (2018). https://doi.org/10.1038/s41558-017-0020-x) show that the highest response of grounded ice towards the same amount of thinning is found downstream of fast-flowing ice.*

l.379: Reviewer Subject: Highlight

"For those regions the most vulnerable parts of the grounding line may be located in shallower parts"

what is the exact meaning of vulnerable here?

*We apologize again for our vague wording. In our manuscript, we have argued that regions with deeper access depth are more vulnerable to warm water inflow from the continental-shelf break i.e. have a potential of losing large portions of upstream ice. Warm CDW resides at the continental-shelf break at mid-depth. Shallow parts of the grounding lines are hence less vulnerable. We will correct this passage in the text.*

l.394: Reviewer Subject: Highlight

what you use is to a large extent an extrapolation of observations, isn't it?

*Yes, it is true that, for the ocean dataset, the observations are extrapolated. We evaluate these with the bathymetry data and grounding line positions from the BedMachine dataset. If a newer dataset becomes available one can update our estimates, as we use the ISMIP6 climatology as present-day temperatures. We will clarify this point in the revised manuscript.*

**Page: 17**

l.407: Reviewer Subject: Comment on Text

what do you mean by this?

*It means that this method is not as complicated as running a coupled-ice ocean model. It can be done by running a few python scripts on a high-performing computer. Our analysis can also be done on a simple personal computer. We will extend on the "feasibility" in the revised manuscript.*

---

## Author Comment (AC3)

*We thank the three reviewers for carefully reviewing our manuscript and are happy to strengthen our study according to their suggestions.*
*In addition to our initial reply to RC1, in the following, we provide a general response to the overall and key aspects mentioned in the three reviews together, as well as a point-to-point response to all reviewer comments respectively.*

*Our point-to-point responses to the individual comments are given below in blue and italics compared to the reviewer comments which are given* in black without italic font.

**Author response to key aspects common to all RCs**

*With regards to the comments made on **novelty and relevance** of our study, in the revised manuscript, we try to be more direct about **our goals** and **main motivation**. Namely, we aim to estimate the Antarctic Ice-Sheet's vulnerability to changes in ocean thermal driving. To get a rough estimate of potential changes in thermal forcing, we identify features from continental shelf bathymetry that are similar to Filchner Trough which could provide warm water masses access to the grounding line once the cavity switches into a "warm" state. Furthermore, we calculate the connectedness of Antarctic grounding lines to the open ocean. Based on this, we derive an upper-bound of temperature (and thus basal melt rate) changes in ice shelf cavities around Antarctica.*

*We strengthen the relevance of our study by additionally providing the derived access depths data for a 500m x 500m grid spacing in the supporting material as well as the code to update the access depths when new data becomes available. In a revised manuscript, we will further extend the section on how our findings may influence PICO tuning in future studies and how this relates to the current settings and other melt parameterizations. To this end, we will now also include an updated delineation of basin boundaries for PICO in the ocean based on our findings (taking up the suggestion made by RC1).*
*As suggested by RC2, we will also revise the title of our manuscript to strengthen the main message i.e. that we look into bathymetric constrained warm mode melt estimates derived from analyzing Oceanic Gateways in Antarctica.*
*As suggested by RC2, we will overall stream-line our manuscript to fit our target audience, namely stand-alone ice-sheet modelers (using PICO) and those working on the coupling to coarse grid ocean models, as our study provides bathymetry-informed estimates for temperature and salinity as used in basal melt parameterizations such as PICO.*

*Concerning the comments made towards our **terminology and methodology,** we will include a more in-depth and more comprehensive explanation of the concept of access depths, through a revised Figure 1 and by rephrasing 1) how access depths are defined and derived, 2) how we define critical access and 3) by showcasing how we conducted the analysis, providing estimates on the computational cost and the numerical code of the flood-fill.*
*In the revised manuscript we will also add further explanations how our approach relates to the **ISMIP6 protocol,** i.e. that we foremost analyze bathymetry and subsequent grounding line connectedness and that our analysis can be adapted to whatever ocean dataset there is available.*

*To reduce complexity, we will rework the use of **our parameter "g"** that we introduced to describe how much percentage of a basin grounding line can be accessed by open ocean*

*water masses at depth. When revising our quantitative results, we will now additionally exclude all parts of the ice sheet where ice is grounded >0m (as correctly pointed out by RC2). We will propose a new Figure visualizing the distribution of access depth at the grounding line (formerly Figure 3).*

*Generally, **more than one ice shelf** can be included in one PICO basin. We will make sure to discuss related caveats in the revised manuscript (which was especially raised in RC2). We will clarify the use and definition of a **prominent gateway**, that we had initially defined as "one or several deep troughs that provide access to most of a region's grounding line". Here, "most of a region's grounding line" had referred to 10% or more of the grounding line accessed at one distinct access depth level in our preprint and was highlighted by the magenta boxes in Fig. 3. In regions where we do not see this feature, but a rather gradual increase in access with grounding line portion/fraction, we actually cannot state that an oceanic gateway is present.*

*In the revised manuscript, we will update our **temperature estimates** and **change the overall narrative on our two scenarios.** We will refrain from speaking about "warm-water intrusions"; instead, we will give an "upper-bound" estimate, as we rather consider a macro-scale/basin wide transition in melt mode associated with prevalent access of ocean water from off-shore. To this end we will also assess how the $T\_CSB$ estimates change when taking the maximum (instead of the mean) temperatures along the CSB. This would follow our refined intention of a bathymetry-derived upper bound to melt rate changes. Further, we will clarify the paragraph in which we define the temperature estimates and add further discussion on how ocean temperatures may change (e.g. CDW->mCDW) when intruding onto the continental shelf (what we do not resolve). While we take the temperature estimates $T\_CSB$ as proxy for mode 2 melting conditions, we will state in the revised manuscript more clearly that $T\_CF$ is representative for mode 1 melting (after Jacobs et al., 1992). To this end, we will consider the bottom temperatures at the calving front, instead of averaging them at the overflow / access depth of that basin.*

*In order to resolve the discrepancy of our estimated "present-day" melt rates to observations (as pointed out by all three reviewers), we will change our methodology for the melt estimates as follows:*

*Our used PICO parameters from Reese et al., 2023 were tuned to represent bulk present-day melt rates as well as to match the melt sensitivity at Filchner-Ronne (cold based) and Amundsen Sea ice shelves (warm based). In the tuning process, the input temperatures from Schmidtko et al., 2014 for each basin were adjusted so that melt rates, as well as melt rate sensitivities, would be in line with observations. These necessary, yet to a point unphysical, temperature corrections can hence be seen as an additional factor in the tuning. To be consistent with the tuned parameters, we will propose to take the forcing field from Reese et al. 2023 as present-day baseline temperatures. For estimating "upper bound" estimates of bathymetric-constrained warm mode onset, we will then add the difference of $T\_CSB$ minus $T\_CF$ (both derived from ISMIP6 dataset) to the existing forcing field. This follows the same "anomaly idea" taken in Kreuzer et al. (in discussion, [doi.org/10.5194/egusphere-2023-2737](doi.org/10.5194/egusphere-2023-2737)). We will make sure to expand the explanation of these temperature adjustments within the PICO tuning process in our method section.*

*Once we have new estimates we will include a more thorough comparison of our temperature as well as basal melt rate estimates to findings from previous literature, specifically in the key regions that the reviewers mentioned e.g. at Ross Ice Shelf and in the Amery region (both mentioned in RC1 on page 12), the gateways we find in the Amundsen Sea (esp. Abbot Cosgrove Trough, mentioned in RC3) as well as our temperature estimates in this region (mentioned in RC2), and subsequent melt rate estimates.*

*We will clarify the sign convention of z vs. depth and align it with commonly used definitions (in reply to RC1) and we will provide melt estimates in Gt/yr (in reply to RC2). We will further rework our Figures as suggested by all reviewers.*

*We will further gratefully take up the suggested language changes to specific wording within the text (see respective point-to-point response below).*

*For the specific comments made by the individual reviewers, please consult the respective point-to-point responses.*

**Response to RC2**

Review of: Oceanic gateways to Antarctic grounding lines – Impact of critical access depths on sub-shelf melt
Lena Nicola et al.
Date: 26 Feb 2024
Assessment: Major revision
Reviewed by: Erwin Lambert

The authors have assessed oceanic gateways for deep water access to the grounding lines of 19 Antarctic basins. Ice shelf melt rates were computed using the PICO parameterisation based on two input forcings: T and S at the calving front, and at the continental shelf break. The melt difference between these two forcings was interpreted as an upper limit for the potential melt increase due to a change in oceanic circulation on the continental shelf (i.e. an onset of an 'optimal mode 2' system).
The idealised approach to representing deep water access and conversion to melt rates is an interesting way to achieve a 'first order' estimate for potential melt increases. Overall the manuscript was an easy read due to a good writing style. I do have a number of concerns though, particularly regarding the methodology, interpretation, and relevance, that in my eyes should be resolved before considering publication. Hence, I recommend a major revision of the manuscript. Below I list my major and minor comments.

*Dear Erwin Lambert,*
*Thank you very much for your detailed review of our manuscript and your feedback.*
*In addition to our general comment to all reviewers, please find our detailed point-by-point responses (written in blue and italics)* to your comments (in black) *below.*

**Major comments**

1. Motivation / scope. At first (partly based on the title), I thought the main aim was to define the oceanic gateways, and was puzzled by the focus on CF and CSD

temperatures. Only when reaching section 3.2, it became clear to me that computing the difference in melt rates (T_CSD – T_CF) was the ultimate goal to quantify the impact of a hypothetical 'mode 2' onset. To better clarify this, I suggest:

    a. Revising the title, with a focus on, in the author's own words, quantifying the melt increase due to 'mode 2' onset, as constrained by bathymetry.

*When revising the manuscript, we will revise the title (see general comment above).*

    b. Throughout introduction and methods, clearly write towards this goal. In particular in section 2.2, this should be very explicit.

*We will stream-line our manuscript accordingly.*

    c. Include a clear narrative how a switch from CF to CSD forcing may occur (e.g. based on 3D modelling) and why, according to the authors, their approach is the optimal way to quantify the 'first order' impact.

*We aim to include more details to this in the revised manuscript.*

2. Possible errors. I detected two results which the authors should critically reassess, as I believe them to result from errors in the methodology.

    a. In the caption of Fig 3, the authors state that 'parts of the grounding lines are situated above 0 m'. This should not be possible and makes me wonder whether cliff faces (boundary between grounded ice and open ocean) are included. In some regions (5, 10, 11, 17), this appears to be the case for >20% of the assessed grounding line.

*Thank you for pointing this out to us. This will be addressed and changed. In our initial analysis we considered the contiguous continental ice mass and its contour as the "grounding line". In the revised manuscript, we will refine the definition of grounding line when deriving the input for PICO and only consider the parts of the ice sheets where melting is applied (hence the correct definition of grounding line in this context here - the triple point of bedrock, ice and ocean inside ice-shelf cavities).*

    b. Present-day temperature at CF in the Amundsen Sea was determined at -0.88 C. This is substantially colder than the +1.1 C observed by Dutrieux et al (2014) at the Pine Island calving front. Note that this is very close to the +1.26 C the authors find at CSD, implying a present day state which is very close to the 'optimal mode 2' state. I also recommend that the authors double-check their tuning parameters, as 20.5 m/yr with a deep water temperature of -0.88 C appears very high to me.

*When revising our manuscript, we will update our temperature and melt rate estimates to be more in line with previously tuned PICO parameters from Reese et al., 2023. We plan to change to an anomaly approach, as laid out in the general comment above. When discussing our temperature estimates we will, especially in the Amundsen Sea region, discuss this discrepancy between the basin average temperature T_CF and the locally observed temperatures that exist to date, especially within those basins incorporating more than one ice shelf. As we use the ISMIP6 dataset for our analysis, we depend on the, in part newer, observations that have been incorporated when creating the forcing fields. As said, we will make sure to include a thorough discussion of this aspect in the revised manuscript.*

3. Methodology. A few methodological choices have been made which I believe require more scrutiny / quantitative assessment / reconsideration.

    a. By design, PICO takes one value for the deep water temperature and applies this to the first box near the grounding line. Hence, even in cases where g = 10%, the associated temperature at this depth is applied to 100% of the

> grounding line. The authors should be transparent about this limitation, and ideally give a quantitative assessment of its impact on their results.

*It is correct that the associated temperature at access depth is applied to the entire grounding line box. Yet, the melt rates in each ice shelf cell of this box consider the cell-specific pressure melting point and therefore the depth of each cell. In the revised manuscript, we will try to more clearly state the limitation and discuss the implications. As laid out above in the general comment, we will refrain from the use of the full range of grounding line access (g) and focus on the deepest access depth we find in the different Antarctic regions.*

> b. The results are highly dependent on the choice of PICO, making it difficult to interpret the results. I would recommend a comparison to the 'quadratic parameterisation' (Favier et al 2019), also used in ISMIP6. This parameterisation can be applied uniformly (as PICO), and regionally (accounting for the limited access to the grounding line). Hence this would allow the authors to quantitatively reflect on point 3a above, and quantitatively assess the impact of the choice for PICO.

*In the revised manuscript we will additionally provide basal melt rate estimates using the min, best-fit and max parameters presented in Reese et al 2023 to address the impact of PICO parameters on our estimates. Using a quadratic melt parameterization is for sure helpful to address the uncertainty arising from (the choice of) PICO, but we see a full comparison between different existing parameterizations beyond the focus of our study, as e.g. the ISMIP6 quadratic parameterization is not implemented in PISM to date yet.*

> c. It is unclear how the authors treat individual cavities. For example, Totten/Moscow University. The authors focus on a trough providing access to the Totten cavity. Does their method implicitly assume that the same oceanic access exists to the Moscow University cavity? If so, the authors should elaborate on this and estimate the quantitative impact of this implicit assumption. If not, the authors should explain how they deal with separated cavities within the Zwally regions.

*In our study we follow the common practice of using one temperature and salinity input per PICO basin. In our initial analysis we have used the 19 PICO basins from Reese et al., 2018. In the revised manuscript, we will put a stronger focus on this limitation of averaging over basins that cover more than one ice shelf and make sure to provide more individual results where appropriate e.g. at Totten/Moscow University ice shelf. As mentioned above, we will also propose new basin boundary delineations in the ocean for the future use of PICO.*

4. Relevance. It is always tricky to explain the relevance of a relatively idealised study (trust me, I've been there!). I have some suggestions though to enhance the relevance of this study.
   a. Coming back to the motivation, the authors should have a clear take home message (in abstract and conclusions).

*We will include our take-home messages, e.g. that only some regions have pronounced oceanic gateway structures that could potentially redirect warm water masses off the continental shelf into the ice-shelf cavities. And if so, melt rates would increase by a certain Gt/yr per basin, assuming that these warm water masses remain unmodified on that pathway. Another valuable outcome of our study is that the PICO boundaries in the ocean could be re-adjusted according to bathymetric features, permitting the connectedness of the grounding lines to the open ocean.*

Also, it is a bit unclear who the target audience is: the oceanographic community (providing guidance for further research on circulation changes) or the ice sheet community (providing an upper limit to melt increases)? The manuscript would benefit from having a clear target audience and a clear message to this audience.

*With our study we had the ice-sheet modeling community in mind, namely stand-alone ice-sheet modelers (for instance those using PICO) and those working on the coupling to coarse resolution ocean models.*

b. Where possible, the results should be compared quantitatively to more realistic studies. For example, the authors cite previous studies on Filchner-Ronne (e.g., Naughten et al 2021), but do not compare their quantitative results to those.

Throughout section 3.3, I suggest the authors maximise the quantitative comparison to previous numerical modelling studies to place their results in perspective. In addition, this quantitative comparison should be reflected on in the discussion.

*As mentioned above in the general comment, once we have our new estimates we will include a more thorough quantitative comparison and discussion of our temperature- as well as basal melt rate estimates to findings from previous literature, if available. Thank you for pointing out this short-coming to us.*

c. Also, the authors should mention and reflect on the discrepancy between present-day melt rates (Fig 5a) and observed ones (Fig 5c), and how this impacts their results. In some regions, the relative difference between observations and T_CF is larger than that between T_CSD and T_CF. Can the authors explain this, and convince me and other readers that this does not impact the trustworthiness of their results and conclusions?

*We hope the discrepancy between present-day melt rates and observed ones can be addressed by our change in methodology as laid out in the general comment above.*

*In the initial manuscript, we have used the PICO parameters from Reese et al 2023 that were tuned to represent bulk present-day melt rates as well as to match the melt sensitivity at Filchner-Ronne (cold based) and Amundsen Sea ice shelves (warm based). In the tuning process (as described in Reese et al. 2023 in detail), present-day input temperatures were adjusted i.e. that the needed temperature correction was another tuning parameter to match melt rates and sensitivities. Using an un-adjusted temperature field, like the T_CF values in our initial manuscript, thus produced large discrepancy to present-day melt rates. This was not clear to us when preparing our initial manuscript and we plan to correct for that when revising.*

*To be in line with the previously tuned PICO parameter, we have therefore decided to take the adjusted temperatures from the tuning process (cf. Reese et al., 2023) as present-day baseline temperatures in the revised manuscript. For estimating the effect of the mode 2 onset, we will thus change to an anomaly approach. For mode 1-representative temperatures from the ISMIP6 dataset, we will extract the bottom-most temperatures at the calving front, T_CF. For mode 2-representative temperatures we will, as done in our initial manuscript, extract temperatures near the continental-shelf break at the determined overflow or access depth. To derive the warm mode onset/"upper bound" estimates, we will then add the difference of the*

*two estimates onto the tuned forcing field from Reese et al., 2023 and see how melt rates will change if we have a large-scale inflow of warm CDW into the ice-shelf cavity.*

    d. As the authors state in their introduction, they aim for a 'first order assessment of the maximum changes in temperature and melt'. To interpret this maximum, the reader requires some estimate of the magnitude of uncertainties associated with made changes. For example, how important is it that off-shelf temperatures are assumed to be constant regardless of the circulation change? In fact, the authors provide a narrative for a thermocline shoaling to provide CDW access. How significant is the assumption of constant off-shelf hydrography? The same holds for assumptions like fixed cavity geometries, and the choice for PICO (see points above). The question that remains in my head: could the author's assessed 'maximum' melt increases be twice as high due to unconsidered processes, or is it a reasonable estimate?

*We see our study as a sensitivity analysis to gauge the effect of the presence of oceanic gateways to Antarctic grounding lines, thus how melt rates change, if a transition from mode 1 to mode 2 occurs, while we explicitly take into account connecting features in the bathymetry around Antarctica. When revising our temperature estimates, we will discuss the uncertainty related to that by, for instance, not only taking the mean along the continental-shelf break, but also considering a wider perimeter in the open ocean and evaluate where the maximum temperature will lie. For sure, this is just an idealized analysis, i.e. a thought-experiment. We know that for precise projections of potential future regime shifts in the Antarctic ice-shelf regions, more sophisticated approaches are needed. These approaches at best have a coupled ice-ocean-atmosphere representation, with interactive ice sheets and ice shelves at high resolution in space and time.*

    e. The authors focus on g=50% in their figures and most quantifications. What is the logic behind this? I would assume that access to the deepest grounding line parts is most important, and **would think that g=10% may perhaps be more relevant**. Does it make sense to have a fixed g for all regions?
Can the discrepancies between present-day melt rates and observations (Fig 5) be (partly) explained by the choice of g? And can a relevant value for g per region be determined from this comparison to observations?
How should future research treat these values? Stick to 50%, or optimise it per region?
Dedicating one or two paragraphs to this in the discussion would significantly enhance the relevance of this work.

*These are very important points and we see that our concept of g(%) has been confusing or arbitrary. In order to reduce complexity, initially we wanted to discuss the results for each basin using a certain access depth and we picked d(50%). As laid out above we will change this approach and reduce the g(%) dimension to the deepest depth that provides access to the grounding lines. We will make sure to include an adequate discussion of this aspect in the revised manuscript.*

**Minor points**

l. 10. the 200-fold larger melt rate is highly dependent on the uncertain reference state.The authors should pick a more relevant metric for their abstract, such as the total increase in BMB (Gt/yr).
*We gladly take up this suggestion. Will be changed in the revised manuscript.*

l. 16. The concept 'mode 1 and 3' is used, but not really explained. Either stick to more generally known concepts, or give a brief explanation here if it's important.
*Thank you for pointing this out to us. We will rephrase this part in the revised manuscript.*

l. 25. 'tens of metres per year'. Regionally yes, ice-shelf average, this is only possibly the case for Thwaites (and perhaps some tiny ice shelves). Revise this statement and provide a reference.
*Thank you. Will be rephrased in the revised manuscript.*

l. 30. Here the transition to a warm cavity is described, without mentioning it explicitly. As this is a central aspect in this study, the authors should be more explicit here that they are talking about a qualitative change in hydrography and not a smooth warming.
*This point will be taken up and strengthened in the revised manuscript.*

l. 52. Provide a reference for these statements. A highly biased suggestion from my side would be Lambert et al. (2023).
*This is definitely a worthwhile reference and will be added in the revised manuscript.*

l. 55. "For instance, Wouters et al. (2015) find a strong link between surface-lowering and an increase in the dynamical ice loss in the Southern Antarctic Peninsula since around 2009." Is this related to near-grounding line melt? If so, mention explicitly, if not, remove this statement.
*Yes, we believe it is related to near-grounding line melt, as Wouters et al. state thinning rates near the grounding line down to −4 m/year, with the average observed elevation lowering rate at −0.42 m/year. We will include this detail in the revised manuscript.*

l. 80. (and other places). I don't think 'diagnose' is the correct verb. Replace it with 'parameterise' or something equivalent.
*Thank you for pointing this out to us. We will be more coherent in our wording, as we used "diagnose" to refer to the melt rates given a temperature and salinity estimate to PICO (in comparison to calculate transient changes in melting). We will change it to "compute".*

l. 82. Does this dataset include the latest version of IBCSO? If so, reference this explicitly, as Bedmachine uses external sources for its bathymetry. If it's not the latest version, mention this explicitly in the methods/discussion.
*The BedMachine v3 uses IBCSO v2 for ocean bathymetry according to https://nsidc.org/data/nsidc-0756/versions/3 from which we have obtained the dataset. We will add this detail in the revised manuscript.*

l. 84. I don't understand this sentence 'The grounding lines…'. Please revise.

*We here want to give a definition on what we considered as "grounding line" in our analysis i.e. at what contour we evaluate our 2D field of access depths. This touches on your point 2a). We will revise this sentence.*

Fig. 1. The straight grounding line is odd, as it does not follow the 'triple point' between ice, bedrock, and ocean. It should deepen in the trough.
*Thank you. We will take up this point when revising Figure 1.*

l. 98. The access depth is implicitly determined in ISMIP6 as well. Explain clearly what the added benefit is of your methodology in reference to ISMIP6. This difference/overlap is a bit unclear to me.
*Our study extends the work by the ISMIP6 focus group who created the ocean climatology dataset, since our approach takes into account the depth of the grounding line to derive the access depths, which is then used to determine ocean temperatures, salinities and ultimately melt rates with PICO. We hope to clarify this point in the revised manuscript.*

l. 110. The equations and text here do not make the whole methodology very clear (to me..). Consider visualising this, either in Fig 1 or in a new schematic figure, so the reader fully understands what's going on. I'm strongly in favour of a new (schematic) figure which also illustrates the parameter g.
*Thank you for this feedback. We will include a new version of Figure 1 in the revised manuscript (see general comment above).*

l. 131. 'Input is based…' Does this refer to Reese et al '23? If so, mention explicitly, if not, explain why you deviate from the ISMIP6-based forcing here.
*Thank you for highlighting this point. We will change it to "The input (T,S) in Reese et al. (2023) is based on temperature and salinity observations" in the revised manuscript, as it refers to the input in the tuning process described in Reese et al., 2023.*

l. 134. Most ice sheet models have PICO included, so I think this is an irrelevant statement (which causes more confusion than clarifying anything). So rather remove it.
*We will change it to "PICO was first implemented in PISM….and has been meanwhile used in many other ice sheet models" as we, technically speaking, use the PICO version within PISM, run on the BedMachine geometry (using topography, ice thickness, mask etc) we found it necessary to mention PISM.*

l. 159. First $T_{CF}$ is 'generally lower' than $T_{CSB}$, in the next sentence, $T_{CSB}$ is 'much warmer than $T_{CF}$. Align these two statements.
*Will be clarified in the revised manuscript.*

Fig 2. For the bars, T=0 is used as a reference, which is a bit arbitrary. I'd use freezing temperature as a reference for visualisation (so that the bar heights reflect the Thermal Forcing).
*While we acknowledge that providing the temperatures relative to the pressure-melting point would directly show the thermal driving, in PICO, the freezing point is evaluated for each grid cell depending on its depth, such that we cannot define a basin-wide freezing point. We therefore would stay with providing the absolute temperatures for each basin.*

l. 172. 'highest grounding line depths' -> 'shallowest/deepest grounding line depths'
*Will be changed to "deepest grounding line depths" in the revised manuscript (see also our general comment above on the sign convention on z).*

l. 218. Compare 4.65 m/yr to numerical modeling studies.
*When updating our melt rate estimates we will update this section in the revised manuscript.*

l. 233. Is this other pathway deeper? How is it (or can it be) relevant to your study?
*Willams et al., 2016 indicate an outflow of Dense Shelf Water (DSW) through Prydz Channel and an intrusion of mCDW over Four Ladies Bank, which is much shallower than Prydz Channel. Here our core assumption that CDW always takes the deepest entry / gateway towards the ice shelf is challenged. We will rephrase this part in the revised manuscript.*

l. 318. *"Our study could therefore be improved by considering specific ocean circulation patterns informed by high-resolution ocean models."*
These kinds of sentences are fine, but compare to other studies where possible.
*Thank you for this remark; we can include some references to the high-resolution ocean modeling studies in the revised manuscript at this point and where else appropriate.*

l. 328. Is it possible to quantify the difference between CDW and mCDW? For example looking at Amundsen Sea (observed $T_{CF}$ = 1.1; $T_{CSB}$ = 1.26). Does this example mean that the difference is negligible?
*When updating our estimates, also using the revised PICO boundaries, we can add a short assessment / comparison of our estimates ($T_{CSB}$) to the observed temperatures in ice-shelf regions where we know that mCDW is present at depth.*

l. 335. Good point regarding the trough width. What ocean dynamics control this minimum width? Are there specific troughs you highlighted in this study which are very narrow where this may have an impact? How can you/future researchers incorporate this concretely?
*We will have a look at the trough widths, also in the smaller Antarctic regions that we find feature oceanic gateways in our analysis. We can include this aspect to a greater depth in our discussion thereafter. The identified oceanic gateways in our analysis are based on the native grid resolution of BedMachine, so that troughs/depressions, if existing, are at least 500 m wide. For our purpose, we assume that water can flow through the gateways i.e. a channel of 500m width is wide enough to fill the water with warm water offshore during a sustained inflow.*

l. 397. MISI is new info, which should not appear in the conclusions. If it's relevant, include it in the intro/discussion. Conclusion should only contain previously presented information that is specific to your study.
*Thank you for pointing this out to us. We do not consider the dynamical response of the ice sheet in our study, but MISI provides a motivation to check for melt perturbations. We will move this part in the revised manuscript.*

l. 409. Again, geoengineering is new info. Put this in the discussion if you want to include it; stick to your own work and its implications/relevance in the conclusions.
*In order to stream-line the revised manuscript, we propose to leave this part out.*

**References:**
Dutrieux et al. 2014. https://doi.org/10.1126/science.1244341
Naughten et al. 2021. https://doi.org/10.1038/s41467-021-22259-0
Lambert et al. 2023. https://doi.org/10.5194/tc-17-3203-2023

---

## Author Comment (AC4)

*We thank the three reviewers for carefully reviewing our manuscript and are happy to strengthen our study according to their suggestions.*
*In addition to our initial reply to RC1, in the following, we provide a general response to the overall and key aspects mentioned in the three reviews together, as well as a point-to-point response to all reviewer comments respectively.*

*Our point-to-point responses to the individual comments are given below in blue and italics compared to the reviewer comments which are given* in black without italic font.

**Author response to key aspects common to all RCs**

*With regards to the comments made on **novelty and relevance** of our study, in the revised manuscript, we try to be more direct about **our goals** and **main motivation**. Namely, we aim to estimate the Antarctic Ice-Sheet's vulnerability to changes in ocean thermal driving. To get a rough estimate of potential changes in thermal forcing, we identify features from continental shelf bathymetry that are similar to Filchner Trough which could provide warm water masses access to the grounding line once the cavity switches into a "warm" state. Furthermore, we calculate the connectedness of Antarctic grounding lines to the open ocean. Based on this, we derive an upper-bound of temperature (and thus basal melt rate) changes in ice shelf cavities around Antarctica.*

*We strengthen the relevance of our study by additionally providing the derived access depths data for a 500m x 500m grid spacing in the supporting material as well as the code to update the access depths when new data becomes available. In a revised manuscript, we will further extend the section on how our findings may influence PICO tuning in future studies and how this relates to the current settings and other melt parameterizations. To this end, we will now also include an updated delineation of basin boundaries for PICO in the ocean based on our findings (taking up the suggestion made by RC1).*
*As suggested by RC2, we will also revise the title of our manuscript to strengthen the main message i.e. that we look into bathymetric constrained warm mode melt estimates derived from analyzing Oceanic Gateways in Antarctica.*
*As suggested by RC2, we will overall stream-line our manuscript to fit our target audience, namely stand-alone ice-sheet modelers (using PICO) and those working on the coupling to coarse grid ocean models, as our study provides bathymetry-informed estimates for temperature and salinity as used in basal melt parameterizations such as PICO.*

*Concerning the comments made towards our **terminology and methodology,** we will include a more in-depth and more comprehensive explanation of the concept of access depths, through a revised Figure 1 and by rephrasing 1) how access depths are defined and derived, 2) how we define critical access and 3) by showcasing how we conducted the analysis, providing estimates on the computational cost and the numerical code of the flood-fill.*
*In the revised manuscript we will also add further explanations how our approach relates to the **ISMIP6 protocol,** i.e. that we foremost analyze bathymetry and subsequent grounding line connectedness and that our analysis can be adapted to whatever ocean dataset there is available.*

*To reduce complexity, we will rework the use of **our parameter "g"** that we introduced to describe how much percentage of a basin grounding line can be accessed by open ocean*

*water masses at depth. When revising our quantitative results, we will now additionally exclude all parts of the ice sheet where ice is grounded >0m (as correctly pointed out by RC2). We will propose a new Figure visualizing the distribution of access depth at the grounding line (formerly Figure 3).*

*Generally, **more than one ice shelf** can be included in one PICO basin. We will make sure to discuss related caveats in the revised manuscript (which was especially raised in RC2). We will clarify the use and definition of a **prominent gateway**, that we had initially defined as "one or several deep troughs that provide access to most of a region's grounding line". Here, "most of a region's grounding line" had referred to 10% or more of the grounding line accessed at one distinct access depth level in our preprint and was highlighted by the magenta boxes in Fig. 3. In regions where we do not see this feature, but a rather gradual increase in access with grounding line portion/fraction, we actually cannot state that an oceanic gateway is present.*

*In the revised manuscript, we will update our **temperature estimates** and **change the overall narrative on our two scenarios.** We will refrain from speaking about "warm-water intrusions"; instead, we will give an "upper-bound" estimate, as we rather consider a macro-scale/basin wide transition in melt mode associated with prevalent access of ocean water from off-shore. To this end we will also assess how the T_CSB estimates change when taking the maximum (instead of the mean) temperatures along the CSB. This would follow our refined intention of a bathymetry-derived upper bound to melt rate changes. Further, we will clarify the paragraph in which we define the temperature estimates and add further discussion on how ocean temperatures may change (e.g. CDW->mCDW) when intruding onto the continental shelf (what we do not resolve). While we take the temperature estimates T_CSB as proxy for mode 2 melting conditions, we will state in the revised manuscript more clearly that T_CF is representative for mode 1 melting (after Jacobs et al., 1992). To this end, we will consider the bottom temperatures at the calving front, instead of averaging them at the overflow / access depth of that basin.*

*In order to resolve the discrepancy of our estimated "present-day" melt rates to observations (as pointed out by all three reviewers), we will change our methodology for the melt estimates as follows:*

*Our used PICO parameters from Reese et al., 2023 were tuned to represent bulk present-day melt rates as well as to match the melt sensitivity at Filchner-Ronne (cold based) and Amundsen Sea ice shelves (warm based). In the tuning process, the input temperatures from Schmidtko et al., 2014 for each basin were adjusted so that melt rates, as well as melt rate sensitivities, would be in line with observations. These necessary, yet to a point unphysical, temperature corrections can hence be seen as an additional factor in the tuning. To be consistent with the tuned parameters, we will propose to take the forcing field from Reese et al. 2023 as present-day baseline temperatures. For estimating "upper bound" estimates of bathymetric-constrained warm mode onset, we will then add the difference of T_CSB minus T_CF (both derived from ISMIP6 dataset) to the existing forcing field. This follows the same "anomaly idea" taken in Kreuzer et al. (in discussion, [doi.org/10.5194/egusphere-2023-2737](doi.org/10.5194/egusphere-2023-2737)). We will make sure to expand the explanation of these temperature adjustments within the PICO tuning process in our method section.*

*Once we have new estimates we will include a more thorough comparison of our temperature as well as basal melt rate estimates to findings from previous literature, specifically in the key regions that the reviewers mentioned e.g. at Ross Ice Shelf and in the Amery region (both mentioned in RC1 on page 12), the gateways we find in the Amundsen Sea (esp. Abbot Cosgrove Trough, mentioned in RC3) as well as our temperature estimates in this region (mentioned in RC2), and subsequent melt rate estimates.*

*We will clarify the sign convention of z vs. depth and align it with commonly used definitions (in reply to RC1) and we will provide melt estimates in Gt/yr (in reply to RC2). We will further rework our Figures as suggested by all reviewers.*

*We will further gratefully take up the suggested language changes to specific wording within the text (see respective point-to-point response below).*

*For the specific comments made by the individual reviewers, please consult the respective point-to-point responses.*

**Response to RC3**

**General comments**

Nicola and coauthors identify key oceanic pathways in the major Antarctic glacier basins and estimate current and future melt rates based on the present water properties, assuming that warm waters from the shelf break reach the grounding lines. I found it interesting to see the range of access depths and ocean water properties around the Antarctic Ice Sheet in one simple figure, and it is interesting to quantify the upper limit of future melting assuming warm water intrusion through these pathways. The data were clearly visualized, making it easy to synthesize the wealth of information presented. The analyses and profiles shown for the major ice shelves were especially nice to see and will make this study of interest to a range of groups who study various components of the Antarctic ice/ocean system.

I do not have any major concerns about the approach or the conclusions, as the caveats associated with this methodology are clearly noted in the discussion. I have several comments mostly relating to the presentation of the study, so I recommend this study for publication after minor revisions. I hope the authors find these comments helpful.

*Dear Anonymous Reviewer 3,*
*Thank you very much for your review of our manuscript and your feedback. We will revise our manuscript keeping your suggestions close in mind. In addition to the general comment to all reviewers, please find our detailed point-by-point responses (written in blue and italics)* to your comments (in black) *below.*

**Specific comments**

1. The motivation of the study could be more clearly laid out. Ocean gateways are important for ice shelf melting, but why does this analysis help the scientific community? Rather than giving a lengthy overview of why troughs and gateways are important and then explaining the approach, I suggest making a more concise overview of the importance, state (more clearly) what the critical uncertainties are, what the approach is, and what the hypothesis is.

*Thank you for pointing this out to us. We will propose a more stream-lined introduction in the revised manuscript.*

2. What are the criteria for classifying something as a 'prominent gateway'? In L145 the authors state that "large portions" of the GLs need to be reached, but how was this determined?

*We apologize for any confusion regarding this definition. We will clarify this in the revised manuscript (see general comment above).*

3. Section 3.3.4 (Amundsen Sea) – the main gateway is identified as the Abbot Cosgrove Trough. Is this distinct from the 'Eastern Trough' commonly referred to in studies about the Amundsen Sea (e.g., Dutrieux et al., 2014 already cited in this study)? Does it feed into the PIG-Thwaites Trough shown in Fig 9a, or is it separate? It would be helpful to clarify which of the oceanic gateways identified are/aren't in agreement with the main pathways identified in previous studies.

*We appreciate you bringing this to our attention. We will elaborate on this in the revised manuscript.*

4. The mismatch in melt rates between the approach in this study and the melt rates of Adusumilli need to be addressed, as that is critical for gauging how reliable the estimates of future melt rates are. Some regions do better than others, perhaps due to different reasons, so I'd suggest explaining this for each of the major ice shelf regions.

*Thank you for this suggestion, when revising our basal melt rate estimates (see general comment above) we will make sure to add a more in-depth discussion to it in the revised manuscript.*

5. I found Section 3.3 rather hard to read. Rather than explaining the results and then reviewing various relevant literature, I'd suggest reviewing the findings from the literature and then presenting the new results within that context. It feels very scattered in its present state.

*Thank you for this feedback. We will restructure this section in the revised manuscript.*

Other suggestions (to improve the presentation, not necessary for publication in my opinion):

1. In Figure 2, it would be nice to see the temperature relative to the pressure melting point.

*We agree that stating the temperatures relative to the pressure-melting point would be insightful, as it would show the thermal driving. In PICO, the freezing point is evaluated for*

*each grid cell depending on its depth, such that we cannot define a basin-wide freezing point. We therefore would stay with providing the absolute temperatures.*

2.  Several figures likely need larger text to for readers to see the numbers in each of the small boxes – probably fine for PDFs, but not printing. Perhaps the numbers are not critical for understanding the figures, as the colors also reflect the values.

*When possible, we will increase the fontsize of the respective figures in the revised manuscript.*

---

## Referee Report (RR1)

Second review of:

Bathymetry-constrained warm-mode melt estimates derived from analysing Oceanic Gateways in Antarctica

Lena Nicola et al.

Date: 1 Oct 2024
Assessment: Minor revision
Reviewed by: Erwin Lambert

The authors have put in a great effort to address my concerns raised in the first review round. The possible errors have been resolved, the methodology has become more logical and several metrics are now presented in a clearer and more concise way. Additionally, the authors have modified their figures to address several concerns.

I do, however, still have some concerns regarding the interpretation and writing which I believe should be improved before I can recommend this manuscript for publication. Although these concerns can be resolved by rewriting solely, without the need for new analysis, I still recommend a major revision. This is because I believe the depth of interpretation and coherence requires a considerable effort to raise the level of the manuscript to the level needed for publication in The Cryosphere. That being said, I am confident that the authors can reach this level based on the current state of the work and the suggestions below.

Major comments:
1. Coherence. At points, the manuscript reads as separate individual sections/paragraphs without a strong coherence. This makes it difficult to interpret the results. In particular:
    a. The authors provide a quantitative distribution of access depths (Fig 4). These distributions are referred to in the main results section (Sec 3.3). However, the impact of these distributions is not tied to the results in terms of temperature/melt changes. In fact, I believe through the use of PICO, there is no impact and the authors appear to agree (l. 467). Would the main results for Filchner-Ronne change if only 1% of the GL were connected to the access depth? If indeed there is no impact of this distribution, what is the role of this analysis in the author's aim to quantify an upper limit of warm mode melting? If there is no impact, the authors should state this clearly and state clearly why they include this quantitative assessment. If there is an impact, the authors should explicitly discuss this in the results section.
    b. The discussion of the Amundsen Sea (Sec 3.3.4) is difficult to follow. The authors present Fig 10, showing present-day warm cavities, and in a single sentence in the middle of the section, point out this well-known characteristic. However, they only half address this when discussing temperatures. Clearly from Fig 10, T_CF,mean is an irrelevant metric for the Amundsen Sea. This should be stated explicitly. From there on, no

more results based on this irrelevant metric should be included. Instead, the authors present their temperature changes and melt increases as if they are unaware of their own finding. If the authors do not draw insights from their own results, it is difficult for a reader to do so. In this Section, I believe the authors should start from the common knowledge (already warm cavities) and reflect on what this means for their core narrative (assess change to warm cavities). Next, after quantifying $T_{CF,mean}$, they should conclude that this temperature is irrelevant, and hence refrain to temperature changes based on $T_{CF,max}$. In the embedding paragraph, they also refer to a study based on a change in $T_{CSB}$, which is in contrast with the author's core narrative that $T_{CSB}$ is constant; hence, this comparison is again irrelevant.

    c. I do like Fig 3, as it nicely visualises the amount of sheltering of different ice shelves through bathymetric constraints. Unfortunately, the authors do not use this figure for their interpretation of their results. Similar to Fig. 4, does this mean that Fig. 3 (except for the numbers) is irrelevant for their results? Or can they use this figure to interpret their results? The sheltering of Filchner-Ronne and Amery appears much more significant than that of Ross. Is this reflected in the results as possibly expected? Or is this not reflected against expectation? If the authors do not link their core results (Fig 6) to these previous results, the reader is tempted to conclude that there is no relation and is left wondering why these previous results are included at all, and what exactly is the core result.

    d. Similar to the previous point, the discussion of the different regions (Sec 3.3) is largely isolated. Each region gets a description of the main gateways, a quantification of temperatures and melt rates, and a comparison to previous literature. All further interpretatation is left to the reader. Fig 6c is a great tool to link the different regions, highlight the general strong sensitivity of the large, cold cavities (FRIS, Ross, Amery) and contrast this against the regions with already warm cavities (Getz – Western AP). In Sec 3.3, I would expect the authors to place the results of the different regions in the larger context and hence refer to Fig 6c (intercomparison) and ideally also Fig 3 and 4 (interpretation).

2. Presentation of aim/motivation/methods. The methodology is somewhat spread out into other sections. Also, the authors have done well in more clearly stating their motivation in, e.g., the title, but have not yet clearly linked their methodological choices to this motivation.

    a. Lines 64-70 in the introduction read like part of the methodology (definition of ocean gateways). Consider rewriting of reorganising to clearly separate intro from methods.

    b. Lines 83-86 also appear in the discussion. More critically, it is unclear to me how this paragraph adds to the reader's understanding of what your study is about.

    c. Rather than this paragraph, I would expect the introduction to end with a concise problem statement and motivation/aim: what can the reader expect to find in the remainder of this study? Please spend a few lines on this in the place where it's expected.

d. Also the buildup of the methodology is not directly logical because the motivation/aim has not been clearly formulated. It takes until the end of Sec 2.2 until it's clear that delta T is a main metric in this study. I strongly recommend that the authors write 1 paragraph at the start of the methodology linking their aforementioned motivation to the general methodolgy. Something along the lines of: 'We aim to quantify an upper bound of melting if cavities switch to a warm mode. To this extent, we will use PICO to compute melting from ocean temperatures. As a present-day estimate, we'll take T at the calving front. For the warm mode, we will take T at the shelf break. In order to account for bathymetric constraints, we will define gateways and derive the warm mode T at the access depth. In this section, we will first define ocean gateways and access depths (2.1), etc etc'. This way, the reader knows what's really happening and why. And the authors have a framework to easily structure the methodology.

e. Lines 182 – 197 are a methodological description and should move to the Methods section.

3. Presentation/organisation of results. The order of the results isn't fully logical in comparison to the general narrative. In the methods, the order is: bathymetry -> temperatures -> melt rates. This is in line with the stated motivation as well and is reflected in each of the subsections 3.3.x. But not in the overall results section, creating confusion about what the authors consider their main results.

a. I recommend trying to keep this logical order in the results section as a guideline as well. In principle, 3.1 is good as a general overview. After this, I'd put the subsections per region (now 3.3.x) keeping the current order. After that, I'd put a general section building on the current 3.2 which puts together the overall results in terms of the core objective: estimating warm-mode melt rates.

b. Also make sure your section titles properly reflect the contents. Right now, the overall majority of the results fall in 3.3 with the header 'ocean gateways'. However, in the title and overall narrative, these gateways are the constraint used to determine melt rates. Hence, the major results should be put in a section with a title containing 'melt'.

c. In the melt section, which I propose as 3.3, following the region-specific results, the bigger interpretation and intercomparison should take place such that your study forms one large narrative. In this section, you can address the points raised in 'major comment 1'.

4. 'oceanic gateways in 7 out of 19 regions'. This quantitative result is based on Fig 4 comes back in several places including conclusions and abstract. So the authors deem this a major result. However, the criterium is unclear, and its relevance to the core motivation/aim is also not discussed.

a. First, the description is quasi-quantitative (l. 107: a 'large fraction'). And it's unclear why these 7 regions are qualitatively different from the other 12. In Fig 4, Larsen looks very similar. How can the reader deduct from your results that Larsen does not contain an oceanic gateway?

b. More generally: if the focus on this quantitative number is so prominent, it should have a clear and unambiguous definition. Either visually from Fig 4, or quantitatively from your data, a reader should be able to reproduce

this number based on the information you've provided. Currently, this is not the case.

c. More generally, the authors have now stated their motivation/aim more clearly: estimating an upper bound on melt rates. There is no dicussion, however, how these melt rates are affected by whether or not a region has such a gateway. For example, the 7 regions do not stand out as a cluster in Fig. 6c. Does this mean the gateways are unimportant for the melt rate changes?

Minor comments

l. 37:   'The highest thinning rates'

l. 44:   Sign should be greater than (>)

l. 76:   'provides'

l. 186:   'previous studies'. Which?

l. 213 'are in the mean much lower'. Odd word order, rephrase.

l. 217 'in the mean'. Odd word order, rephrase.

l. 258 'we are further analysing' -> 'we further analyse'

l. 283 'near Getz' -> 'at Getz'

l. 310 'If' -> 'Whether'

l. 352 Is the uncertainty in bathymetry in this well-sampled region significant in comparison to the uncertainty due to methodological choices made in this study? So is this the dominant reason why these results should be taken with caution?

l. 374 The statement that Amundsen is already warm is very important for this whole section, but this sentence is dropped between paragraphs and then ignored. Make sure to write this section consistently around this well-known fact.

l. 446-458. I don't think this extensive discussion on bathymetric uncertainty is very important. Again: ask yourself: is this a dominant source of uncertainty in your results, compared to uncertainties related to methodological choices? And would that be resolved by a bathymetric dataset at 100x100m resolution? I don't think so. I suggest removing this paragraph and focusing on the dominant/significant stuff

l. 465   'It is to note': odd sentence, rephrase

l. 470   'spatially more resolved': very generic. What do this mean? Is this a limitation in the resolution you used for PICO, or a limitation of PICO itself?

l. 473   'less resolved' and 'more distributed'. Again generic wording, be specific. Note: this is not to criticise PICO or your choice to use PICO, this is valid. But be specific so that readers understand what you mean.

l. 476   Can cite Berends et al (2023) here as well: 10.1017/jog.2023.33

l. 481   'capture to some extent'. What does this mean? Again, be specific

l. 490   The regions in van der Linden et al are taken from Levermann 2020. Rather cite the original: https://doi.org/10.5194/esd-11-35-2020

l. 497   As stated before: your Amundsen Sea results clearly show that, at least in this case, the mean conditions are invalid. So yes, these give higher differences, but is that what you're looking for? A more critical discussion would state that, simply, your method is not designed to assess the switch from a warm to a warm cavity, and that the sensitivity in these regions is not dominated by a qualitative circulation change but by more gradual offshore changes that would affect T_CSB (which is beyond the scope of your study).

l. 514   'combine the latest'

l. 521   'two orders of magnitude' would mean at least a 100-fold increase. This is not what you've found.

l. 520   'parameterisation' -> 'parameterisations'

---

## Author Response (AR2)

**Point-to-point response to reviewer comments - Major Revisions (2. Iteration)**

*Dear Editor, dear reviewer,*

*Thank you again for finding the time to handle and review our manuscript as well as providing such constructive and thorough feedback to rewrite and restructure our manuscript. Following the reviewer's suggestions, we have revised the text, for instance, by re-ordering the different sections as proposed, we have sharpened our definition of what an "oceanic gateway" constitutes and tied our results more strongly together to provide further interpretation and coherence of our results.*

*Our point-to-point responses to the individual comments are given below in blue and italics compared to the reviewer comments which are given* in black without italic font. The Editor comment is given in *orange with italic font. The line numbers given in the responses refer to the revised manuscript of this round (without highlighted/tracked changes).*

**Editor comment**

*Dear Nicola et al. the reviewers noted considerable concerns with the original manuscript and in particular the interpretation of the results. This has been partly answered in the revisions. However, given that there are still key remaining issues I will only reconsider the manuscript after the comments from the second round of review have been addressed in full. Please make sure to respond to all the points, in particular, make sure to work on the interpretation of the results and the coherence of the writing.*

**Reviewer comments**

Second review of: Bathymetry-constrained warm-mode melt estimates derived from analysing Oceanic Gateways in Antarctica by Lena Nicola et al.

Date: 1 Oct 2024, Reviewed by: Erwin Lambert

The authors have put in a great effort to address my concerns raised in the first review round. The possible errors have been resolved, the methodology has become more logical and several metrics are now presented in a clearer and more concise way. Additionally, the authors have modified their figures to address several concerns.

I do, however, still have some concerns regarding the interpretation and writing which I believe should be improved before I can recommend this manuscript for publication. Although these concerns can be resolved by rewriting solely, without the need for new analysis, I still recommend a major revision. This is because I believe the depth of interpretation and coherence requires a considerable effort to raise the level of the manuscript to the level needed for publication in The Cryosphere. That being said, I am confident that the authors can reach this level based on the current state of the work and the suggestions below.

**Major comments:**

1. Coherence.

At points, the manuscript reads as separate individual sections/paragraphs without a strong coherence. This makes it difficult to interpret the results. In particular:

a. The authors provide a quantitative distribution of access depths (Fig 4).  These distributions are referred to in the main results section (Sec 3.3).  However, the impact of these distributions is not tied to the results in terms of temperature/melt changes. In fact, I believe through the use of  PICO, there is no impact and the authors appear to agree (l. 467). Would  the main results for Filchner-Ronne change if only 1% of the GL were  connected to the access depth? If indeed there is no impact of this  distribution, what is the role of this analysis in the author's aim to quantify  an upper limit of warm mode melting? If there is no impact, the authors   should state this clearly and state clearly why they include this quantitative assessment. If there is an impact, the authors should explicitly discuss this in the results section.

*We have included the  access depth analysis in our manuscript (1) to identify oceanic gateway features and (2) by this, gauge the validity of our melt results for the grounding line in each basin. PICO uses one value of temperature and salinity as input and applies the generated melt to the entirety of the region's grounding line. In all regions, we use the deepest access depths found along the grounding line to derive temperature changes in PICO.*
*The analysis of the quantitative distribution of access depths helps us to identify regions with "major" gateways (deepest access depth represents the biggest step in grounding line coverage). In those regions, similar to the case of Filchner Trough, the inflow of warm water masses could be channeled through this major 'oceanic gateway', i.e., where a change in circulation can lead to a full cavity switch from cold to warm conditions. As you note, whether such a gateway exists or not does not impact the melt quantification in PICO, we have added this to the text.*

*Specifically, we have changed/added the following:*
*In the methodology section l. 106ff:*
> *"We define the deepest access depth found along the grounding line of each basin as $d\_GL,0$ and express the fraction of how much the grounding line at that depth is connected to the open ocean with values ranging from 0% to 100%. If **a comparably large part of the grounding line is reached by only a small increase in vertical access level, an 'oceanic gateway' is present** i.e. a deep trough connecting the (overdeepened) ice-shelf cavity to the open ocean past the continental-shelf break. We thus interpret an oceanic gateway to be the horizontal pathway from the open ocean to the grounding line of the ice sheet along the deepest possible ocean-connection between the two. For each region, we **ascribe an oceanic gateway as 'major', if a global maximum (highest peak) in access depth along the grounding line is found at $d\_GL,0$.***

*In Section 3.1, l. 252f.:*
> *"Note **that the existence of an oceanic gateway does not influence the melt rates** calculated with PICO in Section 3.3, as we use $d\_GL,0$ for all regions. However,*

*it allows us to gauge the validity and limitations of our assumptions in each basin."*

*In the discussion (l. 487ff):*

> *"Please note that our results **are not directly dependent on the grounding line coverage** at the deepest access depth, but it enables us to contextualize the results. **Our temperature and melt rate changes would not differ if at d_GL,0 only 1% instead of a higher percentage were horizontally connected to the open ocean.** PICO uses one temperature (and salinity) estimate per basin to compute sub-shelf melt rates. However, the existence of a major oceanic gateway means that a substantial portion of the grounding line is reached at d_GL,0 and the PICO input values are a good representation of potential results. In the case of the Filchner--Ronne basin for instance, d_GL,0 reaches more than three quarters of the cavity and is thus, in our conclusion, adequately representative for the entire shelf e.g. for estimating a bathymetric-constrained 'warm'-mode melt estimate. In the other case, that no major oceanic gateway exists, PICO input values represent an upper bound on the oceanic properties that would reach the grounding line.*

b. The discussion of the Amundsen Sea (Sec 3.3.4) is difficult to follow. The authors present Fig 10, showing present-day warm cavities, and in a single sentence in the middle of the section, point out this well-known characteristic. However, they only half address this when discussing temperatures. Clearly from Fig 10, T_CF,mean is an irrelevant metric for the Amundsen Sea. This should be stated explicitly. From there on, no more results based on this irrelevant metric should be included. Instead, the authors present their temperature changes and melt increases as if they are unaware of their own finding. If the authors do not draw insights from their own results, it is difficult for a reader to do so. In this Section, I believe the authors should start from the common knowledge (already warm cavities) and reflect on what this means for their core narrative (assess change to warm cavities). Next, after quantifying T_CF,mean, they should conclude that this temperature is irrelevant, and hence refrain to temperature changes based on T_CF,max.

*We have rearranged the section so that we start, as suggested, by stating the well-known fact about the Amundsen Sea being in a warm-mode.*

*We follow your suggestion on T_CF,mean (see lines 368ff. in the revised manuscript) and use the difference dT_max-max in this region:*

> *"The highest temperatures along the calving front are found near Pine Island Glacier with a maximum temperature of T_CF, max=1.17°C. In contrast, the mean temperature of -0.23°C is influenced by much colder bottom temperatures found at Crosson Ice Shelf (-1.2°C; cf. Supplement Fig. S7). Considering that warm waters are already present within most of the Amundsen Sea's ice-shelf cavities, T_CF,mean can **thus be considered an unrepresentative metric** for deriving bathymetric constrained 'warm'-mode melt estimates in this region. For this region, we thus use the difference of T_CF,max to T_CSB,max, ΔTmax-max = 0.4°C, to derive an upper bound melt estimate i.e. assuming an inflow of more, unmodified, CDW to all grounding line parts of the region. See Fig. 10 for the other temperature differences, ΔTmean-mean and ΔTmax-mean to compare."*

*We have further clarified this by adjusting former Fig 05b, now Fig 10b of the revised manuscript:*

[Figure]

In the embedding paragraph, they also refer to a study based on a change in T_CSB, which is in contrast with the author's core narrative that T_CSB is constant; hence, this comparison is again irrelevant.

*We agree that mentioning the Goméz-Valdivia study that shows an increase in ocean temperatures reaching the Amundsen Sea region of 1.2°C does not tie in fully with our approach and removed it here. Since, we find it important to consider that even in already 'warm' regions i.e. where warm-mode melting dominates, temperatures can increase in the future.*

*Hence, we have added a slightly adjusted phrase in our discussion part l. 562ff:*

> *"When considering estimates on CDW-inflow driven sub-shelf melting, one has to consider, however, that ocean temperatures are projected to become warmer in the future, for instance, by 1.2°C as found by Gómez-Valdivia et al. (2023) that employ a global climate model on a relatively coarse resolution (1° ocean model)."*

c. I do like Fig 3, as it nicely visualises the amount of sheltering of different ice shelves through bathymetric constraints. Unfortunately, the authors do not use this figure for their interpretation of their results. Similar to Fig. 4, does this mean that Fig. 3 (except for the numbers) is irrelevant for their results? Or can they use this figure to interpret their results? The sheltering of Filchner-Ronne and Amery appears much more significant than that of Ross. Is this reflected in the results as possibly expected? Or is this not reflected against expectation? If the authors do not link their core results (Fig 6) to these previous results, the reader is tempted to conclude that there is no relation and is left wondering why these previous results are included at all, and what exactly is the core result.

*We have included Fig. 3 to visualize our access depth approach as we find it more insightful than a simple 2D map of the access depths, which we have included in Supplement Figure S2. Figure 3 adds information to Figure 4, as it shows how sheltered different cavities are, as you mention. In Figure 4 we show the distribution of access depth at the different region's grounding lines together with the access depth we use for deriving temperature and melt rate estimates (d_GL,0). While the melt estimates are the core result of our study (Fig. 6), the access depth analysis and discussing of oceanic gateways has an importance too. We have amended the text in several instance to tie our results both to Fig. 3 and 4:*

*In Results Section 3.1 (l. 217ff):*

*"At Amery, around 91% of the ice-shelf cavity is shielded by shallower bathymetry, i.e. the access depth is shallower than the topography in 91% of the cavity area. In contrast, this applies only to about a third of the cavity area for basins 7 or 17. We later see that this can be linked to the absence of any oceanic gateway structure in those regions."*

*And later in the same section lines 233 to 235:*

*"Filchner–Ronne, Ross and Amery are the regions where not only the grounding lines, but, together with Fimbul (basin 3) and Totten (basin 8), also the cavities are most shielded by shallower bathymetry (Fig. 3)."*

*And lines 248ff.*

*"In PICO basins 5 (Prince Harald), 19 (Eastern AP), 9 (Cook, Mertz), 15 (Abbot, Cosgrove), 16 (Bellingshausen Sea), 7 (West, Shackleton) and 17 (Western AP), we do not find a significant spike in the access depth distribution along the grounding line and hence conclude that no (seen from an Antarctic-wide scale) oceanic gateway is present. This is despite the fact that parts of the respective cavities are shielded by shallower bathymetry (in all regions at least a third of the cavity is shielded).*

*In line 465ff:*

*"Generally, our analysis highlights the generally strong sensitivity of the large, 'cold' cavities (Filchner–Ronne, Ross, Amery), which stands in contrast to regions with already warm cavities (Getz – Western AP). These 'cold' cavity-regions show 'major' oceanic gateways (Fig. 4) that **allow access into a well-shielded cavity** ( especially true in the case of Filchner–Ronne and Amery, cf. Fig. 3)."*

*In the conclusion (l. 589ff.):*

*"All regions would experience a strong increase in sub-shelf melting, while basal melt rates would increase up to 42-fold in cavities that are currently in a 'cold' state, are **well-shielded by shallower bathymetry and have a 'major' oceanic gateway** that could channel warmer water masses to the grounding lines."*

d. Similar to the previous point, the discussion of the different regions (Sec 3.3) is largely isolated. Each region gets a description of the main gateways, a quantification of temperatures and melt rates, and a comparison to previous literature. All further interpretation is left to the reader. Fig 6c is a great tool to link the different regions, highlight the general strong sensitivity of the large, cold cavities (FRIS, Ross, Amery) and contrast this against the regions with already warm cavities (Getz – Western AP). In Sec 3.3, I would expect the authors to place the results of the different regions in the larger context and hence refer to Fig 6c (intercomparison) and ideally also Fig 3 and 4 (interpretation).

*We have amended the paragraph with former Fig. 6c that now reads (l.457ff):*

*"We find the largest increase at Amery Ice Shelf, where melt rates could increase up to 42-fold, cf. Fig. 11c. With our access depth analysis in these three regions, we have found that, at the access depth used for extracting the temperatures at the CSB, $d\_GL,0$, more than 30% of the respective grounding lines are accessed. This is why we classified these regions to have 'major' oceanic gateways see Fig. 4. This gives our results significance, as in those regions, our assumptions with PICO are*

*particularly valid e.g. warm water from the CSB is channelled to the respective grounding lines."*

*It further reads:*
> *"Ice-shelf regions in West Antarctica do feature oceanic gateways as well (e.g. Getz with 9.8% of its grounding line reached at d_GL,0) but since there is already warm water present at the CF, there is little potential for as drastic changes in the sub-shelf melt rates through bathymetric constrained inflow as in 'cold' cavities at present. Generally, our analysis highlights the general strong sensitivity of the large, 'cold' cavities (Filchner–Ronne, Ross, Amery), which stands in contrast to regions with already warm cavities (Getz – Western AP). These 'cold' cavity-regions show 'major' oceanic gateways (Fig. 4) that allow access into a well-shielded cavity and to the grounding line (the former is especially true in the case of Filchner–Ronne and Amery, cf. Fig. 3). "*

2. Presentation of aim/motivation/methods.

The methodology is somewhat spread out into other sections. Also, the authors have done well in more clearly stating their motivation in, e.g., the title, but have not yet clearly linked their methodological choices to this motivation.
a. Lines 64-70 in the introduction read like part of the methodology  (definition of ocean gateways). Consider rewriting of reorganising to clearly separate intro from methods.
*We have restructured and rewritten the methods as suggested.*

b. Lines 83-86 also appear in the discussion. More critically, it is unclear to me how this paragraph adds to the reader's understanding of what your  study is about.
*Here we wanted to refer to our follow-up study Kreuzer et al 2024 that uses our oceanic gateway concept. We see that it is not appropriate to mention it here and hence deleted it from the introduction.*

c. Rather than this paragraph, I would expect the introduction to end with a concise problem statement and motivation/aim: what can the reader  expect to find in the remainder of this study? Please spend a few lines on this in the place where it's expected.
*Done.  The revised passage in the text now reads (l. 79ff.):  "In this study, we aim to estimate the impact of potential future warm water inflow on basal melting for all Antarctic regions. In order to do so, we (1) analyse the bathymetry and identify trough-like features that potentially provide access of off-shore warm waters into ice-shelf cavities, (2) calculate the increase in thermal forcing resulting from such a regime shift and (3) compute the respective increase in sub-shelf melting."*

d. Also the buildup of the methodology is not directly logical because the  motivation/aim has not been clearly formulated. It takes until the end of  Sec 2.2 until it's clear that delta T is a main metric in this study. I strongly  recommend that the authors write 1 paragraph at the start of the methodology linking their aforementioned motivation to the general  methodology. Something along the lines of: *'We aim to quantify an upper  bound of melting if cavities switch to a warm mode. To this extent, we will  use PICO to compute melting from ocean temperatures. As a present-day  estimate, we'll take T at the calving front. For the warm mode, we will take  T at the shelf break. In order to account for bathymetric constraints, we*

*will define gateways and derive the warm mode T at the access depth. In this section, we will first define ocean gateways and access depths (2.1), etc etc*'. This way, the reader knows what's really happening and why. And the authors have a framework to easily structure the methodology.

*Done. The beginning of the methodology section now reads (l . 86) :*

*"The goal of our approach is to quantify an upper bound to melting if cavities switch to a 'warm' mode. To this extent, we use PISM-PICO to compute ice-shelf basal melting for given ocean temperatures and salinities: As a present-day estimate, we take temperatures at the calving front. For the 'warm' mode, we use temperatures at mid-depth at the continental-shelf break (CSB). In order to constrain that latter depth and to estimate the potential impacts of this selection on the melting at the grounding line (accounting for bathymetric constraints), we define oceanic gateways based on access depths found in each region."*

e. Lines 182 – 197 are a methodological description and should move to the Methods section.

*Done.*

3. Presentation/organisation of results.

The order of the results isn't fully logical in comparison to the general narrative. In the methods, the order is: bathymetry -> temperatures -> melt rates. This is in line with the stated motivation as well and is reflected in each of the subsections 3.3.x. But not in the overall results section, creating confusion about what the authors consider their main results. a. I recommend trying to keep this logical order in the results section as a guideline as well. In principle, 3.1 is good as a general overview. After this, I'd put the subsections per region (now 3.3.x) keeping the current order. After that, I'd put a general section building on the current 3.2 which puts together the overall results in terms of the core objective: estimating warm-mode melt rates.

*Thank you for this suggestion to restructure our text. We have reordered the sections accordingly and hopefully now provide a more coherent presentation of our analysis.*

b. Also make sure your section titles properly reflect the contents. Right now, the overall majority of the results fall in 3.3 with the header 'ocean gateways'. However, in the title and overall narrative, these gateways are the constraint used to determine melt rates. Hence, the major results should be put in a section with a title containing 'melt'.

*Thank you. We have changed the section titles in the results section to:*

c. In the melt section, which I propose as 3.3, following the region-specific results, the bigger interpretation and intercomparison should take place such that your study forms one large narrative. In this section, you can  address the points raised in 'major comment 1'.
*Addressed. Please refer to the section in the revised manuscript.*

4. 'oceanic gateways in 7 out of 19 regions'.

This quantitative result is based on Fig 4 comes back in several places including conclusions and abstract. So the authors deem this a major result. However, the criterium is unclear, and its  relevance to the core motivation/aim is also not discussed.

a. First, the description is quasi-quantitative (l. 107: a 'large fraction'). And  it's unclear why these 7 regions are qualitatively different from the other  12. In Fig 4, Larsen looks very similar. How can the reader deduct from your results that Larsen  does not contain an oceanic gateway?
*Thank you for making us aware of this confusing/incoherent aspect. We have revised the respective section in the manuscript including Figure 04, to clarify our analysis approach. When the largest spike in the access depth distribution along the grounding line occurs at d_GL,0 we call it a "major" gateway. Larsen does show an oceanic gateway - a major oceanic gateway is found that reaches around 11% of the region's grounding line at the deepest access depth d_GL,0. To avoid further confusion, we have revised the definition and included the respective explanation in the methodology section:*

*l. (108ff): "If a comparably large part of the grounding line is reached by only a small increase in vertical access level, an 'oceanic gateway' is present i.e. a deep trough connecting the (overdeepened) ice-shelf cavity to the open ocean past the continental-shelf break. We thus interpret an oceanic gateway to be the horizontal pathway from the open ocean to the grounding line of the ice sheet along the deepest possible ocean-connection between the two. For each region, we ascribe an oceanic gateway as 'major', if a global maximum (highest peak) in access depth along the grounding line is found at d_GL,0."*

*Figure 04 now includes additional labels indicating the found gateways and the regions (sub-panels of Fig.04) are sorted according to the accessed percentage of total grounding line i.e. the "majority" of the gateway so-to-speak for the region at d_GL,0, the depth we use later to derive melt changes.*

*7 out of 19 regions thus have a major gateway at d_GL,0 and we state this result throughout the text. In the abstract we revised it to "Here we identify potential oceanic gateways in at least 7 out of 19 regions subdividing the Antarctic continent" to acknowledge those regions that incorporate more than one ice shelf with a distinct peak (but not at d_GL,0) in the access depth distribution.*

b. More generally: if the focus on this quantitative number is so prominent, it  should **have a clear and unambiguous definition.** Either visually from Fig  4, or quantitatively from your data, a reader should be able to reproduce this number based on the information you've provided. Currently, this is  not the case.
*See above. We have revised the definition of a (major) 'oceanic gateway'. 7 out of 19 PICO regions have the highest peak in the access depth distribution at their deepest grounding line*

*access depth, d_GL,0.*

c. More generally, the authors have now stated their motivation/aim more clearly: estimating an upper bound on melt rates. There is **no discussion**, however, how these melt rates are affected by whether or not a region has such a gateway. For example, the 7 regions do not stand out as a cluster in Fig. 6c. Does this mean the gateways are unimportant for the melt rate changes?

*Thank you for pointing this out to us and we agree that we have not made ourselves clear enough. We have now more clearly stated that when a 'major' oceanic gateway is present, our results with PICO are more representative for the entire basin. In other regions our results represent an upper bound estimate, because only a very small amount of the grounding line is accessed at the deep access depth we use for deriving melt changes, but we assume that it is valid for the entire region. Using the temperature estimates from that depth (i.e. where warm CDW resides off the continental shelf), thus creates a very large melt increase for the entire region.*

*To clarify our findings further, in the conclusion section we have added in lines 550ff:*
*" Using the 19 PICO basins in our study, we conclude for basins 1, 6, 12, 14, 11, 18 and 13 (7 out of 19 regions; cf. Fig.4), that d_GL,0 is representative for the entire basin in the case of an oceanic-gateway driven switch to 'warm'-mode melt conditions, as d_GL,0 represents the largest grounding line share. In other regions (e.g. in basins 3, 2, 10), however, d_GL,0 and subsequent temperature offsets and melt rates based on that estimate are less representative for the entire basin but constitute even more an upper-bound estimate as d_GL,0 is lower, hence represents warmer CDW. This could be fixed by simulating each individual ice shelf separately. Finer resolutions i.e. on the individual ice shelf level would reveal more individual gateways but this analysis is out of scope of this manuscript".*

**Minor comments**

l. 37: 'The highest thinning rates'
*Corrected.*

l. 44: Sign should be greater than (>)
*We have cut this from the revised manuscript.*

l. 76: 'provides'
*Thank you. We have deleted this part as we have rewritten this section for the revised manuscript.*

l. 186: 'previous studies'. Which?
*Added Reese et al 2018, Reese et al 2023, Sutter et al. 2023 and Wirths et al. 2024.*

*The revised section of the manuscript (l. 195ff) now reads:*
> *"The boundaries on land are based on ice drainage basins from Zwally et al. (2012), were consolidated to 19 regions in Reese et al. (2018a) and for the use for PICO mainly extended along meridians into the ocean. In previous studies (Reese et al., 2018a, 2023; Sutter et al., 2023; Wirths et al., 2024), those basin boundaries in the ocean were used to extract a basin average for temperature and salinity (i.e. average over the region) to feed into the box model."*

l. 213 'are in the mean much lower'. Odd word order, rephrase.
*Changed to: " Average temperatures along the ice-shelf fronts, $T\_CF$, mean, which are derived at the ocean floor in the individual basins, are lower than temperatures found at the relevant access depth at the continental-shelf break, $T\_CSB$, mean, see Figure 10."*

l. 217 'in the mean'. Odd word order, rephrase.
*Changed to: "Mean $T\_CF$ estimates range" …*

l. 258 'we are further analysing' -> 'we further analyse'
*Corrected.*

l. 283 'near Getz' -> 'at Getz'
*Corrected.*

l. 310 'If' -> 'Whether'
*Corrected.*

l. 352 Is the uncertainty in bathymetry in this well-sampled region significant in comparison to the uncertainty due to methodological choices made in this study? So is this the dominant reason why these results should be taken with caution?
*Thank you for this comment. We agree that our methodology only provides an overall approximation to gauge warm-mode melt estimates around Antarctica. We believe, however, that reviewing our methodology i.e. providing a critical view on the connected-component analysis is important as well. We have tested whether the different routing of waters (according to our connected-component analysis) yields a difference: the access depth at the Amundsen Sea would only be around 25m shallower. For clarity, we have cut this part of the revised manuscript.*

l. 374 The statement that Amundsen is already warm is very important for this whole section, but this sentence is dropped between paragraphs and then ignored. Make sure to write this section consistently around this well-known fact.
*We have placed this statement now in the beginning of the subsection covering the ice shelves in the Amundsen Sea. Section 3.2.4 now begins with: "At present, ice shelves in the Amundsen Sea have 'warm' cavities and therefore dominate the current mass loss in Antarctica (see e.g. Pritchard et al., 2012), indicating that this region is already out of balance with the current oceanic forcing. Here, comparably warm water masses have already found their way underneath the ice shelves, in contrast to the three ice-shelf regions detailed before."*

l. 446-458. I don't think this extensive discussion on bathymetric uncertainty is very important. Again: ask yourself: is this a dominant source of uncertainty in your results, compared to uncertainties related to methodological choices? And would that be resolved by a bathymetric dataset at 100x100m resolution? I don't think so. I suggest removing this paragraph and focusing on the dominant/significant stuff
*Thank you for the suggestion. We have removed the paragraph.*

l. 465 'It is to note': odd sentence, rephrase
*Changed to: "It should be noted that…"*

l. 470: [Quote from manuscript: "With a spatially more resolved approach, one could apply the extracted temperature forcing to only those parts of the grounding lines that are connected to the open ocean at the deepest access depth found at the ice-shelf region's grounding line."]

'spatially more resolved': very generic. What do this mean? Is this a limitation in the resolution you used for PICO, or a limitation of PICO itself?

*Rephrased so that lines 521-523 now read: "With a spatially more explicit approach, with which one could provide temperature (and melt) locally to each grid cell, one could apply the extracted temperature offset to only those parts of the grounding lines that are connected to the open ocean at the deepest access depth found at the region's grounding line."*

l. 473 'less resolved' and 'more distributed'. Again generic wording, be specific. Note: this is not to criticise PICO or your choice to use PICO, this is valid. But be specific so that readers understand what you mean.

*Rephrased to "Furthermore, the melt pattern in PICO is spatially less variable than in ocean circulation models or observations, which means that PICO does not reach the very high melting at the order of 100 m yr−1 reported close to grounding lines (Dutrieux et al., 2014; Paolo et al., 2015).", see lines 524-527.*

l. 476 Can cite Berends et al (2023) here as well: 10.1017/jog.2023.33
*Done.*

l. 481 'capture to some extent'. What does this mean? Again, be specific
*Sorry for the confusion. We have changed it to: "to capture the parameter uncertainty", because later in the sentence we already acknowledge that a full ensemble is needed to explore the uncertainty in full.*

l. 490 The regions in van der Linden et al are taken from Levermann 2020. Rather cite the original: https://doi.org/10.5194/esd-11-35-2020
*Changed to: "[...] as for example in van der Linden et al. (2023), following Levermann et al. (2020), [...]" in the revised manuscript (l. 543).*

l. 497 As stated before: your Amundsen Sea results clearly show that, at least in this case, the mean conditions are invalid. So yes, these give higher differences, but is that what you're looking for? A more critical discussion would state that, simply, your method is **not designed to assess the switch from a warm to a warm cavity**, and that the sensitivity in these regions is not dominated by a qualitative circulation change but by more gradual offshore changes that would affect T_CSB (which is beyond the scope of your study).

*Together with addressing your comment from 1b) we have changed this part in the discussion that it now reads (cf. l. 557ff):*

*"When it comes to the effects of the potential warm water inflow, as analysed in our study, the difference in temperatures is small in some regions for physical reasons: this can be the case if the access depth of the basin is shallow and encompasses slightly colder water masses at the CSB, i.e. representing surface waters not CDW, or if the calving front temperatures are already relatively warm, as in the case of the Amundsen region. In those regions, changes in melting may be more sensitive to gradual offshore changes in continental-shelf break temperatures instead of a qualitative circulation change, i.e. a regime shift of cavity inflow leading to a switch from a 'cold' to a 'warm' cavity, **which our method is designed to assess**. When considering estimates on CDW-inflow driven sub-shelf melting,*

*one has to consider, however, that ocean temperatures are projected to become warmer in the future, for instance, by 1.2°C as found by Gómez-Valdivia et al. (2023) that employ a global climate model on a relatively coarse resolution (1° ocean model)."*

l. 514 'combine the latest'
*Due to newer publications, e.g. Charrassin et al., 2025 (https://www.nature.com/articles/s41598-024-81599-1), we decided to cut "latest". It now reads: "We combine available bathymetry data [...]"*

l. 521 'two orders of magnitude' would mean at least a 100-fold increase. This is not what you've found.
*Thank you for spotting this typo. We have corrected it to "would increase up to 42-fold".*

l. 520 'parameterisation' -> 'parameterisations'
*Corrected.*